# Revisiting the Minimalist Approach to Offline Reinforcement Learning

**Denis Tarasov  Vladislav Kurenkov  Alexander Nikulin  Sergey Kolesnikov**
Tinkoff
{den.tarasov, v.kurenkov, a.p.nikulin, s.s.kolesnikov}@tinkoff.ai

## Abstract

Recent years have witnessed significant advancements in offline reinforcement learning (RL), resulting in the development of numerous algorithms with varying degrees of complexity. While these algorithms have led to noteworthy improvements, many incorporate seemingly minor design choices that impact their effectiveness beyond core algorithmic advances. However, the effect of these design choices on established baselines remains understudied. In this work, we aim to bridge this gap by conducting a retrospective analysis of recent works in offline RL and propose ReBRAC, a minimalistic algorithm that integrates such design elements built on top of the TD3+BC method. We evaluate ReBRAC on 51 datasets with both proprioceptive and visual state spaces using D4RL and V-D4RL benchmarks, demonstrating its state-of-the-art performance among ensemble-free methods in both offline and offline-to-online settings. To further illustrate the efficacy of these design choices, we perform a large-scale ablation study and hyperparameter sensitivity analysis on the scale of thousands of experiments.[1]

## 1  Introduction

Interest of the reinforcement learning (RL) community in the offline setting has led to a myriad of new algorithms specifically tailored to learning highly performant policies without the ability to interact with an environment (Levine et al., 2020; Prudencio et al., 2022). Yet, similar to the advances in online RL (Engstrom et al., 2020; Henderson et al., 2018), many of those algorithms come with an added complexity – design and implementation choices beyond core algorithmic innovations, requiring a delicate effort in reproduction, hyperparameter tuning, and causal attribution of performance gains.

Indeed, the issue of complexity was already raised in the offline RL community by Fujimoto & Gu (2021); the authors highlighted veiled design and implementation-level adjustments (e.g., different architectures or actor pre-training) and then demonstrated how a simple behavioral cloning regularization added to the TD3 (Fujimoto et al., 2018) constitutes a strong baseline in the offline setting. This minimalistic and uncluttered algorithm, TD3+BC, has become a de-facto standard baseline to be compared against. Indeed, most new algorithms juxtapose against it and claim significant gains over (Akimov et al., 2022; An et al., 2021; Nikulin et al., 2023; Wu et al., 2022; Chen et al., 2022b; Ghasemipour et al., 2022). However, application of newly emerged design and implementation choices to this baseline is still missing.

In this work, we build upon Fujimoto & Gu (2021) line of research and ask: *what is the extent to which newly emerged minor design choices can advance the minimalistic offline RL algorithm?* The answer is illustrated in Figure 1: we propose an extension to TD3+BC, ReBRAC (Section 3), that simply adds on recently appeared design decisions upon it. We test our algorithm on both proprioceptive and

---

[1]Our implementation is available at `https://github.com/DT6A/ReBRAC`

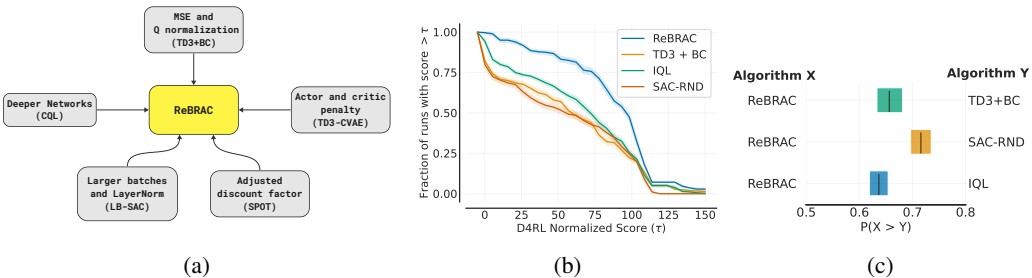

Figure 1: (a) The schema of our approach ReBRAC (b) Performance profiles (c) Probability of improvement. The curves (Agarwal et al., 2021) are for D4RL benchmark spanning all Gym-MuJoCo, AntMaze, and Adroit datasets (Fu et al., 2020).

visual state space problems using D4RL (Fu et al., 2020) and V-D4RL (Lu et al., 2022) benchmarks (Section 4) demonstrating its state-of-the-art performance across ensemble-free methods. Moreover, our approach demonstrates state-of-the-art performance in offline-to-offline setup on D4RL datasets (Section 4.3) while not being specifically designed for this setup. To further highlight the efficacy of the proposed modifications, we then conduct a large-scale ablation study (Section 4.4). We hope the described approach can serve as a strong baseline under different hyperparameter search budgets (Section 4.6), further accentuating the importance of seemingly minor design choices introduced along with core algorithmic innovations.

## 2 Preliminaries

### 2.1 Offline Reinforcement Learning

A standard Reinforcement Learning problem is defined as a Markov Decision Process (MDP) with the tuple $\{S, A, P, R, \gamma\}$, where $S \subset \mathbb{R}^n$ is the state space, $A \subset \mathbb{R}^m$ is the action space, $P : S \times A \to S$ is the transition function, $R : S \times A \to \mathbb{R}$ is the reward function, and $\gamma \in (0, 1)$ is the discount factor. The ultimate objective is to find a policy $\pi(a|s)$ that maximizes the cumulative discounted return $\mathbb{E}_\pi \sum_{t=0}^{\infty} \gamma^t R(s_t, a_t)$. This policy improves by interacting with the environment, observing states, and taking actions that provide rewards.

In offline RL, policies cannot interact with the environment and can only access a static transaction dataset $D$ collected by one or more other policies. This setting presents new challenges, such as estimating values for state-action pairs not included in the dataset while exploration is unavailable (Levine et al., 2020).

### 2.2 Behavior Regularized Actor-Critic

Behavior Regularized Actor-Critic (BRAC) is an offline RL framework introduced in Wu et al. (2019). The core idea behind BRAC is that actor-critic algorithms can be penalized in two ways to solve offline RL tasks: actor penalization and critic penalization. In this framework, the actor objective is represented as in Equation (1), and the critic objective as in Equation (2), where $F$ is a divergence function between dataset actions and policy actions distributions. The differences from a vanilla actor-critic are highlighted in blue.

$$\pi = \operatorname*{argmax}_{\pi} \mathbb{E}_{(s,a) \sim D} \left[ Q_\theta(s, \pi(s)) - \alpha \cdot F(\pi(s), a) \right] \tag{1}$$

$$\theta = \operatorname*{argmin}_{\theta} \mathbb{E}_{\substack{(s,a,r,s',\hat{a}') \sim D \\ a' \sim \pi(s')}} \left[ (Q_\theta(s, a) - (r + \gamma(Q_{\bar{\theta}}(s', a') - \alpha \cdot F(a', \hat{a}'))))^2 \right] \tag{2}$$

In the original work, various choices of $F$ were evaluated when used as the regularization term for the actor or critic. The authors tested KL divergence, Kernel MMD, and Wasserstein distance but did not observe any consistent advantage. Finally, it is essential to note that, originally, both regularizations coefficients had the same weight.

Subsequently, TD3+BC (Fujimoto & Gu, 2021) was introduced, utilizing Mean Squared Error (MSE) as the regularization term $F$ for the actor. TD3+BC is considered to be the minimalist approach to offline RL as it modifies existing RL algorithms by simply adding behavior cloning term into actor loss which is easy to implement and does not bring any significant computational overhead.

## 3  ReBRAC: Distilling Key Design Choices

In this section, we describe the proposed method along with the discussion of the new design choices met in the offline RL literature (Table 1). Our approach is a more general version of BRAC built on top of the TD3+BC (Fujimoto & Gu, 2021) algorithm with different modifications in design while keeping it simple (Figure 1). We refer to our method as **Re**visited **BRAC** (ReBRAC).

Table 1: Adoption of implementation and design choices beyond core algorithmic advancements in some recently introduced algorithms.

| Modification | TD3+BC | CQL | EDAC | MSG | CNF | LB-SAC | SAC-RND |
|---|---|---|---|---|---|---|---|
| Deeper networks | ✗ | ✓ | ✓ | ✓ | ✓ | ✓ | ✓ |
| Larger batches | ✗ | ✗ | ✗ | ✗ | ✓ | ✓ | ✓ |
| Layer Normalization | ✗ | ✗ | ✗ | ✗ | ✗ | ✓ | ✓ |
| Decoupled penalization | ✗ | ✗ | ✗ | ✗ | ✗ | ✗ | ✓ |
| Adjusted discount factor | ✗ | ✗ | ✗ | ✗ | ✗ | ✗ | ✓ |

**Deeper Networks**    The use of deeper neural networks has been a critical factor in the success of many Deep Learning models, with model quality generally increasing as depth increases, provided there is enough data to support this scaling (Kaplan et al., 2020). Similarly, recent studies in RL (Neumann & Gros, 2022; Sinha et al., 2020) and offline RL (Lee et al., 2022; Kumar et al., 2022) have demonstrated the importance of depth in achieving high performance. Although most offline RL algorithms are based on SAC (Haarnoja et al., 2018) or TD3 (Fujimoto et al., 2018), which by default employ two hidden layers, recent work (Kumar et al., 2020) uses three hidden layers for SAC instead, which appears to be an important change (Fujimoto & Gu, 2021). The change in network size has been adopted by later works (An et al., 2021; Yang et al., 2022; Zhuang et al., 2023; Nikulin et al., 2023) and may be one of the critical modifications that improve final performance.

The original BRAC and TD3+BC algorithms used only two hidden layers for their actor and critic networks, while most state-of-the-art solutions use deeper networks. Specifically, three hidden layers have become a common choice for recent offline RL algorithms. In ReBRAC, we follow this trend and use three hidden layers for the actor and critic networks. Additionally, we provide an ablation study in Section 4.5 to investigate the effect of the number of layers on ReBRAC's performance.

**LayerNorm**    LayerNorm (Ba et al., 2016) is a widely used technique in deep learning that helps improve network convergence. In Hiraoka et al. (2021), authors add dropout and LayerNorm to different RL algorithms, notably boosting their performance. This technique is also applied in Smith et al. (2022), and it appears that boost is achieved primarily because of LayerNorm. Specifically, in offline RL, various studies have tested the effect of normalizations between layers (Bhatt et al., 2019; Kumar et al., 2022; Nikulin et al., 2022, 2023). A parallel study by Ball et al. (2023) empirically shows that LayerNorm helps to prevent catastrophic value extrapolation for the Q function when using offline datasets in online RL. Following these works, in our approach, we also apply LayerNorm between each layer of the critic networks.

**Larger Batches**    Another technique to accelerate neural network convergence is large batch optimization (You et al., 2017, 2019). While studying batch sizes larger than 256 is limited, some prior works have used them. For instance, the convergence of SAC-N was accelerated in Nikulin et al. (2022). More recently proposed algorithms also use larger batches for training, although without providing detailed analyses (Akimov et al., 2022; Nikulin et al., 2023).

The usage of large batches in offline RL is still understudied, and its benefits and limitations are not fully understood. Our experiments show that in some domains, using large batches can lead to significant performance improvements, while in others, it might not have a notable impact or

even drop the performance (see Table 8). We increased the batch size to 1024 samples and scaled the learning rate for D4RL Gym-MuJoCo tasks, following the approach proposed by Nikulin et al. (2022).

**Actor and critic penalty decoupling** The original BRAC framework proposed penalizing the actor and the critic with the same magnitude. Most of the previous algorithms restrict only actor (Fujimoto & Gu, 2021; Wu et al., 2022) or only critic (Kumar et al., 2020; An et al., 2021; Ghasemipour et al., 2022). TD3-CVAE (Rezaeifar et al., 2022) penalizes both using the same coefficients, while another study, SAC-RND (Nikulin et al., 2023), shows that decoupling the penalization in offline RL is beneficial for algorithm performance, although ablations on using only one of the penalties are missing.

Our method allows simultaneous penalization of actor and critic with decoupled parameters. Inspired by TD3+BC (Fujimoto & Gu, 2021), Mean Squared Error (MSE) is used as a divergence function $F$, which we found simple and effective. The actor objective is shown in Equation (3), and the critic objective is shown in Equation (4). Differences from the original actor-critic are highlighted in red. Following TD3+BC (Fujimoto & Gu, 2021), the Q function is normalized to make the algorithm less sensitive to regularization parameters. Nonetheless, we forego the utilization of state normalization, as initially suggested in TD3 + BC, driven by our intention to execute the algorithm online and the observation that this adjustment typically results in negligible impact. Since our approach principally builds upon TD3+BC, the differences in their performances should be considered the most important ones and do not appear only because of the additional hyperparameters search.

$$\pi = \operatorname*{argmax}_{\pi} \mathbb{E}_{(s,a)\sim D} \left[ Q_\theta(s, \pi(s)) - \beta_1 \cdot (\pi(s) - a)^2 \right] \quad (3)$$

$$\theta = \operatorname*{argmin}_{\theta} \mathbb{E}_{\substack{(s,a,r,s',\hat{a}')\sim D \\ a'\sim\pi(s')}} \left[ (Q_\theta(s, a) - (r + \gamma(Q_{\overline{\theta}}(s', a') - \beta_2 \cdot (a' - \hat{a}')^2)))^2 \right] \quad (4)$$

**Discount factor $\gamma$ value change** The choice of discount factor is an important aspect in solving RL problems (Jiang et al., 2015). A recent study (Hu et al., 2022) suggests that decreasing the default value of $\gamma$ from 0.99 may lead to better results in offline RL settings. In contrast, in SPOT (Wu et al., 2022), the authors increased the value of $\gamma$ up to 0.995 when fine-tuning AntMaze tasks with sparse rewards, which resulted in state-of-the-art solutions. Similarly, in the offline setting SAC-RND (Nikulin et al., 2023), increasing $\gamma$ also achieved high performance on the same domain. The choice of increased $\gamma$ for AntMaze tasks was motivated by the sparse reward, i.e., a low $\gamma$ value may not propagate the training signal well. However, further ablations are needed to understand if the change in the parameter is directly responsible for the improved performance. In our experiments, we also find that increasing $\gamma$ from the default value of 0.99 to 0.999 is vital for improved performance on this set of tasks (see Table 8).

## 4 Experiments

### 4.1 Evaluation on offline D4RL

We evaluate the proposed approach on three sets of D4RL tasks: Gym-MuJoCo, AntMaze, and Adroit. For each domain, we consider all of the available datasets. We compare our results to several ensemble-free baselines, including TD3+BC (Fujimoto & Gu, 2021), IQL (Kostrikov et al., 2021), CQL (Kumar et al., 2020) and SAC-RND (Nikulin et al., 2023).

The majority of the hyperparameters are adopted from TD3+BC, while $\beta_1$ and $\beta_2$ parameters from Equation (3) and Equation (4) are tuned. We examine the sensitivity to these parameters in Section 4.6. For a complete overview of the experimental setup and details, see Appendix A.

Following Wu et al. (2022), we tune hyperparameters over four seeds (referred to as training seeds) and evaluate the best parameters over ten new seeds (referred to as unseen training seeds), reporting the average performance of the last checkpoints for D4RL tasks. On V-D4RL, we use two and five seeds, respectively. This helps to avoid overfitting during hyperparameters search and outputs more just and reproducible results. For a fair comparison, we tune TD3+BC and IQL following the same protocol. We also rerun SAC-RND on Gym-MuJoCo and AntMaze tasks and tune it for the Adroit

Table 2: Average normalized score over the final evaluation and ten unseen training seeds on Gym-MuJoCo tasks. CQL scores were taken from An et al. (2021). The symbol ± represents the standard deviation across the seeds. To make a fair comparison against TD3+BC and IQL, we extensively tuned their hyperparameters.

| Task Name | TD3+BC | IQL | CQL | SAC-RND | ReBRAC, our |
|---|---|---|---|---|---|
| halfcheetah-random | $\underline{30.9} \pm 0.4$ | $19.5 \pm 0.8$ | $\mathbf{31.1} \pm 3.5$ | $27.6 \pm 2.1$ | $29.5 \pm 1.5$ |
| halfcheetah-medium | $54.7 \pm 0.9$ | $50.0 \pm 0.2$ | $46.9 \pm 0.4$ | $\mathbf{66.4} \pm 1.4$ | $\underline{65.6} \pm 1.0$ |
| halfcheetah-expert | $93.4 \pm 0.4$ | $95.5 \pm 2.1$ | $97.3 \pm 1.1$ | $102.6 \pm 4.2$ | $\mathbf{105.9} \pm 1.7$ |
| halfcheetah-medium-expert | $89.1 \pm 5.6$ | $92.7 \pm 2.8$ | $95.0 \pm 1.4$ | $\mathbf{108.1} \pm 1.5$ | $\underline{101.1} \pm 5.2$ |
| halfcheetah-medium-replay | $45.0 \pm 1.1$ | $42.1 \pm 3.6$ | $45.3 \pm 0.3$ | $\mathbf{51.2} \pm 3.2$ | $\underline{51.0} \pm 0.8$ |
| halfcheetah-full-replay | $75.0 \pm 2.5$ | $75.0 \pm 0.7$ | $76.9 \pm 0.9$ | $\underline{81.2} \pm 1.3$ | $\mathbf{82.1} \pm 1.1$ |
| hopper-random | $8.5 \pm 0.7$ | $\underline{10.1} \pm 5.9$ | $5.3 \pm 0.6$ | $\mathbf{19.6} \pm 12.4$ | $8.1 \pm 2.4$ |
| hopper-medium | $60.9 \pm 7.6$ | $65.2 \pm 4.2$ | $61.9 \pm 6.4$ | $\underline{91.1} \pm 10.1$ | $\mathbf{102.0} \pm 1.0$ |
| hopper-expert | $\underline{109.6} \pm 3.7$ | $108.8 \pm 3.1$ | $106.5 \pm 9.1$ | $\mathbf{109.8} \pm 0.5$ | $100.1 \pm 8.3$ |
| hopper-medium-expert | $87.8 \pm 10.5$ | $85.5 \pm 29.7$ | $96.9 \pm 15.1$ | $\mathbf{109.8} \pm 0.6$ | $\underline{107.0} \pm 6.4$ |
| hopper-medium-replay | $55.1 \pm 31.7$ | $89.6 \pm 13.2$ | $86.3 \pm 7.3$ | $\underline{97.2} \pm 9.0$ | $\mathbf{98.1} \pm 5.3$ |
| hopper-full-replay | $97.9 \pm 17.5$ | $104.4 \pm 10.8$ | $101.9 \pm 0.6$ | $\mathbf{107.4} \pm 0.8$ | $\underline{107.1} \pm 0.4$ |
| walker2d-random | $2.0 \pm 3.6$ | $11.3 \pm 7.0$ | $5.1 \pm 1.7$ | $\mathbf{18.7} \pm 6.9$ | $\underline{18.4} \pm 4.5$ |
| walker2d-medium | $77.7 \pm 2.9$ | $80.7 \pm 3.4$ | $79.5 \pm 3.2$ | $\mathbf{92.7} \pm 1.2$ | $\underline{82.5} \pm 3.6$ |
| walker2d-expert | $\underline{110.0} \pm 0.6$ | $96.9 \pm 32.3$ | $109.3 \pm 0.1$ | $104.5 \pm 22.8$ | $\mathbf{112.3} \pm 0.2$ |
| walker2d-medium-expert | $110.4 \pm 0.6$ | $\mathbf{112.1} \pm 0.5$ | $109.1 \pm 0.2$ | $104.6 \pm 11.2$ | $\underline{111.6} \pm 0.3$ |
| walker2d-medium-replay | $68.0 \pm 19.2$ | $75.4 \pm 9.3$ | $76.8 \pm 10.0$ | $\mathbf{89.4} \pm 3.8$ | $\underline{77.3} \pm 7.9$ |
| walker2d-full-replay | $90.3 \pm 5.4$ | $97.5 \pm 1.4$ | $94.2 \pm 1.9$ | $\mathbf{105.3} \pm 3.2$ | $\underline{102.2} \pm 1.7$ |
| Average | $70.3$ | $72.9$ | $73.6$ | $\mathbf{82.6}$ | $\underline{81.2}$ |

Table 3: Average normalized score over the final evaluation and ten unseen training seeds on AntMaze tasks. CQL scores were taken from Ghasemipour et al. (2022). The symbol ± represents the standard deviation across the seeds. To make a fair comparison against TD3+BC and IQL, we extensively tuned their hyperparameters.

| Task Name | TD3+BC | IQL | CQL | SAC-RND | ReBRAC, our |
|---|---|---|---|---|---|
| antmaze-umaze | $66.3 \pm 6.2$ | $83.3 \pm 4.5$ | $74.0$ | $\underline{97.0} \pm 1.5$ | $\mathbf{97.8} \pm 1.0$ |
| antmaze-umaze-diverse | $53.8 \pm 8.5$ | $70.6 \pm 3.7$ | $\underline{84.0}$ | $66.0 \pm 25.0$ | $\mathbf{88.3} \pm 13.0$ |
| antmaze-medium-play | $26.5 \pm 18.4$ | $\underline{64.6} \pm 4.9$ | $61.2$ | $38.5 \pm 29.4$ | $\mathbf{84.0} \pm 4.2$ |
| antmaze-medium-diverse | $25.9 \pm 15.3$ | $61.7 \pm 6.1$ | $53.7$ | $\underline{74.7} \pm 10.7$ | $\mathbf{76.3} \pm 13.5$ |
| antmaze-large-play | $0.0 \pm 0.0$ | $42.5 \pm 6.5$ | $15.8$ | $\underline{43.9} \pm 29.2$ | $\mathbf{60.4} \pm 26.1$ |
| antmaze-large-diverse | $0.0 \pm 0.0$ | $27.6 \pm 7.8$ | $14.9$ | $\underline{45.7} \pm 28.5$ | $\mathbf{54.4} \pm 25.1$ |
| Average | $28.7$ | $58.3$ | $50.6$ | $\underline{60.9}$ | $\mathbf{76.8}$ |

domain. In other cases, we report results from previous works, meaning that scores for other methods can be lower if evaluated under our protocol.

The results of our tests on D4RL's Gym-MuJoCo, AntMaze, and Adroit tasks are available in Table 2, Table 3, Table 4, respectively. The mean-wise best results among algorithms are highlighted with **bold**, and the second best performance is underlined. Our approach, ReBRAC, achieves state-of-the-art results on Gym-MuJoCo, AntMaze, and Adroit tasks outperforming all baselines on average, except SAC-RND on Gym-MuJoCo tasks, which is slightly better. Performance profiles and probability of improvement (Agarwal et al., 2021) in Figure 1b and Figure 1c also demonstrate that ReBRAC is competitive when compared to the algorithms that we contrast against. Our method is also comparable to ensemble-based approaches (see Appendix C for additional comparisons).

## 4.2 Evaluation on offline V-D4RL

In addition to testing ReBRAC on D4RL, we evaluated its performance on V-D4RL benchmark (Lu et al., 2022). Our motivation for doing so was the fact that scores on D4RL Gym-MuJoCo tasks have saturated in recent years, and even after the introduction of ensemble-based offline RL methods

Table 4: Average normalized score over the final evaluation and ten unseen training seeds on Adroit tasks. BC and CQL scores were taken from Yang et al. (2022). The symbol ± represents the standard deviation across the seeds. To make a fair comparison against TD3+BC and IQL, we extensively tuned their hyperparameters.

| Task Name | BC | TD3+BC | IQL | CQL | SAC-RND | ReBRAC, our |
|---|---|---|---|---|---|---|
| pen-human | 34.4 | _81.8_ ± 14.9 | 81.5 ± 17.5 | 37.5 | 5.6 ± 5.8 | **103.5** ± 14.1 |
| pen-cloned | 56.9 | 61.4 ± 19.3 | _77.2_ ± 17.7 | 39.2 | 2.5 ± 6.1 | **91.8** ± 21.7 |
| pen-expert | 85.1 | _146.0_ ± 7.3 | 133.6 ± 16.0 | 107.0 | 45.4 ± 22.9 | **154.1** ± 5.4 |
| door-human | 0.5 | -0.1 ± 0.0 | _3.1_ ± 2.0 | **9.9** | 0.0 ± 0.1 | 0.0 ± 0.0 |
| door-cloned | -0.1 | 0.1 ± 0.6 | _0.8_ ± 1.0 | 0.4 | 0.2 ± 0.8 | **1.1** ± 2.6 |
| door-expert | 34.9 | 84.6 ± 44.5 | **105.3** ± 2.8 | 101.5 | 73.6 ± 26.7 | _104.6_ ± 2.4 |
| hammer-human | 1.5 | 0.4 ± 0.4 | _2.5_ ± 1.9 | **4.4** | -0.1 ± 0.1 | 0.2 ± 0.2 |
| hammer-cloned | 0.8 | 0.8 ± 0.7 | 1.1 ± 0.5 | _2.1_ | 0.1 ± 0.4 | **6.7** ± 3.7 |
| hammer-expert | 125.6 | 117.0 ± 30.9 | _129.6_ ± 0.5 | 86.7 | 24.8 ± 39.4 | **133.8** ± 0.7 |
| relocate-human | 0.0 | -0.2 ± 0.0 | _0.1_ ± 0.1 | **0.2** | 0.0 ± 0.0 | 0.0 ± 0.0 |
| relocate-cloned | -0.1 | -0.1 ± 0.1 | _0.2_ ± 0.4 | -0.1 | 0.0 ± 0.0 | **0.9** ± 1.6 |
| relocate-expert | 101.3 | **107.3** ± 1.6 | 106.5 ± 2.5 | 95.0 | 3.4 ± 4.5 | _106.6_ ± 3.2 |
| Average w/o expert | 11.7 | 18.0 | _20.8_ | 11.7 | 1.0 | **25.5** |
| Average | 36.7 | 49.9 | _53.4_ | 40.3 | 12.9 | **58.6** |

Table 5: Average normalized score over the final evaluation and five unseen training seeds on V-D4RL tasks. The score is mapped from a range of [0, 1000] to [0, 100]. The symbol ± represents the standard deviation across the seeds.

| Environment | | Offline DV2 | DrQ+BC | CQL | BC | LOMPO | ReBRAC, our |
|---|---|---|---|---|---|---|---|
| walker-walk | random | **28.7** ±13.0 | 5.5 ±0.9 | 14.4 ±12.4 | 2.0 ±0.2 | _21.9_ ±8.1 | 15.9 ± 2.3 |
| | mixed | **56.5** ±18.1 | 28.7 ±6.9 | 11.4 ±12.4 | 16.5 ±4.3 | 34.7 ±19.7 | _41.6_ ± 8.0 |
| | medium | 34.1 ±19.7 | _46.8_ ±2.3 | 14.8 ±16.1 | 40.9 ±3.1 | 43.4 ±11.1 | **52.5** ± 3.2 |
| | medexp | 43.9 ±34.4 | _86.4_ ±5.6 | 56.4 ±38.4 | 47.7 ±3.9 | 39.2 ±19.5 | **92.7** ± 1.3 |
| | expert | 4.8 ±0.6 | 68.4 ±7.5 | _89.6_ ±6.0 | **91.5** ±3.9 | 5.3 ±7.7 | 81.4 ± 10.0 |
| cheetah-run | random | **31.7** ±2.7 | 5.8 ±0.6 | 5.9 ±8.4 | 0.0 ±0.0 | 11.4 ±5.1 | _12.9_ ± 2.2 |
| | mixed | **61.6** ±1.0 | 44.8 ±3.6 | 10.7 ±12.8 | 25.0 ±3.6 | 36.3 ±13.6 | _46.8_ ± 0.7 |
| | medium | 17.2 ±3.5 | _53.0_ ±3.0 | 40.9 ±5.1 | 51.6 ±1.4 | 16.4 ±8.3 | **59.0** ± 0.7 |
| | medexp | 10.4 ±3.5 | 50.6 ±8.2 | 20.9 ±5.5 | _57.5_ ±6.3 | 11.9 ±1.9 | **58.3** ± 11.7 |
| | expert | 10.9 ±3.2 | 34.5 ±8.3 | _61.5_ ±4.3 | **67.4** ±6.8 | 14.0 ±3.8 | 35.6 ± 5.3 |
| humanoid-walk | random | 0.1 ±0.0 | 0.1 ±0.0 | 0.2 ±0.1 | 0.1 ±0.0 | 0.1 ±0.0 | 0.1 ± 0.0 |
| | mixed | 0.2 ±0.1 | 15.9 ±3.8 | 0.1 ±0.0 | **18.8** ±4.2 | 0.2 ±0.0 | _16.0_ ± 2.7 |
| | medium | 0.2 ±0.1 | 6.2 ±2.4 | 0.1 ±0.0 | **13.5** ±4.1 | 0.1 ±0.0 | _9.0_ ± 2.3 |
| | medexp | 0.1 ±0.0 | 7.0 ±2.3 | 0.1 ±0.0 | **17.2** ±4.7 | 0.2 ±0.0 | _7.8_ ± 2.4 |
| | expert | 0.2 ±0.1 | 2.7 ±0.9 | 1.6 ±0.5 | **6.1** ±3.7 | 0.1 ±0.0 | _2.9_ ± 0.9 |
| Average | | 20.0 | _30.4_ | 21.9 | 30.3 | 15.6 | **35.5** |

by An et al. (2021), there has been no notable progress on these tasks. On the other hand, V-D4RL provides a similar set of problems, with datasets collected in the same way as in D4RL but with the agent's observations now being images from the environment.

We tested our algorithm on all available single-task datasets without distractors and compared it to the baselines from the original V-D4RL work (Lu et al., 2022). The results are reported in Table 5. Our proposed approach achieves state-of-the-art or close-to-state-of-the-art results on most of the tasks, and it is the only method that, on average, performs notably better than naive Behavioral Cloning.

## 4.3 Evaluation on offline-to-online D4RL

The evaluation of offline-to-online performance is a pivotal aspect for reinforcement learning (RL) algorithms, particularly in light of recent developments. In this context, we conducted additional tests on ReBRAC, as it stands out as a promising algorithm for several compelling reasons.

First and foremost, ReBRAC demonstrates a remarkable proficiency following offline pre-training. Secondly, our algorithm shares notable similarities with TD3+BC, a method that has exhibited effectiveness in online fine-tuning as observed by Beeson et al. (Beeson & Montana, 2022)

For the sake of simplicity, we opted to disable critic penalization during the online fine-tuning. Furthermore, we linearly decay the actor's penalty to half of its initial value, following the approach described by Beeson & Montana (2022). Notably, no hyperparameter tuning was performed in this process.

To evaluate our approach in the offline-to-online setting, we followed the methodology outlined by Tarasov et al. (2022). In our comparative analysis, we consider the following algorithms: TD3+BC (Fujimoto & Gu, 2021), IQL (Kostrikov et al., 2021), CQL (Kumar et al., 2020), SPOT (Wu et al., 2022), and Cal-CQL (Nakamoto et al., 2023). The scores after the offline stage and online tuning, are reported in Table 6. We also provide finetuning cumulative regret proposed by Nakamoto et al. (2023) in Table 7.

Table 6: Normalized performance after offline pretraining and online finetuning on D4RL. Baselines scores except TD3+BC are taken from Tarasov et al. (2022). ReBRAC and TD3+BC scores are averaged over ten random seeds, and all others are averaged over four as in Tarasov et al. (2022).

| Task Name | TD3 + BC | IQL | SPOT | Cal-QL | ReBRAC, our |
|---|---|---|---|---|---|
| antmaze-u-v2 | 66.8 → 91.4 | 77.00 → 96.50 | 91.00 → 99.50 | 76.75 → **99.75** | 97.8 → **99.8** |
| antmaze-u-d-v2 | 59.1 → 48.4 | 59.50 → 63.75 | 36.25 → 95.00 | 32.00 → **98.50** | 85.7 → 98.1 |
| antmaze-m-p-v2 | 59.2 → 94.8 | 71.75 → 89.75 | 67.25 → 97.25 | 71.75 → **98.75** | 78.4 → 97.7 |
| antmaze-m-d-v2 | 62.6 → 94.1 | 64.25 → 92.25 | 73.75 → 94.50 | 62.00 → 98.25 | 78.6 → **98.5** |
| antmaze-l-p-v2 | 21.5 → 0.1 | 38.50 → 64.50 | 31.50 → 87.00 | 31.75 → **97.25** | 47.0 → 39.5 |
| antmaze-l-d-v2 | 9.5 → 0.4 | 26.75 → 64.25 | 17.50 → 81.00 | 44.00 → **91.50** | 66.7 → 77.6 |
| **AntMaze avg** | 46.4 → 54.8 (+8.4) | 56.29 → 78.50 (+22.21) | 52.88 → 92.38 (+39.50) | 53.04 → 97.33 (+24.29) | 75.7 → 85.2 (+9.5) |
| pen-c-v1 | 86.1 → 110.3 | 84.19 → 102.02 | 6.19 → 43.63 | -2.66 → -2.68 | 91.8 → **152.0** |
| door-c-v1 | 0.0 → 3.4 | 1.19 → 20.34 | -0.21 → 0.02 | -0.33 → -0.33 | 0.4 → **104.9** |
| hammer-c-v1 | 2.4 → 11.6 | 1.35 → 57.27 | 3.97 → 3.73 | 0.25 → 0.17 | 4.1 → **131.2** |
| relocate-c-v1 | -0.1 → 0.1 | 0.04 → 0.32 | -0.24 → -0.15 | -0.31 → -0.31 | 0.0 → **12.3** |
| **Adroit Avg** | 22.1 → 31.3 (+9.2) | 21.69 → 44.99 (+23.3) | 2.43 → 11.81 (+9.38) | -0.76 → -0.79 (-0.03) | 24.0 → **100.1** (+76.1) |
| **Total avg** | 36.7 → 45.4 (+8.7) | 42.45 → 65.10 (+22.65) | 32.70 → 60.15 (+27.45) | 31.52 → 58.08 (+26.56) | 55.0 → **91.1** (+36.1) |

Table 7: Cumulative regret of online finetuning calculated as $1 -$ *average success rate*. Baselines scores except TD3+BC are taken from Tarasov et al. (2022). ReBRAC and TD3+BC regrets are averaged over ten random seeds, and all others are averaged over four as in Tarasov et al. (2022).

| Task Name | TD3 + BC | CQL | IQL | SPOT | Cal-QL | ReBRAC, our |
|---|---|---|---|---|---|---|
| antmaze-umaze-v2 | 0.09 ± 0.08 | 0.02 ± 0.00 | 0.07 ± 0.00 | 0.02 ± 0.00 | 0.01 ± 0.00 | **0.00** ± 0.00 |
| antmaze-umaze-diverse-v2 | 0.47 ± 0.16 | 0.09 ± 0.01 | 0.43 ± 0.11 | 0.22 ± 0.07 | **0.05** ± 0.01 | 0.06 ± 0.13 |
| antmaze-medium-play-v2 | 0.12 ± 0.05 | 0.08 ± 0.01 | 0.09 ± 0.01 | 0.06 ± 0.00 | 0.04 ± 0.01 | **0.03** ± 0.01 |
| antmaze-medium-diverse-v2 | 0.09 ± 0.02 | 0.08 ± 0.00 | 0.10 ± 0.01 | 0.05 ± 0.01 | 0.04 ± 0.01 | **0.02** ± 0.00 |
| antmaze-large-play-v2 | 0.99 ± 0.00 | 0.21 ± 0.02 | 0.34 ± 0.05 | 0.29 ± 0.07 | **0.13** ± 0.02 | 0.36 ± 0.30 |
| antmaze-large-diverse-v2 | 0.99 ± 0.01 | 0.21 ± 0.03 | 0.41 ± 0.03 | 0.23 ± 0.08 | 0.13 ± 0.02 | **0.10** ± 0.07 |
| **AntMaze avg** | 0.45 | 0.11 | 0.24 | 0.15 | **0.07** | 0.09 |
| pen-cloned-v1 | 0.30 ± 0.10 | 0.97 ± 0.00 | 0.37 ± 0.01 | 0.58 ± 0.02 | 0.98 ± 0.01 | **0.08** ± 0.00 |
| door-cloned-v1 | 0.97 ± 0.03 | 1.00 ± 0.00 | 0.83 ± 0.03 | 0.99 ± 0.01 | 1.00 ± 0.00 | **0.26** ± 0.10 |
| hammer-cloned-v1 | 0.92 ± 0.14 | 1.00 ± 0.00 | 0.65 ± 0.10 | 0.98 ± 0.01 | 1.00 ± 0.00 | **0.13** ± 0.02 |
| relocate-cloned-v1 | 0.99 ± 0.01 | 1.00 ± 0.00 | 1.00 ± 0.00 | 1.00 ± 0.00 | 1.00 ± 0.00 | **0.85** ± 0.07 |
| **Adroit avg** | 0.79 | 0.99 | 0.71 | 0.89 | 0.99 | **0.33** |
| **Total avg** | 0.59 | 0.47 | 0.43 | 0.44 | 0.44 | **0.18** |

ReBRAC exhibits competitive performance, surpassing four out of six AntMaze datasets and achieving state-of-the-art results in terms of final scores on Adroit tasks. On average, ReBRAC outperforms its closest competitor, Cal-CQL, which was specifically designed for the offline-to-online problem in the concurrent work (Nakamoto et al., 2023).

Regarding regret, ReBRAC outperforms all other algorithms, with Cal-QL being the sole exception, showing slightly better results on average only in the AntMaze domain.

## 4.4 Ablating Design Choices

To better understand the source of improved performance, we conducted an ablation study on the modifications made to the algorithm. Results can be found in Table 8. Additional ablation studies

for all datasets can be found in Appendix G. One modification at a time was disabled, while all other changes were retained, including layer normalization in the critic network, additional linear layers in the actor and critic networks, adding an MSE penalty to the critic and actor loss. In the case of AntMaze, we also attempted to use the default $\gamma$ value instead of the increased one. To further demonstrate the efficacy of our modifications, we also ran our implementation as equivalent to the original TD3+BC, with all changes disabled and hyperparameters were taken from the original paper. This experiment serves to show that the improved scores are due to the proposed changes in the algorithm and not just different implementations. Furthermore, we searched for the regularization parameter for our implementation of TD3+BC to demonstrate that tuning this parameter is not the sole source of improvement. Moreover, we tested the TD3 + BC by adding each of the ReBRAC's modifications independently to demonstrate that each individual modifications is not sufficient for achieving performance of ReBRAC where the combination of modification appear.

Additionally, we run ablation which to validate the importance of decoupling by disabling it and searching the best penalty parameter value from the set of all previously used values listed in the Appendix B.

Table 8: ReBRAC's design choices ablations: each modification was disabled while keeping all the others. For brevity, we report the mean of average normalized scores over four unseen training seeds across domains. We also include tuned results for TD3+BC to highlight that the improvement does not come from hyperparameter search. For dataset-specific results, please refer to Appendix G.

| Ablation | Gym-MuJoCo | AntMaze | Adroit | All |
|---|---|---|---|---|
| TD3+BC, paper | - | 27.3 | 0.0 | - |
| TD3+BC, our | 63.4 | 18.5 | 52.3 | 52.2 |
| TD3+BC, tuned | 71.8 (-10.9%) | 27.9 (-62.9%) | 53.5 (-25.9%) | 58.3 (-19.2%) |
| TD3+BC w/ $\gamma$ change | - | 17.5 (-76.7%) | - | - |
| TD3+BC w/ LN | 71.4 (-11.4%) | 35.6 (-52.7%) | 55.6 (-4.1%) | 60.2 (-16.6%) |
| TD3+BC w/ large batch | 14.4 (-82.1%) | 0.0 (-100.0%) | 1.6 (-97.2%) | 7.9 (-89.0%) |
| TD3+BC w/ layer | 71.2 (-11.6%) | 44.1 (-41.4%) | 56.4 (-2.7%) | 61.9 (-14.2%) |
| ReBRAC w/o large batch | 75.9 (-5.8%) | - | - | - |
| ReBRAC w large batch | - | 41.0 (-45.6%) | 55.4 (-4.6%) | - |
| ReBRAC w/o $\gamma$ change | - | 21.0 (-72.1%) | - | - |
| ReBRAC w/o LN | 59.2 (-26.5%) | 0.0 (-100.0%) | 25.1 (-56.7%) | 38.0 (-47.3%) |
| ReBRAC w/o layer | 78.5 (-2.6%) | 18.1 (-75.9%) | 59.0 (+1.7%) | 61.9 (-14.2%) |
| ReBRAC w/o actor penalty | 22.8 (-71.7%) | 0.1 (-99.8%) | 0.0 (-100.0%) | 11.4 (-84.2%) |
| ReBRAC w/o critic penalty | 81.1 (+0.6%) | 72.2 (-4.1%) | 56.9 (-1.8%) | 71.5 (-0.9%) |
| ReBRAC w/o decoupling | 79.8 (-0.9%) | 76.9 (+2.1%) | 56.7 (-2.2%) | 71.6 (-0.8%) |
| ReBRAC | 80.6 | 75.3 | 58.0 | 72.2 |

The ablation results show that ReBRAC outperforms TD3+BC not because of different implementations or actor regularization parameter choice. All domains suffer when the LayerNorm is disabled, leading to the halved average performance overall. Removing additional layers leads to a notable drop in AntMaze tasks, while on Gym-MuJoCo decrease is small, and on Adroit tasks, we can see a slight boost. The algorithm fails to learn most tasks when the actor penalty is disabled. Notably, the critic penalty plays a minor role in improving the performance on most of the problems as well as the decoupling penalties. Using standard batch size on Gym-MuJoCo tasks significantly decreases final scores, while using the increased discount factor for AntMaze is crucial for obtaining state-of-the-art performance.

Based on the conducted ablations studies, the proposed configuration of design choices leads to the best performance on average. Note that tuning these choices for each task independently makes it possible to get even higher scores than we report in Section 4.1. We also can change the number of hidden layers in the networks, leading to better performance, see Section 4.5. But in real life, algorithm evaluation might be costly, so we limit ourselves to tuning only regularization parameters.

## 4.5 Stacking Even More Layers

As ablations show, the depth of the network plays an important role when solving AntMaze and HalfCheetah tasks (see Appendix G). We conduct additional experiments to check how the perfor-

mance depends on the network depth in more detail. Ball et al. (2023) used networks of depth four when solving AntMaze tasks. Our goal is to find the point where the performance saturates. For this, we run our algorithm on AntMaze tasks increasing the number of layers to six while keeping other parameters unchanged. We attempt to increase actor and critic separately and at once. We also decreased each network's size by one layer similarly. Results can be found in Figure 2.

Several conclusions can be drawn from the results. First, further increments in the number of layers can lead to better results on the AntMaze domain when layers are scaled up to five. For six layers, performance drops or does not improve. Second, decreasing the critic's size leads to the worst performance on most datasets. Lastly, there is no clear pattern on how the performance changes even within a single domain. The drop on six layers is the only common feature that can be seen. On average, four critic layers and three actor layers were the best. Changing only the actor's network is more stable on average.

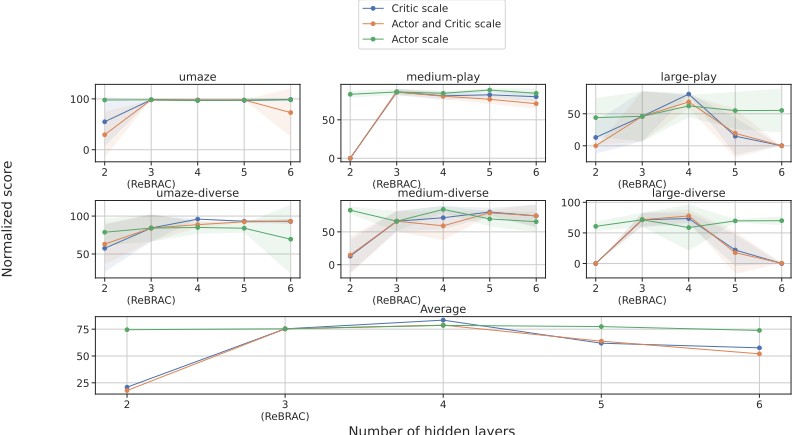

Figure 2: Impact of networks' depth on the final performance for the AntMaze tasks. Scores are averaged over four unseen training seeds. Shaded areas represent one standard deviation across seeds. These graphics demonstrate that one can achieve marginally better scores by tuning the number of layers for certain tasks.

## 4.6 Penalization Sensitivity Analysis

Following Kurenkov & Kolesnikov (2022), we demonstrate the sensitivity of ReBRAC to the choice of $\beta_1$ and $\beta_2$ hyperparameters under uniform policy selection on D4RL tasks using Expected Online Performance (EOP) and comparing it to the TD3+BC and IQL. EOP shows the best performance expected depending on the number of policies that can be deployed for online evaluation. Results are demonstrated in Table 9. As one can see, approximately ten policies are required for ReBRAC to attain ensemble-free state-of-the-art performance. ReBRAC's EOP is higher for any number of online policies when compared to TD3+BC and better than IQL on all domains when the evaluation budget is larger than two policies. See Appendix F for EOP separated by tasks.

Table 9: Expected Online Performance (Kurenkov & Kolesnikov, 2022) under uniform policy selection aggregated over D4RL domains across four training seeds. This demonstrates the sensitivity to the choice of hyperparameters given a certain budget for online evaluation. For dataset-specific results, please see Appendix F.

| Domain | Algorithm | 1 policy | 2 policies | 3 policies | 5 policies | 10 policies | 15 policies | 20 policies |
|---|---|---|---|---|---|---|---|---|
| Gym-MuJoCo | TD3+BC | $49.8 \pm 21.4$ | $61.0 \pm 14.5$ | $65.3 \pm 9.3$ | $67.8 \pm 3.9$ | - | - | - |
| | IQL | $\mathbf{65.0} \pm 9.1$ | $69.9 \pm 5.6$ | $71.7 \pm 3.5$ | $72.9 \pm 1.7$ | $73.6 \pm 0.8$ | $73.8 \pm 0.7$ | $74.0 \pm 0.6$ |
| | ReBRAC | $62.0 \pm 17.1$ | $\mathbf{70.6} \pm 9.9$ | $\mathbf{73.3} \pm 5.5$ | $\mathbf{74.8} \pm 2.1$ | $\mathbf{75.6} \pm 0.8$ | $\mathbf{75.8} \pm 0.6$ | $\mathbf{76.0} \pm 0.5$ |
| AntMaze | TD3+BC | $6.9 \pm 7.0$ | $10.7 \pm 6.8$ | $13.0 \pm 6.0$ | $15.5 \pm 4.6$ | - | - | - |
| | IQL | $29.8 \pm 15.5$ | $38.0 \pm 15.4$ | $43.1 \pm 13.8$ | $48.7 \pm 10.2$ | $53.2 \pm 4.4$ | $54.3 \pm 2.1$ | $54.7 \pm 1.2$ |
| | ReBRAC | $\mathbf{67.9} \pm 10.0$ | $\mathbf{73.6} \pm 7.4$ | $\mathbf{76.1} \pm 5.5$ | $\mathbf{78.3} \pm 3.4$ | $\mathbf{79.9} \pm 1.7$ | $\mathbf{80.4} \pm 1.1$ | - |
| Adroit | TD3+BC | $23.6 \pm 19.9$ | $34.6 \pm 17.7$ | $40.6 \pm 14.5$ | $46.4 \pm 9.8$ | - | - | - |
| | IQL | $\mathbf{53.1} \pm 0.7$ | $\mathbf{53.5} \pm 0.6$ | $53.7 \pm 0.5$ | $53.9 \pm 0.3$ | $54.1 \pm 0.2$ | $54.2 \pm 0.2$ | $54.2 \pm 0.1$ |
| | ReBRAC | $44.1 \pm 18.4$ | $53.2 \pm 10.9$ | $\mathbf{56.1} \pm 6.1$ | $\mathbf{57.8} \pm 2.3$ | $\mathbf{58.6} \pm 0.9$ | $\mathbf{58.9} \pm 0.7$ | $\mathbf{59.1} \pm 0.6$ |

## 5 Related Work

**Ensemble-free offline RL methods.** In recent years, many offline reinforcement learning algorithms were developed. TD3+BC (Fujimoto & Gu, 2021) represents a minimalist approach to offline RL, which incorporates a Behavioral Cloning component into the actor loss, enabling online actor-critic algorithms to operate in an offline setting. CQL (Kumar et al., 2020) drives the critic network to assign lower values to out-of-distribution state-action pairs and higher values to in-distribution pairs. IQL (Kostrikov et al., 2021) proposes a method for learning a policy without sampling out-of-distribution actions.

Despite this, more sophisticated methods may be necessary to achieve state-of-the-art results in an ensemble-free setup. For instance, Chen et al. (2022b); Akimov et al. (2022) pre-train different forms of encoders for actions, then optimize the actor to predict actions in the latent space. SPOT (Wu et al., 2022) pre-trains Variational Autoencoder and uses its uncertainty to penalize actor for sampling OOD actions while SAC-RND (Nikulin et al., 2023) applies Random Network Distillation and penalizes actor and critic.

**Ensemble-based offline RL methods.** A significant number of works in offline reinforcement learning have also leveraged ensemble methods for uncertainty estimation. The recently introduced SAC-N (An et al., 2021) algorithm outperformed all previous approaches on D4RL Gym-MuJoCo tasks; however, it necessitated large ensembles for some tasks, such as the hopper task, which required an ensemble size of 500 and imposed a significant computational burden. To mitigate this, the EDAC algorithm was introduced in the same work, which utilized ensemble diversification to reduce the ensemble size from 500 to 50. Despite the reduction, the ensemble size remains substantial compared to ensemble-free alternatives. It is worth mentioning that neither SAC-N nor EDAC is capable of solving the complex AntMaze tasks (Tarasov et al., 2022).

Another state-of-the-art algorithm in the Gym-MuJoCo tasks is RORL (Yang et al., 2022), which is a modification of SAC-N that makes the Q function more robust and smooth by perturbing state-action pairs with the use of out-of-distribution actions. RORL also requires an ensemble size of up to 20. On the other hand, MSG (Ghasemipour et al., 2022) utilizes independent targets for each ensemble member and achieves good performance on the Gym-MuJoCo tasks with an ensemble size of four but requires 64 ensemble members to achieve state-of-the-art performance on the AntMaze tasks.

**Design choices ablations.** Ablations of different design choices on established baselines are very limited, especially in offline RL. It is only shown by Fujimoto & Gu (2021) that CQL performs poorly if proposed non-algorithmic differences are eliminated. A parallel study (Ball et al., 2023) shows that some of the considered modifications (LayerNorm and networks depth) are important when used in online RL with offline data setting, which is different from pure offline.

## 6 Conlusion, Limitations, and Future work

In this work, we revisit recent advancements in the offline RL field over the last two years and incorporate a modest set of improvements to a previously established minimalistic TD3+BC baseline. Our experiments demonstrate that despite these limited updates, we can achieve more than competitive results on offline and offline-to-online D4RL and offline V-D4RL benchmarks under different hyperparameter budgets.

Despite the noteworthy results, our work is limited to one approach and a subset of possible design changes. It is imperative to explore the potential impact of these modifications on other offline RL methods (e.g., IQL, CQL, MSG) and to investigate other design choices used in offline RL, e.g., learning rate schedules, dropout (like in IQL), wider networks, or selection between stochastic and deterministic policies.

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

# A    Experimental Details

In order to generate the results presented in Table 2 Table 3 and Table 4, we conducted a hyperparameter search and selected the best results from the final evaluations for each dataset. Our algorithm was implemented using JAX for the D4RL benchmark. For V-D4RL, we implement our approach using PyTorch adopting the TD3+BC implementation from Clean Offline RL (Tarasov et al., 2022). The experiments were conducted on V100 and A100 GPUs.

**Gym-MuJoCo and Adroit tasks.**   Our study utilized the latest version of the datasets – v2 for Gym-MuJoCo and v1 for Adroit. The agents were trained for one million steps and evaluated over ten episodes.

For ReBRAC, we fine-tuned the $\beta_1$ parameter for the actor, which was selected from $0.001, 0.01, 0.05, 0.1$. Similarly, the $\beta_2$ parameter for the critic was selected from a range of $0, 0.001, 0.01, 0.1, 0.5$. The selected best parameters for each dataset are reported in Table 11.

For TD3+BC here and in the AntMaze domain, we use the same grid used in ReBRAC for actor regularization parameter $\alpha$ and add the default value of $0.4$.

For IQL here and in the AntMaze domain, we selected $\beta$ value from a range of $0.5, 1, 3, 6, 10$ and IQL $\tau$ value from a range of $0.5, 0.7, 0.9, 0.95$. We used the implementation from Clean Offline RL (Tarasov et al., 2022) and kept other parameters unchanged.

For SAC-RND in Adroit domain we tune $\beta_1$ (actor parameter) in the range of $0.5, 1.0, 2.5, 5.0, 10.0$ and $\beta_2$ (critic parameter) in the range of $0.01, 0.1, 1.0, 5.0, 10.0$.

**AntMaze tasks.**   In our work, we utilized v2 of the datasets. It's worth noting that previous studies have reported results using v0 datasets, which were found to contain numerous issues[2]. Each agent was trained for 1 million steps and evaluated over 100 episodes. Following Chen et al. (2022a), we modified the reward function by multiplying it by 100.

For ReBRAC, the $\beta_1$ (actor) and $\beta_2$ (critic) hyperparameters were carefully selected from the respective ranges of $0.0005, 0.001, 0.002, 0.003$ and $0, 0.0001, 0.0005, 0.001$. In addition, the actor and critic learning rates were optimized from $0.0001, 0.0002, 0.0003, 0.0005$ and $0.0003, 0.0005, 0.001$, respectively. The optimal hyperparameters for each dataset are presented in Table 11.

We also modified the $\gamma$ value for ReBRAC when addressing these tasks, driven by the following motivation. The length of the episodes in AntMaze can be as long as 1000 steps, while the reward is sparse and can only be obtained at the end of the episode. As a result, the discount for the reward with the default $\gamma$ can be as low as $0.99^{1000} = 4 \cdot 10^{-5}$, which is extremely low for signal propagation, even when multiplying the reward by 100. By increasing $\gamma$ to 0.999, the minimum discount value becomes $0.999^{1000} = 0.36$, which is more favorable for signal propagation.

**V-D4RL.**   We used single-task datasets without distraction with a resolution of $84 \times 84$ pixels. For ReBRAC $\beta_1$ (actor) parameter was selected from the range of $\{0.03, 0.1, 0.3, 1.0\}$ and $\beta_2$ (critic) parameter from the range of $\{0.0, 0.001, 0.005, 0.01, 0.1\}$.

**Offline-to-offline.**   We used the same parameters for the offline-to-online setup with the only difference of setting $\beta_2$ to zero and lineary decaying $\beta_1$ to half of it's initial value during the online stage.

---

[2]https://github.com/Farama-Foundation/D4RL/issues/77

# B Hyperparameters

## B.1 ReBRAC

Table 10: ReBRAC's general hyperparameters.

| Parameter | Value |
|---|---|
| optimizer | Adam Kingma & Ba (2014) |
| batch size | 1024 on Gym-MuJoCo, 256 on other |
| learning rate (all networks) | 1e-3 on Gym-MuJoCo, 3e-4 on Adroit and V-D4RL, 1e-4 on Antmaze |
| tau ($\tau$) | 5e-3 |
| hidden dim (all networks) | 256 |
| num hidden layers (all networks) | 3 |
| gamma ($\gamma$) | 0.999 on AntMaze, 0.99 on other |
| nonlinearity | ReLU |

Table 11: ReBRAC's best hyperparameters used in D4RL benchmark.

| Task Name | $\beta_1$ (actor) | $\beta_2$ (critic) |
|---|---|---|
| halfcheetah-random | 0.001 | 0.1 |
| halfcheetah-medium | 0.001 | 0.01 |
| halfcheetah-expert | 0.01 | 0.01 |
| halfcheetah-medium-expert | 0.01 | 0.1 |
| halfcheetah-medium-replay | 0.01 | 0.001 |
| halfcheetah-full-replay | 0.001 | 0.1 |
| hopper-random | 0.001 | 0.01 |
| hopper-medium | 0.01 | 0.001 |
| hopper-expert | 0.1 | 0.001 |
| hopper-medium-expert | 0.1 | 0.01 |
| hopper-medium-replay | 0.05 | 0.5 |
| hopper-full-replay | 0.01 | 0.01 |
| walker2d-random | 0.01 | 0.0 |
| walker2d-medium | 0.05 | 0.1 |
| walker2d-expert | 0.01 | 0.5 |
| walker2d-medium-expert | 0.01 | 0.01 |
| walker2d-medium-replay | 0.05 | 0.01 |
| walker2d-full-replay | 0.01 | 0.01 |
| antmaze-umaze | 0.003 | 0.002 |
| antmaze-umaze-diverse | 0.003 | 0.001 |
| antmaze-medium-play | 0.001 | 0.0005 |
| antmaze-medium-diverse | 0.001 | 0.0 |
| antmaze-large-play | 0.002 | 0.001 |
| antmaze-large-diverse | 0.002 | 0.002 |
| pen-human | 0.1 | 0.5 |
| pen-cloned | 0.05 | 0.5 |
| pen-expert | 0.01 | 0.01 |
| door-human | 0.1 | 0.1 |
| door-cloned | 0.01 | 0.1 |
| door-expert | 0.05 | 0.01 |
| hammer-human | 0.01 | 0.5 |
| hammer-cloned | 0.1 | 0.5 |
| hammer-expert | 0.01 | 0.01 |
| relocate-human | 0.1 | 0.01 |
| relocate-cloned | 0.1 | 0.01 |
| relocate-expert | 0.05 | 0.01 |

Table 12: ReBRAC's best hyperparameters used in V-D4RL benchmark.

| Task Name | $\beta_1$ (actor) | $\beta_2$ (critic) |
|---|---|---|
| walker-walk-random | 0.03 | 0.1 |
| walker-walk-medium | 0.03 | 0.005 |
| walker-walk-expert | 0.1 | 0.01 |
| walker-walk-medium-expert | 0.3 | 0.005 |
| walker-walk-medium-replay | 0.3 | 0.01 |
| cheetah-run-random | 0.1 | 0.01 |
| cheetah-run-medium | 0.1 | 0.1 |
| cheetah-run-expert | 0.01 | 0.01 |
| cheetah-run-medium-expert | 1.0 | 0.001 |
| cheetah-run-medium-replay | 0.03 | 0.1 |
| humanoid-walk-random | 1.0 | 0.01 |
| humanoid-walk-medium | 1.0 | 0.005 |
| humanoid-walk-expert | 1.0 | 0.1 |
| humanoid-walk-medium-expert | 1.0 | 0.005 |
| humanoid-walk-medium-replay | 1.0 | 0.001 |

## B.2 IQL

Table 13: IQL's best hyperparameters used in D4RL benchmark.

| Task Name | $\beta$ | IQL $\tau$ |
|---|---|---|
| halfcheetah-random | 3.0 | 0.95 |
| halfcheetah-medium | 3.0 | 0.95 |
| halfcheetah-expert | 6.0 | 0.9 |
| halfcheetah-medium-expert | 3.0 | 0.7 |
| halfcheetah-medium-replay | 3.0 | 0.95 |
| halfcheetah-full-replay | 1.0 | 0.7 |
| hopper-random | 1.0 | 0.95 |
| hopper-medium | 3.0 | 0.7 |
| hopper-expert | 3.0 | 0.5 |
| hopper-medium-expert | 6.0 | 0.7 |
| hopper-medium-replay | 6.0 | 0.7 |
| hopper-full-replay | 10.0 | 0.9 |
| walker2d-random | 0.5 | 0.9 |
| walker2d-medium | 6.0 | 0.5 |
| walker2d-expert | 6.0 | 0.9 |
| walker2d-medium-expert | 1.0 | 0.5 |
| walker2d-medium-replay | 0.5 | 0.7 |
| walker2d-full-replay | 1.0 | 0.7 |
| antmaze-umaze | 10.0 | 0.7 |
| antmaze-umaze-diverse | 10.0 | 0.95 |
| antmaze-medium-play | 6.0 | 0.9 |
| antmaze-medium-diverse | 6.0 | 0.9 |
| antmaze-large-play | 10.0 | 0.9 |
| antmaze-large-diverse | 6.0 | 0.9 |
| pen-human | 1.0 | 0.95 |
| pen-cloned | 10.0 | 0.9 |
| pen-expert | 10.0 | 0.8 |
| door-human | 0.5 | 0.9 |
| door-cloned | 6.0 | 0.7 |
| door-expert | 0.5 | 0.7 |
| hammer-human | 3.0 | 0.9 |
| hammer-cloned | 6.0 | 0.7 |
| hammer-expert | 0.5 | 0.95 |
| relocate-human | 1.0 | 0.95 |
| relocate-cloned | 6.0 | 0.9 |
| relocate-expert | 10.0 | 0.9 |

## B.3   TD3+BC

Table 14: TD3+BC's best hyperparameters used in D4RL benchmark.

| Task Name | $\alpha$ |
|---|---|
| halfcheetah-random | 0.001 |
| halfcheetah-medium | 0.01 |
| halfcheetah-expert | 0.4 |
| halfcheetah-medium-expert | 0.1 |
| halfcheetah-medium-replay | 0.05 |
| halfcheetah-full-replay | 0.01 |
| hopper-random | 0.4 |
| hopper-medium | 0.05 |
| hopper-expert | 0.1 |
| hopper-medium-expert | 0.1 |
| hopper-medium-replay | 0.4 |
| hopper-full-replay | 0.01 |
| walker2d-random | 0.001 |
| walker2d-medium | 0.4 |
| walker2d-expert | 0.05 |
| walker2d-medium-expert | 0.1 |
| walker2d-medium-replay | 0.1 |
| walker2d-full-replay | 0.1 |
| antmaze-umaze | 0.4 |
| antmaze-umaze-diverse | 0.4 |
| antmaze-medium-play | 0.003 |
| antmaze-medium-diverse | 0.003 |
| antmaze-large-play | 0.003 |
| antmaze-large-diverse | 0.003 |
| pen-human | 0.1 |
| pen-cloned | 0.4 |
| pen-expert | 0.4 |
| door-human | 0.1 |
| door-cloned | 0.4 |
| door-expert | 0.1 |
| hammer-human | 0.4 |
| hammer-cloned | 0.4 |
| hammer-expert | 0.4 |
| relocate-human | 0.1 |
| relocate-cloned | 0.1 |
| relocate-expert | 0.4 |

Table 15: SAC-RND's best hyperparameters used in D4RL Adroit tasks.

| Task Name | $\beta_1$ (actor) | $\beta_2$ (critic) |
|---|---|---|
| pen-human | 1.0 | 10.0 |
| pen-cloned | 2.5 | 0.01 |
| pen-expert | 10.0 | 5.0 |
| door-human | 5.0 | 0.01 |
| door-cloned | 5.0 | 1.0 |
| door-expert | 10.0 | 10.0 |
| hammer-human | 10.0 | 0.01 |
| hammer-cloned | 1.0 | 1.0 |
| hammer-expert | 2.5 | 10.0 |
| relocate-human | 5.0 | 0.01 |
| relocate-cloned | 5.0 | 1.0 |
| relocate-expert | 10.0 | 10.0 |

# C Comparison to Ensemble-based Methods

Comparison of ReBRAC with the ensemble-based methods is presented in Table 16, Table 17, and Table 18. We add the following ensemble-based methods: RORL for each domain (Yang et al., 2022), SAC-N/EDAC (An et al., 2021) for the Gym-MuJoCo and Adroit tasks[3] and MSG (Ghasemipour et al., 2022) for AntMaze tasks[4]. The mean-wise best results among algorithms are highlighted with **bold**, and the second-best performance is underlined. Our approach, ReBRAC, shows competitive results on the Gym-MuJoCo datasets. On AntMaze tasks, ReBRAC achieves state-of-the-art results among ensemble-free algorithms and a good score compared to ensemble-based algorithms. And on Adroit tasks, our approach outperforms both families of algorithms.

Table 16: ReBRAC evaluation on the Gym domain. We report the final normalized score averaged over 10 unseen training seeds on v2 datasets. CQL, SAC-N and EDAC scores are taken from An et al. (2021). RORL scores are taken from Yang et al. (2022).

| | Ensemble-free | | | | Ensemble-based | | | |
|---|---|---|---|---|---|---|---|---|
| Task Name | TD3+BC | IQL | CQL | SAC-RND | SAC-N | EDAC | RORL | ReBRAC, our |
| halfcheetah-random | 30.9 ± 0.4 | 19.5 ± 0.8 | **31.1** ± 3.5 | 27.6 ± 2.1 | 28.0 ± 0.9 | 28.4 ± 1.0 | 28.5 ± 0.8 | 29.5 ± 1.5 |
| halfcheetah-medium | 54.7 ± 0.9 | 50.0 ± 0.2 | 46.9 ± 0.4 | 66.4 ± 1.4 | 67.5 ± 1.2 | 65.9 ± 0.6 | **66.8** ± 0.7 | 65.6 ± 1.0 |
| halfcheetah-expert | 93.4 ± 0.4 | 95.5 ± 2.1 | 97.3 ± 1.1 | 102.6 ± 4.2 | 105.2 ± 2.6 | **106.8** ± 3.4 | 105.2 ± 0.7 | **105.9** ± 1.7 |
| halfcheetah-medium-expert | 89.1 ± 5.6 | 92.7 ± 2.8 | 95.0 ± 1.4 | **108.1** ± 1.5 | 107.1 ± 2.0 | 106.3 ± 1.9 | 107.8 ± 1.1 | 101.1 ± 5.2 |
| halfcheetah-medium-replay | 45.0 ± 1.1 | 42.1 ± 3.6 | 45.3 ± 0.3 | 51.2 ± 3.2 | **63.9** ± 0.8 | 61.3 ± 1.9 | 61.9 ± 1.5 | 51.0 ± 0.8 |
| halfcheetah-full-replay | 75.0 ± 2.5 | 75.0 ± 0.7 | 76.9 ± 0.9 | 81.2 ± 1.3 | 84.5 ± 1.2 | **84.6** ± 0.9 | - | 82.1 ± 1.1 |
| hopper-random | 8.5 ± 0.6 | 10.1 ± 5.9 | 5.3 ± 0.6 | 19.6 ± 12.4 | 31.3 ± 0.0 | 25.3 ± 10.4 | **31.4** ± 0.1 | 8.1 ± 2.4 |
| hopper-medium | 60.9 ± 7.6 | 65.2 ± 4.2 | 61.9 ± 6.4 | 91.1 ± 10.1 | 100.3 ± 0.3 | 101.6 ± 0.6 | **104.8** ± 0.1 | 102.0 ± 1.0 |
| hopper-expert | 109.6 ± 3.7 | 108.8 ± 3.1 | 106.5 ± 9.1 | 109.8 ± 0.5 | 110.3 ± 0.3 | 110.1 ± 0.1 | **112.8** ± 0.2 | 100.1 ± 8.3 |
| hopper-medium-expert | 87.8 ± 10.5 | 85.5 ± 29.7 | 96.9 ± 15.1 | 109.8 ± 0.6 | 110.1 ± 0.3 | 110.7 ± 0.1 | **112.7** ± 0.2 | 107.0 ± 6.4 |
| hopper-medium-replay | 55.1 ± 31.7 | 89.6 ± 13.2 | 86.3 ± 7.3 | 97.2 ± 9.0 | 101.8 ± 0.5 | 101.0 ± 0.5 | **102.8** ± 0.5 | 98.1 ± 1.3 |
| hopper-full-replay | 97.9 ± 17.5 | 104.4 ± 10.8 | 101.9 ± 0.6 | **107.4** ± 0.8 | 102.9 ± 0.3 | 105.4 ± 0.7 | - | 107.1 ± 0.4 |
| walker2d-random | 2.0 ± 3.6 | 11.3 ± 7.0 | 5.1 ± 1.7 | 18.7 ± 6.9 | **21.7** ± 0.0 | 16.6 ± 7.0 | 21.4 ± 0.2 | 18.1 ± 4.5 |
| walker2d-medium | 77.7 ± 2.9 | 80.7 ± 3.4 | 79.5 ± 3.2 | 92.7 ± 1.2 | 87.9 ± 0.2 | 92.5 ± 0.8 | **102.4** ± 1.4 | 82.5 ± 3.6 |
| walker2d-expert | 110.0 ± 0.6 | 96.9 ± 32.3 | 109.3 ± 0.1 | 104.5 ± 22.8 | 107.4 ± 2.4 | 115.1 ± 1.9 | **115.4** ± 0.5 | 112.3 ± 0.2 |
| walker2d-medium-expert | 110.4 ± 0.6 | 112.1 ± 0.5 | 109.1 ± 0.2 | 104.6 ± 11.2 | 116.7 ± 0.4 | 114.7 ± 0.9 | **121.2** ± 1.5 | 111.6 ± 0.3 |
| walker2d-medium-replay | 68.0 ± 19.2 | 75.4 ± 9.3 | 76.8 ± 10.0 | 89.4 ± 3.8 | 78.7 ± 0.7 | 87.1 ± 2.4 | **90.4** ± 0.5 | 77.3 ± 7.9 |
| walker2d-full-replay | 90.3 ± 5.4 | 97.5 ± 1.4 | 94.2 ± 1.9 | **105.3** ± 3.2 | 94.6 ± 0.5 | 99.8 ± 0.7 | - | 102.2 ± 1.7 |
| Average w/o full-replay | 66.8 | 70.1 | 70.1 | 79.5 | 82.4 | **82.9** | **85.7** | 78.0 |
| Average | 70.3 | 72.9 | 73.6 | 82.6 | 84.4 | **85.2** | - | 81.2 |

[3]SAC-N and EDAC score 0 on medium and large AntMaze tasks (Tarasov et al., 2022).

[4]MSG numerical results are not available for Gym-MuJoCo tasks and Adroit tasks were not benchmarked.

Table 17: ReBRAC evaluation on AntMaze domain. We report the final normalized score averaged over 10 unseen training seeds on v2 datasets. CQL scores are taken from Ghasemipour et al. (2022). RORL scores are taken from Yang et al. (2022).

| | Ensemble-free | | | | Ensemble-based | | |
|---|---|---|---|---|---|---|---|
| Task Name | TD3+BC | IQL | CQL | SAC-RND | RORL | MSG | ReBRAC, our |
| antmaze-umaze | $66.3 \pm 6.2$ | $83.3 \pm 4.5$ | 74.0 | $97.0 \pm 1.5$ | $97.7 \pm 1.9$ | $\mathbf{97.9} \pm 1.3$ | $\underline{97.8} \pm 1.0$ |
| antmaze-umaze-diverse | $53.8 \pm 8.5$ | $70.6 \pm 3.7$ | 84.0 | $66.0 \pm 25.0$ | $\mathbf{90.7} \pm 2.9$ | $79.3 \pm 3.0$ | $\underline{88.3} \pm 13.0$ |
| antmaze-medium-play | $26.5 \pm 18.4$ | $64.6 \pm 4.9$ | 61.2 | $38.5 \pm 29.4$ | $76.3 \pm 2.5$ | $\mathbf{85.9} \pm 3.9$ | $\underline{84.0} \pm 4.2$ |
| antmaze-medium-diverse | $25.9 \pm 15.3$ | $61.7 \pm 6.1$ | 53.7 | $74.7 \pm 10.7$ | $69.3 \pm 3.3$ | $\mathbf{84.6} \pm 5.2$ | $\underline{76.3} \pm 13.5$ |
| antmaze-large-play | $0.0 \pm 0.0$ | $42.5 \pm 6.5$ | 15.8 | $43.9 \pm 29.2$ | $16.3 \pm 11.1$ | $\mathbf{64.3} \pm 12.7$ | $\underline{60.4} \pm 26.1$ |
| antmaze-large-diverse | $0.0 \pm 0.0$ | $27.6 \pm 7.8$ | 14.9 | $45.7 \pm 28.5$ | $41.0 \pm 10.7$ | $\mathbf{71.3} \pm 5.3$ | $\underline{54.4} \pm 25.1$ |
| Average | 28.7 | 58.3 | 50.6 | 60.9 | 65.2 | $\mathbf{80.5}$ | $\underline{76.8}$ |

Table 18: ReBRAC evaluation on Adroit domain. We report the final normalized score averaged over 10 unseen training seeds on v1 datasets. BC, CQL, EDAC and RORL scores are taken from Yang et al. (2022).

| | | Ensemble-free | | | | Ensemble-based | | |
|---|---|---|---|---|---|---|---|---|
| Task Name | BC | TD3+BC | IQL | CQL | SAC-RND | RORL | EDAC | ReBRAC, our |
| pen-human | 34.4 | $\underline{81.8} \pm 14.9$ | $81.5 \pm 17.5$ | 37.5 | $5.6 \pm 5.8$ | $33.7 \pm 7.6$ | $51.2 \pm 8.6$ | $\mathbf{103.5} \pm 14.1$ |
| pen-cloned | 56.9 | $61.4 \pm 19.3$ | $\underline{77.2} \pm 17.7$ | 39.2 | $2.5 \pm 6.1$ | $35.7 \pm 35.7$ | $68.2 \pm 7.3$ | $\mathbf{91.8} \pm 21.7$ |
| pen-expert | 85.1 | $\underline{146.0} \pm 7.3$ | $133.6 \pm 16.0$ | 107.0 | $45.4 \pm 22.9$ | $130.3 \pm 4.2$ | $122.8 \pm 14.1$ | $\mathbf{154.1} \pm 5.4$ |
| door-human | 0.5 | $-0.1 \pm 0.0$ | $3.1 \pm 2.0$ | $\underline{9.9}$ | $0.0 \pm 0.0$ | $3.7 \pm 0.7$ | $\mathbf{10.7} \pm 6.8$ | $0.0 \pm 0.1$ |
| door-cloned | -0.1 | $0.1 \pm 0.6$ | $0.8 \pm 1.0$ | 0.4 | $0.2 \pm 0.8$ | $-0.1 \pm 0.1$ | $\mathbf{9.6} \pm 8.3$ | $\underline{1.1} \pm 2.6$ |
| door-expert | 34.9 | $84.6 \pm 44.5$ | $\mathbf{105.3} \pm 2.8$ | 101.5 | $73.6 \pm 26.7$ | $\underline{104.9} \pm 0.9$ | $-0.3 \pm 0.1$ | $104.6 \pm 2.4$ |
| hammer-human | 1.5 | $0.4 \pm 0.4$ | $\underline{2.5} \pm 1.9$ | $\mathbf{4.4}$ | $-0.1 \pm 0.1$ | $2.3 \pm 2.3$ | $0.8 \pm 0.4$ | $0.2 \pm 0.2$ |
| hammer-cloned | 0.8 | $0.8 \pm 0.7$ | $1.1 \pm 0.5$ | 2.1 | $0.1 \pm 0.4$ | $\underline{1.7} \pm 1.7$ | $0.3 \pm 0.0$ | $\mathbf{6.7} \pm 3.7$ |
| hammer-expert | 125.6 | $117.0 \pm 30.9$ | $129.6 \pm 0.5$ | 86.7 | $24.8 \pm 39.4$ | $\underline{132.2} \pm 0.7$ | $0.2 \pm 0.0$ | $\mathbf{133.8} \pm 0.7$ |
| relocate-human | 0.0 | $-0.2 \pm 0.0$ | $\underline{0.1} \pm 0.1$ | $\mathbf{0.2}$ | $0.0 \pm 0.0$ | $0.0 \pm 0.0$ | $\underline{0.1} \pm 0.1$ | $0.0 \pm 0.0$ |
| relocate-cloned | -0.1 | $-0.1 \pm 0.1$ | $\underline{0.2} \pm 0.4$ | -0.1 | $0.0 \pm 0.0$ | $0.0 \pm 0.0$ | $0.0 \pm 0.0$ | $\mathbf{0.9} \pm 1.6$ |
| relocate-expert | 101.3 | $\mathbf{107.3} \pm 1.6$ | $106.5 \pm 2.5$ | 95.0 | $3.4 \pm 4.5$ | $47.8 \pm 13.5$ | $-0.3 \pm 0.0$ | $106.6 \pm 3.2$ |
| Average w/o expert | 11.7 | 18.0 | $\underline{20.8}$ | 11.7 | 1.0 | 9.6 | 17.4 | $\mathbf{25.5}$ |
| Average | 36.7 | 49.9 | $\underline{53.4}$ | 40.3 | 12.9 | 41.0 | 21.9 | $\mathbf{58.6}$ |

# D   Feature Normalization

Table 19: Average normalized score over the final evaluation and ten unseen training seeds on D4RL tasks for 2 types of normalization: LayerNorm (LN) and Feature Norm (FN). The symbol ± represents the standard deviation across the seeds. For both variatns we tune hyperparameters using the same grid.

| Task Name | ReBRAC + LN | ReBRAC + FN |
|---|---|---|
| halfcheetah-random | $29.5 \pm 1.5$ | $\mathbf{31.4} \pm 2.7$ |
| halfcheetah-medium | $65.6 \pm 1.0$ | $\mathbf{66.1} \pm 1.2$ |
| halfcheetah-expert | $\mathbf{105.9} \pm 1.7$ | $104.1 \pm 3.7$ |
| halfcheetah-medium-expert | $\mathbf{101.1} \pm 5.2$ | $100.9 \pm 4.7$ |
| halfcheetah-medium-replay | $51.0 \pm 0.8$ | $\mathbf{54.7} \pm 1.0$ |
| halfcheetah-full-replay | $\mathbf{82.1} \pm 1.1$ | $81.5 \pm 1.6$ |
| hopper-random | $8.1 \pm 2.4$ | $\mathbf{8.2} \pm 2.2$ |
| hopper-medium | $102.0 \pm 1.0$ | $\mathbf{102.4} \pm 0.2$ |
| hopper-expert | $\mathbf{100.1} \pm 8.3$ | $99.7 \pm 11.7$ |
| hopper-medium-expert | $107.0 \pm 6.4$ | $\mathbf{107.7} \pm 6.4$ |
| hopper-medium-replay | $\mathbf{98.1} \pm 5.3$ | $91.0 \pm 15.3$ |
| hopper-full-replay | $\mathbf{107.1} \pm 0.4$ | $106.7 \pm 0.5$ |
| walker2d-random | $\mathbf{18.4} \pm 4.5$ | $0.0 \pm 0.7$ |
| walker2d-medium | $\mathbf{82.5} \pm 3.6$ | $81.8 \pm 4.7$ |
| walker2d-expert | $\mathbf{112.3} \pm 0.2$ | $110.7 \pm 3.2$ |
| walker2d-medium-expert | $\mathbf{111.6} \pm 0.3$ | $100.6 \pm 34.3$ |
| walker2d-medium-replay | $77.3 \pm 7.9$ | $\mathbf{80.2} \pm 8.1$ |
| walker2d-full-replay | $\mathbf{102.2} \pm 1.7$ | $101.2 \pm 2.7$ |
| Gym-MuJoCo average | $\mathbf{81.2}$ | $79.3$ |
| antmaze-umaze | $\mathbf{97.8} \pm 1.0$ | $96.8 \pm 1.6$ |
| antmaze-umaze-diverse | $\mathbf{88.3} \pm 13.0$ | $88.5 \pm 7.5$ |
| antmaze-medium-play | $84.0 \pm 4.2$ | $\mathbf{84.1} \pm 10.1$ |
| antmaze-medium-diverse | $\mathbf{76.3} \pm 13.5$ | $75.6 \pm 13.7$ |
| antmaze-large-play | $\mathbf{60.4} \pm 26.1$ | $55.0 \pm 30.0$ |
| antmaze-large-diverse | $54.4 \pm 25.1$ | $\mathbf{66.4} \pm 7.4$ |
| AntMaze average | $76.8$ | $\mathbf{77.7}$ |
| pen-human | $103.5 \pm 14.1$ | $\mathbf{107.0} \pm 13.8$ |
| pen-cloned | $\mathbf{91.8} \pm 21.7$ | $84.9 \pm 20.1$ |
| pen-expert | $\mathbf{154.1} \pm 5.4$ | $151.6 \pm 4.7$ |
| door-human | $0.0 \pm 0.1$ | $0.0 \pm 0.0$ |
| door-cloned | $\mathbf{1.1} \pm 2.6$ | $0.1 \pm 0.1$ |
| door-expert | $104.6 \pm 2.4$ | $\mathbf{105.1} \pm 1.4$ |
| hammer-human | $\mathbf{0.2} \pm 0.2$ | $\mathbf{0.2} \pm 0.1$ |
| hammer-cloned | $6.7 \pm 3.7$ | $\mathbf{10.1} \pm 9.5$ |
| hammer-expert | $\mathbf{133.8} \pm 0.7$ | $133.4 \pm 1.5$ |
| relocate-human | $0.0 \pm 0.0$ | $0.0 \pm 0.0$ |
| relocate-cloned | $0.9 \pm 1.6$ | $\mathbf{1.2} \pm 1.9$ |
| relocate-expert | $106.6 \pm 3.2$ | $\mathbf{108.7} \pm 3.0$ |
| Adroit average w/o expert | $\mathbf{25.5}$ | $25.4$ |
| Adroit average | $\mathbf{58.6}$ | $58.5$ |

# E  Computational costs

Table 20: Computational costs for algorithms in Table 2.

| Algorithm | Number of runs | Approximate hours per run |
|---|---|---|
| TD3+BC, tuning | 360 | 0.3 |
| IQL, tuning | 1440 | 1.8 |
| ReBRAC, tuning | 1440 | 0.4 |
| TD3+BC, eval | 180 | 0.2 |
| IQL, eval | 180 | 1.8 |
| SAC-RND, eval | 180 | 1.8 |
| ReBRAC, eval | 180 | 0.4 |
| **Sum** | 3960 | 4032.0 |

Table 21: Computational costs for algorithms in Table 3 and Table 17.

| Algorithm | Number of runs | Approximate hours per run |
|---|---|---|
| TD3+BC, tuning | 96 | 0.5 |
| IQL, tuning | 480 | 2.1 |
| ReBRAC, tuning | 384 | 0.6 |
| TD3+BC, eval | 60 | 0.5 |
| IQL, eval | 60 | 2.0 |
| SAC-RND, eval | 60 | 2.9 |
| MSG, eval | 60 | 5.1 |
| ReBRAC, eval | 60 | 0.4 |
| **Sum** | 1260 | 1940.4 |

Table 22: Computational costs for algorithms in Table 4.

| Algorithm | Number of runs | Approximate hours per run |
|---|---|---|
| TD3+BC, tuning | 240 | 0.3 |
| IQL, tuning | 960 | 1.8 |
| SAC-RND, tuning | 1200 | 1.1 |
| ReBRAC, tuning | 960 | 0.3 |
| TD3+BC, eval | 120 | 0.2 |
| IQL, eval | 120 | 1.9 |
| SAC-RND, eval | 120 | 1.1 |
| ReBRAC, eval | 120 | 0.3 |
| **Sum** | 3840 | 3828.0 |

Table 23: Computational costs for algorithms in Table 5.

| Algorithm | Number of runs | Approximate hours per run |
|---|---|---|
| ReBRAC, tuning | 600 | 10.6 |
| ReBRAC, eval | 75 | 10.5 |
| **Sum** | 675 | 7147.5 |

Table 24: Computational costs for algorithms in Table 8 and Figure 2.

| Algorithm | Number of runs | Approximate hours per run |
|---|---|---|
| ReBRAC, ablations eval | 1104 | 1.4 |
| **Sum** | **1104** | **1545.6** |

# F   Expected Online Performance

Table 25: TD3+BC, IQL and ReBRAC Expected Online Performance under uniform policy selection on HalfCheetah tasks.

| | random | | | medium | | | expert | | | medium-expert | | | medium-replay | | | full-replay | | |
|---|---|---|---|---|---|---|---|---|---|---|---|---|---|---|---|---|---|---|
| Policies | TD3+BC | IQL | ReBRAC | TD3+BC | IQL | ReBRAC | TD3+BC | IQL | ReBRAC | TD3+BC | IQL | ReBRAC | TD3+BC | IQL | ReBRAC | TD3+BC | IQL | ReBRAC |
| 1 | 14.6 ± 9.3 | 10.2 ± 6.8 | 17.6 ± 8.2 | 48.0 ± 5.8 | 48.0 ± 1.3 | 56.1 ± 6.3 | 59.5 ± 40.5 | 93.9 ± 4.2 | 90.7 ± 21.5 | 68.1 ± 31.4 | 87.7 ± 5.5 | 97.7 ± 6.8 | 34.7 ± 14.2 | 43.4 ± 1.3 | 47.7 ± 3.0 | 67.7 ± 12.5 | 73.1 ± 1.9 | 78.7 ± 3.3 |
| 2 | 19.8 ± 8.0 | 14.1 ± 5.9 | 22.2 ± 7.0 | 51.1 ± 5.4 | 48.8 ± 1.1 | 59.6 ± 5.8 | 80.4 ± 28.1 | 95.6 ± 1.5 | 100.8 ± 11.4 | 83.7 ± 18.1 | 90.8 ± 3.7 | 101.2 ± 3.8 | 41.5 ± 7.7 | 44.2 ± 0.8 | 49.4 ± 2.6 | 73.8 ± 7.0 | 74.1 ± 1.1 | 80.5 ± 2.6 |
| 3 | 22.5 ± 6.8 | 16.1 ± 4.6 | 24.6 ± 5.9 | 52.9 ± 5.0 | 49.1 ± 0.9 | 61.6 ± 4.9 | 88.2 ± 18.1 | 96.0 ± 0.8 | 103.6 ± 6.0 | 88.5 ± 10.1 | 92.0 ± 2.7 | 102.4 ± 2.4 | 43.6 ± 4.2 | 44.4 ± 0.6 | 50.3 ± 2.1 | 75.8 ± 4.2 | 74.5 ± 0.8 | 81.4 ± 2.0 |
| 4 | 24.2 ± 5.7 | 17.2 ± 3.6 | 26.1 ± 4.9 | 54.0 ± 4.7 | 49.4 ± 0.7 | 62.8 ± 4.2 | 91.3 ± 11.5 | 96.2 ± 0.5 | 104.7 ± 3.5 | 90.3 ± 5.9 | 92.7 ± 2.2 | 102.9 ± 1.9 | 44.5 ± 2.6 | 44.6 ± 0.4 | 50.8 ± 1.6 | 76.7 ± 3.1 | 74.7 ± 0.7 | 81.9 ± 1.6 |
| 5 | 25.3 ± 4.8 | 17.9 ± 2.8 | 27.0 ± 4.1 | 54.9 ± 4.3 | 49.5 ± 0.6 | 63.6 ± 3.5 | 92.7 ± 7.3 | 96.3 ± 0.4 | 105.2 ± 2.5 | 91.0 ± 3.6 | 93.2 ± 1.8 | 103.3 ± 1.6 | 44.9 ± 1.8 | 44.6 ± 0.4 | 51.1 ± 1.3 | 77.3 ± 2.5 | 74.9 ± 0.6 | 82.2 ± 1.2 |
| 6 | - | 18.4 ± 2.3 | 27.7 ± 3.4 | - | 49.6 ± 0.5 | 64.1 ± 3.0 | - | 96.3 ± 0.3 | 105.6 ± 2.1 | - | 93.5 ± 1.5 | 103.5 ± 1.5 | - | 44.7 ± 0.3 | 51.3 ± 1.1 | - | 75.0 ± 0.5 | 82.3 ± 1.0 |
| 7 | - | 18.7 ± 1.8 | 28.1 ± 2.9 | - | 49.7 ± 0.5 | 64.5 ± 2.6 | - | 96.4 ± 0.3 | 105.9 ± 1.9 | - | 93.7 ± 1.3 | 103.7 ± 1.4 | - | 44.8 ± 0.3 | 51.4 ± 0.9 | - | 75.0 ± 0.5 | 82.5 ± 0.8 |
| 8 | - | 18.9 ± 1.5 | 28.5 ± 2.5 | - | 49.7 ± 0.4 | 64.8 ± 2.3 | - | 96.4 ± 0.3 | 106.1 ± 1.7 | - | 93.8 ± 1.1 | 103.9 ± 1.4 | - | 44.8 ± 0.2 | 51.5 ± 0.7 | - | 75.1 ± 0.5 | 82.6 ± 0.7 |
| 9 | - | 19.0 ± 1.3 | 28.7 ± 2.1 | - | 49.8 ± 0.4 | 65.0 ± 2.0 | - | 96.4 ± 0.2 | 106.3 ± 1.6 | - | 93.9 ± 1.0 | 104.0 ± 1.3 | - | 44.8 ± 0.2 | 51.6 ± 0.7 | - | 75.1 ± 0.4 | 82.6 ± 0.6 |
| 10 | - | 19.1 ± 1.1 | 28.9 ± 1.8 | - | 49.8 ± 0.4 | 65.2 ± 1.7 | - | 96.5 ± 0.2 | 106.4 ± 1.5 | - | 94.0 ± 0.9 | 104.1 ± 1.3 | - | 44.8 ± 0.2 | 51.6 ± 0.6 | - | 75.2 ± 0.4 | 82.7 ± 0.6 |
| 11 | - | 19.2 ± 0.9 | 29.0 ± 1.6 | - | 49.9 ± 0.4 | 65.3 ± 1.5 | - | 96.5 ± 0.2 | 106.6 ± 1.3 | - | 94.1 ± 0.8 | 104.2 ± 1.3 | - | 44.8 ± 0.2 | 51.7 ± 0.6 | - | 75.2 ± 0.4 | 82.7 ± 0.5 |
| 12 | - | 19.3 ± 0.8 | 29.2 ± 1.4 | - | 49.9 ± 0.3 | 65.4 ± 1.3 | - | 96.5 ± 0.2 | 106.7 ± 1.2 | - | 94.2 ± 0.7 | 104.3 ± 1.3 | - | 44.9 ± 0.2 | 51.7 ± 0.5 | - | 75.2 ± 0.4 | 82.8 ± 0.4 |
| 13 | - | 19.3 ± 0.7 | 29.2 ± 1.2 | - | 49.9 ± 0.3 | 65.5 ± 1.1 | - | 96.5 ± 0.2 | 106.8 ± 1.1 | - | 94.2 ± 0.7 | 104.4 ± 1.3 | - | 44.9 ± 0.2 | 51.7 ± 0.5 | - | 75.3 ± 0.4 | 82.8 ± 0.4 |
| 14 | - | 19.4 ± 0.6 | 29.3 ± 1.1 | - | 49.9 ± 0.3 | 65.5 ± 1.0 | - | 96.5 ± 0.2 | 106.8 ± 1.1 | - | 94.3 ± 0.6 | 104.4 ± 1.2 | - | 44.9 ± 0.1 | 51.8 ± 0.5 | - | 75.3 ± 0.3 | 82.8 ± 0.4 |
| 15 | - | 19.4 ± 0.5 | 29.4 ± 1.0 | - | 49.9 ± 0.3 | 65.6 ± 0.9 | - | 96.5 ± 0.2 | 106.9 ± 1.0 | - | 94.3 ± 0.6 | 104.5 ± 1.2 | - | 44.9 ± 0.1 | 51.8 ± 0.5 | - | 75.3 ± 0.3 | 82.9 ± 0.3 |
| 16 | - | 19.5 ± 0.5 | 29.4 ± 0.9 | - | 50.0 ± 0.3 | 65.6 ± 0.8 | - | 96.5 ± 0.1 | 107.0 ± 0.9 | - | 94.4 ± 0.5 | 104.6 ± 1.2 | - | 44.9 ± 0.1 | 51.8 ± 0.5 | - | 75.3 ± 0.3 | 82.9 ± 0.3 |
| 17 | - | 19.5 ± 0.4 | 29.5 ± 0.9 | - | 50.0 ± 0.3 | 65.7 ± 0.7 | - | 96.6 ± 0.1 | 107.0 ± 0.8 | - | 94.4 ± 0.5 | 104.6 ± 1.2 | - | 44.9 ± 0.1 | 51.9 ± 0.5 | - | 75.3 ± 0.3 | 82.9 ± 0.3 |
| 18 | - | 19.5 ± 0.4 | 29.5 ± 0.8 | - | 50.0 ± 0.3 | 65.7 ± 0.6 | - | 96.6 ± 0.1 | 107.1 ± 0.8 | - | 94.4 ± 0.5 | 104.7 ± 1.2 | - | 44.9 ± 0.1 | 51.9 ± 0.5 | - | 75.4 ± 0.3 | 82.9 ± 0.2 |
| 19 | - | 19.5 ± 0.4 | 29.6 ± 0.8 | - | 50.0 ± 0.3 | 65.7 ± 0.5 | - | 96.6 ± 0.1 | 107.1 ± 0.7 | - | 94.5 ± 0.4 | 104.7 ± 1.1 | - | 44.9 ± 0.1 | 51.9 ± 0.4 | - | 75.4 ± 0.3 | 82.9 ± 0.2 |
| 20 | - | 19.5 ± 0.3 | 29.6 ± 0.7 | - | 50.0 ± 0.3 | 65.7 ± 0.5 | - | 96.6 ± 0.1 | 107.1 ± 0.7 | - | 94.5 ± 0.4 | 104.8 ± 1.1 | - | 44.9 ± 0.1 | 51.9 ± 0.4 | - | 75.4 ± 0.3 | 82.9 ± 0.2 |

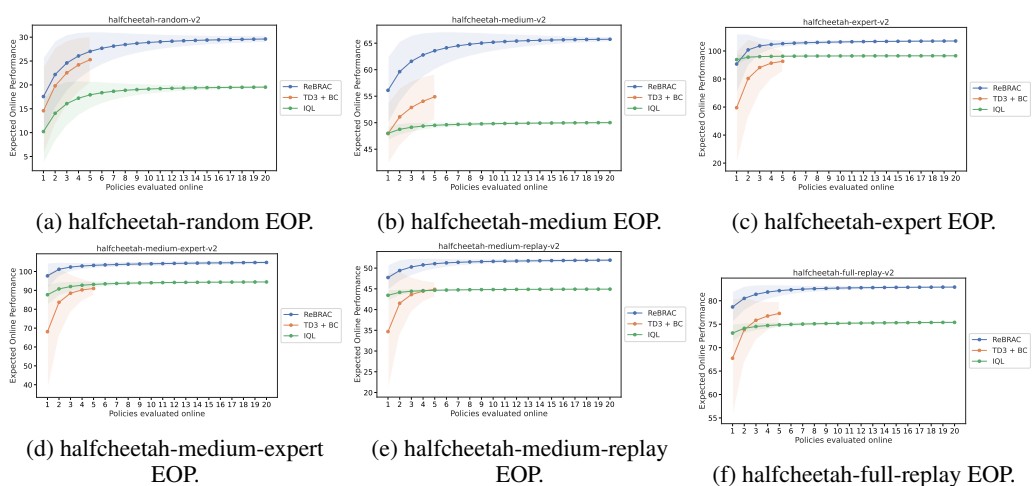

(a) halfcheetah-random EOP.  (b) halfcheetah-medium EOP.  (c) halfcheetah-expert EOP.

(d) halfcheetah-medium-expert EOP.  (e) halfcheetah-medium-replay EOP.  (f) halfcheetah-full-replay EOP.

Figure 3: TD3+BC, IQL and ReBRAC visualised Expected Online Performance under uniform policy selection on HalfCheetah tasks.

Table 26: TD3+BC, IQL and ReBRAC Expected Online Performance under uniform policy selection on Hopper tasks.

| | random | | | medium | | | expert | | | medium-expert | | | medium-replay | | | full-replay | | |
|---|---|---|---|---|---|---|---|---|---|---|---|---|---|---|---|---|---|---|
| Policies | TD3+BC | IQL | ReBRAC | TD3+BC | IQL | ReBRAC | TD3+BC | IQL | ReBRAC | TD3+BC | IQL | ReBRAC | TD3+BC | IQL | ReBRAC | TD3+BC | IQL | ReBRAC |
| 1 | 8.3 ± 4.5 | 7.5 ± 1.1 | 7.5 ± 0.9 | 39.8 ± 33.0 | 59.0 ± 4.9 | 69.5 ± 32.6 | 72.2 ± 47.1 | 96.6 ± 17.3 | 58.3 ± 40.5 | 55.0 ± 45.5 | 83.3 ± 28.4 | 58.7 ± 39.8 | 62.5 ± 14.9 | 63.8 ± 28.3 | 67.2 ± 28.4 | 68.7 ± 27.6 | 94.5 ± 20.8 | 96.7 ± 17.8 |
| 2 | 10.8 ± 4.1 | 8.1 ± 1.4 | 8.0 ± 0.5 | 57.7 ± 27.1 | 61.8 ± 3.8 | 86.6 ± 19.7 | 96.2 ± 32.3 | 105.8 ± 10.5 | 80.7 ± 30.8 | 79.3 ± 36.3 | 98.5 ± 17.5 | 81.1 ± 31.4 | 70.7 ± 10.2 | 78.9 ± 19.5 | 82.7 ± 22.8 | 83.7 ± 25.7 | 104.0 ± 9.9 | 104.5 ± 7.2 |
| 3 | 12.2 ± 3.7 | 8.4 ± 1.6 | 8.1 ± 0.4 | 66.5 ± 20.8 | 63.1 ± 3.2 | 92.8 ± 12.8 | 105.0 ± 20.6 | 109.0 ± 6.4 | 90.8 ± 22.1 | 90.9 ± 26.5 | 104.1 ± 11.8 | 91.8 ± 23.8 | 74.0 ± 7.1 | 85.2 ± 14.2 | 90.1 ± 16.6 | 92.2 ± 22.0 | 106.2 ± 4.7 | 106.2 ± 3.4 |
| 4 | 13.1 ± 3.4 | 8.7 ± 1.7 | 8.2 ± 0.3 | 71.4 ± 16.3 | 63.9 ± 2.9 | 95.9 ± 9.3 | 108.4 ± 13.0 | 110.3 ± 4.1 | 95.9 ± 16.0 | 96.8 ± 19.3 | 106.7 ± 8.3 | 97.5 ± 18.2 | 75.6 ± 5.3 | 88.6 ± 11.5 | 93.9 ± 11.8 | 97.4 ± 18.4 | 107.0 ± 2.5 | 106.9 ± 1.9 |
| 5 | 13.7 ± 3.1 | 8.9 ± 1.8 | 8.3 ± 0.2 | 74.4 ± 13.3 | 64.5 ± 2.6 | 97.7 ± 7.3 | 109.8 ± 8.2 | 111.0 ± 2.7 | 98.7 ± 11.8 | 100.2 ± 14.6 | 108.1 ± 6.1 | 101.0 ± 14.3 | 76.5 ± 4.0 | 90.8 ± 9.9 | 95.8 ± 8.4 | 100.8 ± 15.3 | 107.3 ± 1.5 | 107.2 ± 1.3 |
| 6 | - | 9.2 ± 1.8 | 8.3 ± 0.2 | - | 64.9 ± 2.4 | 98.8 ± 5.9 | - | 111.4 ± 2.0 | 100.3 ± 8.9 | - | 108.9 ± 4.5 | 103.2 ± 11.4 | - | 92.3 ± 8.7 | 97.0 ± 6.1 | - | 107.5 ± 1.1 | 107.4 ± 0.9 |
| 7 | - | 9.4 ± 1.8 | 8.3 ± 0.2 | - | 65.3 ± 2.1 | 99.6 ± 4.9 | - | 111.6 ± 1.5 | 101.4 ± 6.9 | - | 109.5 ± 3.5 | 104.8 ± 9.4 | - | 93.5 ± 7.7 | 97.6 ± 4.5 | - | 107.6 ± 0.9 | 107.5 ± 0.7 |
| 8 | - | 9.5 ± 1.8 | 8.4 ± 0.2 | - | 65.5 ± 2.0 | 100.2 ± 4.2 | - | 111.8 ± 1.2 | 102.1 ± 5.5 | - | 109.8 ± 2.8 | 105.8 ± 7.8 | - | 94.4 ± 6.8 | 98.1 ± 3.4 | - | 107.8 ± 0.8 | 107.5 ± 0.5 |
| 9 | - | 9.7 ± 1.8 | 8.4 ± 0.2 | - | 65.8 ± 1.8 | 100.6 ± 3.6 | - | 111.9 ± 1.0 | 102.7 ± 4.5 | - | 110.1 ± 2.4 | 106.6 ± 6.7 | - | 95.0 ± 6.0 | 98.4 ± 2.6 | - | 107.8 ± 0.7 | 107.6 ± 0.4 |
| 10 | - | 9.9 ± 1.8 | 8.4 ± 0.1 | - | 65.9 ± 1.7 | 100.9 ± 3.1 | - | 112.0 ± 0.9 | 103.4 ± 3.9 | - | 110.3 ± 2.2 | 107.3 ± 5.8 | - | 95.6 ± 5.3 | 98.6 ± 2.1 | - | 108.0 ± 0.6 | 107.6 ± 0.3 |
| 11 | - | 10.0 ± 1.8 | 8.4 ± 0.1 | - | 66.1 ± 1.6 | 101.2 ± 2.7 | - | 112.0 ± 0.8 | 103.4 ± 3.5 | - | 110.4 ± 2.0 | 107.7 ± 5.1 | - | 96.0 ± 4.7 | 98.7 ± 1.8 | - | 108.0 ± 0.6 | 107.6 ± 0.3 |
| 12 | - | 10.1 ± 1.8 | 8.4 ± 0.1 | - | 66.2 ± 1.5 | 101.3 ± 2.3 | - | 112.1 ± 0.7 | 103.9 ± 2.9 | - | 110.6 ± 1.9 | 108.1 ± 4.5 | - | 96.3 ± 4.1 | 98.9 ± 1.5 | - | 108.1 ± 0.5 | 107.7 ± 0.3 |
| 13 | - | 10.2 ± 1.7 | 8.4 ± 0.1 | - | 66.3 ± 1.4 | 101.5 ± 2.0 | - | 112.2 ± 0.7 | 103.9 ± 2.9 | - | 110.7 ± 1.8 | 108.4 ± 4.0 | - | 96.5 ± 3.6 | 99.0 ± 1.4 | - | 108.1 ± 0.5 | 107.7 ± 0.2 |
| 14 | - | 10.3 ± 1.7 | 8.4 ± 0.1 | - | 66.4 ± 1.3 | 101.6 ± 1.8 | - | 112.2 ± 0.6 | 104.1 ± 2.8 | - | 110.8 ± 1.8 | 108.7 ± 3.6 | - | 96.7 ± 3.2 | 99.1 ± 1.2 | - | 108.1 ± 0.5 | 107.7 ± 0.2 |
| 15 | - | 10.4 ± 1.7 | 8.4 ± 0.1 | - | 66.5 ± 1.3 | 101.7 ± 1.6 | - | 112.2 ± 0.6 | 104.3 ± 2.6 | - | 110.9 ± 1.7 | 108.9 ± 3.3 | - | 96.9 ± 2.8 | 99.1 ± 1.1 | - | 108.1 ± 0.5 | 107.7 ± 0.2 |
| 16 | - | 10.5 ± 1.6 | 8.5 ± 0.1 | - | 66.6 ± 1.2 | 101.8 ± 1.4 | - | 112.3 ± 0.6 | 104.4 ± 2.5 | - | 111.0 ± 1.7 | 109.1 ± 3.0 | - | 97.0 ± 2.5 | 99.2 ± 1.1 | - | 108.2 ± 0.4 | 107.7 ± 0.1 |
| 17 | - | 10.6 ± 1.6 | 8.5 ± 0.1 | - | 66.6 ± 1.2 | 101.9 ± 1.3 | - | 112.3 ± 0.6 | 104.6 ± 2.4 | - | 111.1 ± 1.7 | 109.3 ± 2.7 | - | 97.1 ± 2.2 | 99.3 ± 1.0 | - | 108.2 ± 0.4 | 107.7 ± 0.1 |
| 18 | - | 10.7 ± 1.5 | 8.5 ± 0.1 | - | 66.7 ± 1.1 | 101.9 ± 1.2 | - | 112.3 ± 0.5 | 104.7 ± 2.3 | - | 111.2 ± 1.7 | 109.4 ± 2.5 | - | 97.2 ± 1.9 | 99.3 ± 1.0 | - | 108.2 ± 0.4 | 107.7 ± 0.1 |
| 19 | - | 10.8 ± 1.5 | 8.5 ± 0.1 | - | 66.8 ± 1.1 | 102.0 ± 1.1 | - | 112.4 ± 0.5 | 104.8 ± 2.2 | - | 111.2 ± 1.6 | 109.5 ± 2.3 | - | 97.3 ± 1.7 | 99.4 ± 0.9 | - | 108.2 ± 0.4 | 107.7 ± 0.1 |
| 20 | - | 10.8 ± 1.5 | 8.5 ± 0.1 | - | 66.8 ± 1.0 | 102.0 ± 1.0 | - | 112.4 ± 0.5 | 104.9 ± 2.1 | - | 111.3 ± 1.6 | 109.6 ± 2.1 | - | 97.3 ± 1.5 | 99.4 ± 0.9 | - | 108.2 ± 0.3 | 107.7 ± 0.1 |

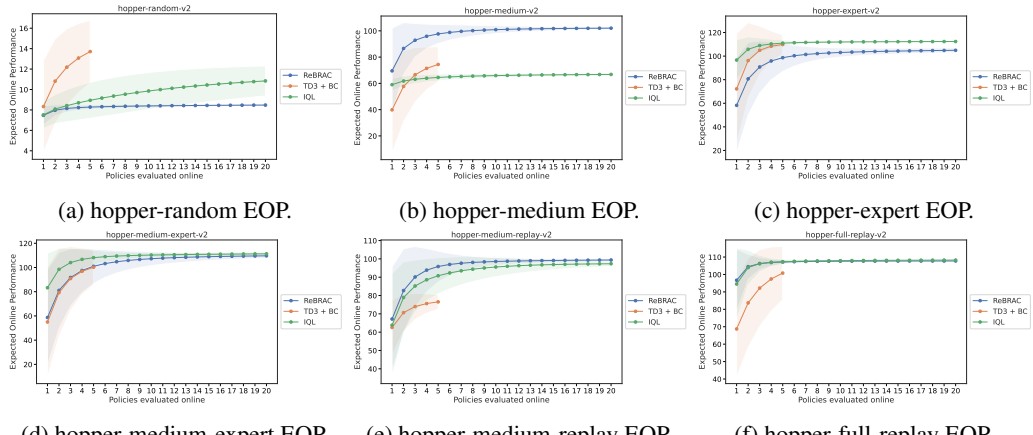

(a) hopper-random EOP.  (b) hopper-medium EOP.  (c) hopper-expert EOP.

(d) hopper-medium-expert EOP.  (e) hopper-medium-replay EOP.  (f) hopper-full-replay EOP.

Figure 4: TD3+BC, IQL and ReBRAC visualised Expected Online Performance under uniform policy selection on Hopper tasks.

Table 27: TD3+BC, IQL and ReBRAC Expected Online Performance under uniform policy selection on Walker2d tasks.

| | random | | | medium | | | expert | | | medium-expert | | | medium-replay | | | full-replay | | |
|---|---|---|---|---|---|---|---|---|---|---|---|---|---|---|---|---|---|---|
| Policies | TD3+BC | IQL | ReBRAC | TD3+BC | IQL | ReBRAC | TD3+BC | IQL | ReBRAC | TD3+BC | IQL | ReBRAC | TD3+BC | IQL | ReBRAC | TD3+BC | IQL | ReBRAC |
| 1 | 3.2 ± 1.0 | 6.3 ± 2.9 | **7.9 ± 6.5** | 41.3 ± 28.6 | **65.3 ± 17.8** | 54.1 ± 34.5 | 67.0 ± 52.4 | **110.3 ± 4.0** | 84.2 ± 45.7 | 70.9 ± 46.5 | **103.8 ± 12.2** | 83.2 ± 46.9 | 36.4 ± 25.2 | 51.9 ± 28.3 | **54.5 ± 26.0** | 78.9 ± 15.4 | 71.8 ± 28.7 | **86.1 ± 21.5** |
| 2 | 3.7 ± 0.8 | 7.8 ± 2.8 | **11.4 ± 6.6** | 57.0 ± 23.4 | **75.0 ± 13.0** | 72.5 ± 23.4 | 92.9 ± 39.0 | **112.1 ± 1.8** | 104.3 ± 25.4 | 94.8 ± 32.2 | **109.6 ± 6.4** | 103.8 ± 26.3 | 50.0 ± 19.0 | 67.5 ± 21.0 | **68.8 ± 18.7** | 87.1 ± 10.5 | 87.1 ± 17.9 | **96.2 ± 10.2** |
| 3 | 4.0 ± 0.7 | 8.8 ± 2.6 | **13.6 ± 6.3** | 64.7 ± 17.7 | 79.0 ± 8.7 | **79.6 ± 14.9** | 103.2 ± 26.0 | **112.7 ± 1.2** | 109.5 ± 13.0 | 103.7 ± 20.6 | **111.2 ± 3.3** | 109.2 ± 13.4 | 56.1 ± 13.8 | 74.2 ± 14.8 | **74.8 ± 13.1** | 90.3 ± 6.8 | 92.6 ± 11.1 | **99.0 ± 5.7** |
| 4 | 4.2 ± 0.6 | 9.4 ± 2.3 | **15.1 ± 5.9** | 68.9 ± 13.4 | 80.7 ± 5.8 | **82.6 ± 9.5** | 107.4 ± 16.8 | **113.0 ± 1.0** | 111.0 ± 6.6 | 107.3 ± 13.1 | **111.7 ± 1.7** | 110.7 ± 6.8 | 59.2 ± 10.5 | 77.4 ± 10.4 | **77.6 ± 9.2** | 91.6 ± 4.4 | 94.9 ± 7.2 | **100.3 ± 3.9** |
| 5 | 4.3 ± 0.4 | 9.9 ± 2.1 | **16.3 ± 5.4** | 71.3 ± 10.4 | 81.6 ± 3.9 | **83.9 ± 6.2** | 109.1 ± 10.7 | **113.2 ± 0.8** | 111.5 ± 3.4 | 108.6 ± 8.3 | **111.9 ± 0.9** | 111.2 ± 3.5 | 61.0 ± 8.5 | 79.1 ± 7.4 | **79.1 ± 6.5** | 92.3 ± 3.0 | 96.0 ± 4.8 | **100.8 ± 3.2** |
| 6 | - | 10.2 ± 1.9 | **17.2 ± 5.0** | - | 82.0 ± 2.7 | **84.6 ± 4.1** | - | **113.3 ± 0.7** | 111.7 ± 1.8 | - | **112.0 ± 0.6** | 111.4 ± 1.9 | - | **80.1 ± 5.4** | 80.0 ± 4.7 | - | 96.7 ± 3.4 | **101.5 ± 2.8** |
| 7 | - | 10.5 ± 1.7 | **17.9 ± 4.6** | - | 82.2 ± 1.9 | **85.0 ± 2.8** | - | **113.4 ± 0.6** | 111.8 ± 1.1 | - | **112.1 ± 0.4** | 111.5 ± 1.1 | - | **80.7 ± 4.1** | 80.5 ± 3.6 | - | 97.0 ± 2.5 | **101.9 ± 2.5** |
| 8 | - | 10.7 ± 1.6 | **18.4 ± 4.2** | - | 82.3 ± 1.3 | **85.2 ± 2.0** | - | **113.5 ± 0.6** | 111.9 ± 0.7 | - | **112.1 ± 0.3** | 111.6 ± 0.8 | - | **81.1 ± 3.2** | 80.9 ± 2.9 | - | 97.3 ± 1.9 | **102.1 ± 2.3** |
| 9 | - | 10.9 ± 1.4 | **18.9 ± 3.8** | - | 82.4 ± 0.9 | **85.3 ± 1.5** | - | **113.6 ± 0.5** | 111.9 ± 0.6 | - | **112.2 ± 0.3** | 111.7 ± 0.6 | - | **81.4 ± 2.6** | 81.2 ± 2.4 | - | 97.4 ± 1.5 | **102.4 ± 2.0** |
| 10 | - | 11.0 ± 1.3 | **19.3 ± 3.5** | - | 82.4 ± 0.7 | **85.4 ± 1.2** | - | **113.6 ± 0.5** | 112.0 ± 0.5 | - | **112.2 ± 0.3** | 111.7 ± 0.5 | - | **81.6 ± 2.1** | 81.4 ± 2.1 | - | 97.5 ± 1.2 | **102.5 ± 1.8** |
| 11 | - | 11.1 ± 1.3 | **19.6 ± 3.2** | - | 82.4 ± 0.5 | **85.5 ± 1.0** | - | **113.6 ± 0.4** | 112.0 ± 0.4 | - | **112.2 ± 0.3** | 111.7 ± 0.4 | - | **81.8 ± 1.9** | 81.5 ± 1.9 | - | 97.6 ± 1.0 | **102.7 ± 1.6** |
| 12 | - | 11.2 ± 1.2 | **19.8 ± 3.0** | - | 82.5 ± 0.3 | **85.6 ± 0.8** | - | **113.7 ± 0.4** | 112.0 ± 0.4 | - | **112.2 ± 0.3** | 111.8 ± 0.4 | - | **81.9 ± 1.7** | 81.7 ± 1.8 | - | 97.7 ± 0.9 | **102.8 ± 1.5** |
| 13 | - | 11.3 ± 1.1 | **20.1 ± 2.8** | - | 82.5 ± 0.3 | **85.6 ± 0.8** | - | **113.7 ± 0.4** | 112.0 ± 0.3 | - | **112.3 ± 0.2** | 111.8 ± 0.3 | - | **82.0 ± 1.5** | 81.8 ± 1.7 | - | 97.8 ± 0.7 | **102.9 ± 1.3** |
| 14 | - | 11.4 ± 1.0 | **20.2 ± 2.6** | - | 82.5 ± 0.2 | **85.7 ± 0.7** | - | **113.7 ± 0.3** | 112.1 ± 0.3 | - | **112.3 ± 0.2** | 111.8 ± 0.3 | - | **82.1 ± 1.4** | 81.9 ± 1.6 | - | 97.8 ± 0.7 | **103.0 ± 1.2** |
| 15 | - | 11.5 ± 1.0 | **20.4 ± 2.4** | - | 82.5 ± 0.1 | **85.7 ± 0.7** | - | **113.7 ± 0.3** | 112.1 ± 0.2 | - | **112.3 ± 0.2** | 111.8 ± 0.3 | - | **82.2 ± 1.3** | 82.0 ± 1.6 | - | 97.9 ± 0.6 | **103.0 ± 1.1** |
| 16 | - | 11.5 ± 0.9 | **20.6 ± 2.2** | - | 82.5 ± 0.1 | **85.8 ± 0.6** | - | **113.8 ± 0.3** | 112.1 ± 0.2 | - | **112.3 ± 0.2** | 111.8 ± 0.2 | - | **82.3 ± 1.2** | 82.1 ± 1.5 | - | 97.9 ± 0.5 | **103.1 ± 0.9** |
| 17 | - | 11.6 ± 0.9 | **20.7 ± 2.1** | - | 82.5 ± 0.1 | **85.8 ± 0.6** | - | **113.8 ± 0.3** | 112.1 ± 0.2 | - | **112.3 ± 0.2** | 111.9 ± 0.2 | - | **82.4 ± 1.2** | 82.2 ± 1.4 | - | 97.9 ± 0.5 | **103.1 ± 0.8** |
| 18 | - | 11.6 ± 0.8 | **20.8 ± 1.9** | - | 82.5 ± 0.1 | **85.8 ± 0.6** | - | **113.8 ± 0.3** | 112.1 ± 0.2 | - | **112.3 ± 0.2** | 111.9 ± 0.2 | - | **82.4 ± 1.1** | 82.3 ± 1.4 | - | 97.9 ± 0.5 | **103.2 ± 0.8** |
| 19 | - | 11.7 ± 0.8 | **20.9 ± 1.8** | - | 82.5 ± 0.1 | **85.9 ± 0.6** | - | **113.8 ± 0.2** | 112.1 ± 0.1 | - | **112.3 ± 0.2** | 111.9 ± 0.2 | - | **82.5 ± 1.0** | 82.3 ± 1.3 | - | 98.0 ± 0.4 | **103.2 ± 0.7** |
| 20 | - | 11.7 ± 0.7 | **21.0 ± 1.7** | - | 82.5 ± 0.1 | **85.9 ± 0.6** | - | **113.8 ± 0.2** | 112.1 ± 0.1 | - | **112.3 ± 0.2** | 111.9 ± 0.2 | - | **82.5 ± 1.0** | 82.4 ± 1.3 | - | 98.0 ± 0.4 | **103.2 ± 0.6** |

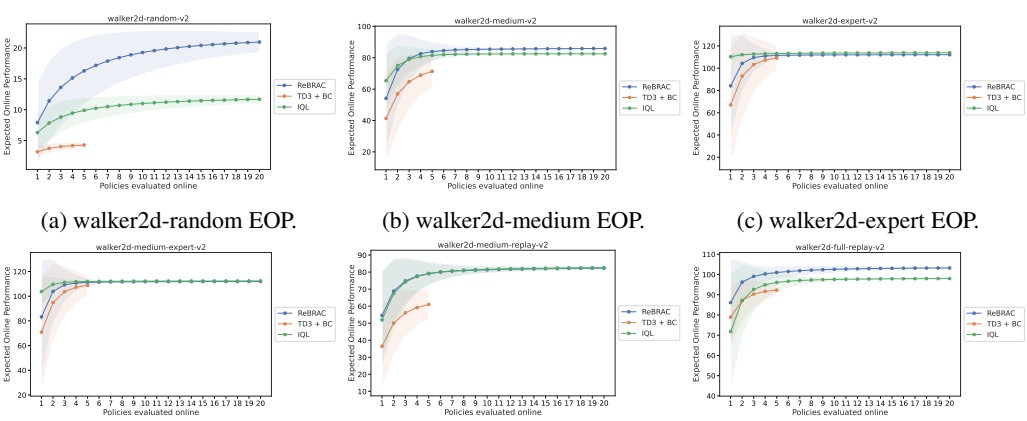

(a) walker2d-random EOP.  (b) walker2d-medium EOP.  (c) walker2d-expert EOP.

(d) walker2d-medium-expert EOP.  (e) walker2d-medium-replay EOP.  (f) walker2d-full-replay EOP.

Figure 5: TD3+BC, IQL and ReBRAC visualised Expected Online Performance under uniform policy selection on Walker2d tasks.

Table 28: TD3+BC, IQL and ReBRAC Expected Online Performance under uniform policy selection on AntMaze tasks.

| Policies | umaze | | | medium-play | | | large-play | | | umaze-diverse | | | medium-diverse | | | large-diverse | | |
|---|---|---|---|---|---|---|---|---|---|---|---|---|---|---|---|---|---|---|
| | TD3+BC | IQL | ReBRAC | TD3+BC | IQL | ReBRAC | TD3+BC | IQL | ReBRAC | TD3+BC | IQL | ReBRAC | TD3+BC | IQL | ReBRAC | TD3+BC | IQL | ReBRAC |
| 1 | 12.4 ± 24.8 | 64.3 ± 12.0 | 87.5 ± 10.9 | 7.5 ± 11.7 | 22.7 ± 29.5 | 75.0 ± 14.8 | 0.0 ± 0.0 | 10.9 ± 16.2 | 52.7 ± 21.4 | 9.6 ± 19.2 | 52.8 ± 11.1 | 70.4 ± 16.2 | 11.9 ± 13.8 | 21.3 ± 26.8 | 65.3 ± 26.3 | 0.2 ± 0.2 | 6.7 ± 10.3 | 56.8 ± 17.0 |
| 2 | 22.3 ± 29.8 | 71.1 ± 9.2 | 93.2 ± 6.2 | 13.0 ± 13.3 | 37.5 ± 30.9 | 82.7 ± 8.9 | 0.0 ± 0.0 | 18.6 ± 17.8 | 64.1 ± 13.0 | 17.3 ± 23.0 | 58.7 ± 11.3 | 79.3 ± 11.2 | 19.2 ± 14.0 | 35.0 ± 27.7 | 79.2 ± 16.0 | 0.3 ± 0.2 | 11.6 ± 11.6 | 66.2 ± 11.3 |
| 3 | 30.3 ± 31.0 | 74.3 ± 7.5 | 95.1 ± 3.7 | 17.0 ± 13.2 | 47.4 ± 28.5 | 85.4 ± 5.7 | 0.0 ± 0.0 | 24.0 ± 16.9 | 68.5 ± 9.3 | 23.4 ± 24.0 | 62.5 ± 10.4 | 83.1 ± 8.4 | 23.8 ± 12.7 | 44.0 ± 25.1 | 84.2 ± 9.8 | 0.4 ± 0.2 | 15.2 ± 11.4 | 70.1 ± 8.5 |
| 4 | 36.6 ± 30.5 | 76.2 ± 6.5 | 96.0 ± 2.4 | 20.0 ± 12.6 | 54.1 ± 25.0 | 86.7 ± 4.2 | 0.0 ± 0.0 | 27.9 ± 15.3 | 70.8 ± 7.3 | 28.3 ± 23.6 | 65.1 ± 9.3 | 85.2 ± 7.0 | 26.9 ± 11.1 | 50.0 ± 21.7 | 86.4 ± 6.6 | 0.4 ± 0.2 | 17.9 ± 10.6 | 72.2 ± 7.0 |
| 5 | 41.7 ± 29.1 | 77.5 ± 5.9 | 96.7 ± 1.8 | 22.2 ± 11.7 | 58.7 ± 21.4 | 87.5 ± 3.5 | 0.0 ± 0.0 | 30.8 ± 13.6 | 72.2 ± 6.0 | 32.3 ± 22.5 | 67.0 ± 8.2 | 86.5 ± 6.3 | 29.0 ± 9.7 | 54.0 ± 18.4 | 87.6 ± 4.9 | 0.5 ± 0.1 | 19.9 ± 9.7 | 73.5 ± 6.2 |
| 6 | - | 78.4 ± 5.4 | 96.7 ± 1.4 | - | 61.9 ± 18.2 | 88.1 ± 3.2 | - | 32.8 ± 11.8 | 73.1 ± 4.9 | - | 68.4 ± 7.3 | 87.6 ± 5.7 | - | 56.8 ± 15.4 | 88.4 ± 3.9 | - | 21.5 ± 8.8 | 74.5 ± 5.7 |
| 7 | - | 79.2 ± 4.9 | 96.9 ± 1.1 | - | 64.1 ± 15.3 | 88.5 ± 2.9 | - | 34.3 ± 10.3 | 73.7 ± 4.1 | - | 69.4 ± 6.4 | 88.4 ± 5.3 | - | 58.8 ± 12.9 | 88.9 ± 3.2 | - | 22.7 ± 8.0 | 75.3 ± 5.4 |
| 8 | - | 79.8 ± 4.6 | 97.0 ± 0.9 | - | 65.7 ± 12.8 | 88.9 ± 2.7 | - | 35.5 ± 8.9 | 74.2 ± 3.5 | - | 70.2 ± 5.7 | 89.0 ± 4.9 | - | 60.1 ± 10.7 | 89.3 ± 2.8 | - | 23.7 ± 7.2 | 75.9 ± 5.1 |
| 9 | - | 80.3 ± 4.3 | 97.1 ± 0.7 | - | 66.8 ± 10.7 | 89.2 ± 2.5 | - | 36.3 ± 7.7 | 74.5 ± 2.9 | - | 70.8 ± 5.1 | 89.5 ± 4.5 | - | 61.1 ± 9.0 | 89.6 ± 2.4 | - | 24.4 ± 6.5 | 76.5 ± 4.9 |
| 10 | - | 80.7 ± 4.0 | 97.1 ± 0.6 | - | 67.6 ± 9.0 | 89.4 ± 2.4 | - | 37.0 ± 6.7 | 74.8 ± 2.5 | - | 71.3 ± 4.6 | 89.9 ± 4.2 | - | 61.8 ± 7.5 | 89.8 ± 2.1 | - | 25.1 ± 5.9 | 76.9 ± 4.7 |
| 11 | - | 81.1 ± 3.8 | 97.2 ± 0.6 | - | 68.2 ± 7.5 | 89.6 ± 2.2 | - | 37.5 ± 5.9 | 75.0 ± 2.1 | - | 71.7 ± 4.2 | 90.3 ± 3.9 | - | 62.4 ± 6.3 | 90.0 ± 1.9 | - | 25.6 ± 5.4 | 77.3 ± 4.6 |
| 12 | - | 81.4 ± 3.5 | 97.2 ± 0.5 | - | 68.7 ± 6.3 | 89.8 ± 2.1 | - | 37.9 ± 5.2 | 75.1 ± 1.8 | - | 72.0 ± 3.8 | 90.6 ± 3.6 | - | 62.8 ± 5.3 | 90.1 ± 1.6 | - | 26.0 ± 4.9 | 77.7 ± 4.4 |
| 13 | - | 81.6 ± 3.3 | 97.3 ± 0.4 | - | 69.0 ± 5.2 | 90.0 ± 2.0 | - | 38.2 ± 4.6 | 75.2 ± 1.6 | - | 72.3 ± 3.5 | 90.9 ± 3.3 | - | 63.0 ± 4.5 | 90.3 ± 1.5 | - | 26.4 ± 4.5 | 78.0 ± 4.3 |
| 14 | - | 81.9 ± 3.1 | 97.3 ± 0.4 | - | 69.2 ± 4.4 | 90.1 ± 1.9 | - | 38.5 ± 4.2 | 75.3 ± 1.4 | - | 72.6 ± 3.2 | 91.1 ± 3.1 | - | 63.3 ± 3.8 | 90.4 ± 1.3 | - | 26.7 ± 4.2 | 78.3 ± 4.1 |
| 15 | - | 82.1 ± 3.0 | 97.3 ± 0.4 | - | 69.4 ± 3.7 | 90.2 ± 1.8 | - | 38.8 ± 3.8 | 75.4 ± 1.2 | - | 72.8 ± 3.0 | 91.3 ± 2.9 | - | 63.5 ± 3.2 | 90.4 ± 1.2 | - | 27.0 ± 3.9 | 78.5 ± 4.0 |
| 16 | - | 82.2 ± 2.8 | 97.3 ± 0.3 | - | 69.5 ± 3.1 | 90.3 ± 1.7 | - | 39.0 ± 3.5 | 75.4 ± 1.1 | - | 72.9 ± 2.8 | 91.4 ± 2.7 | - | 63.6 ± 2.8 | 90.5 ± 1.1 | - | 27.2 ± 3.6 | 78.7 ± 3.9 |
| 17 | - | 82.4 ± 2.7 | - | - | 69.6 ± 2.6 | - | - | 39.1 ± 3.3 | - | - | 73.1 ± 2.7 | - | - | 63.7 ± 2.4 | - | - | 27.4 ± 3.4 | - |
| 18 | - | 82.5 ± 2.5 | - | - | 69.7 ± 2.2 | - | - | 39.3 ± 3.1 | - | - | 73.2 ± 2.5 | - | - | 63.8 ± 2.1 | - | - | 27.6 ± 3.2 | - |
| 19 | - | 82.7 ± 2.4 | - | - | 69.8 ± 1.9 | - | - | 39.4 ± 2.9 | - | - | 73.4 ± 2.4 | - | - | 63.9 ± 1.8 | - | - | 27.7 ± 3.0 | - |
| 20 | - | 82.8 ± 2.3 | - | - | 69.8 ± 1.6 | - | - | 39.6 ± 2.8 | - | - | 73.5 ± 2.3 | - | - | 64.0 ± 1.6 | - | - | 27.9 ± 2.9 | - |

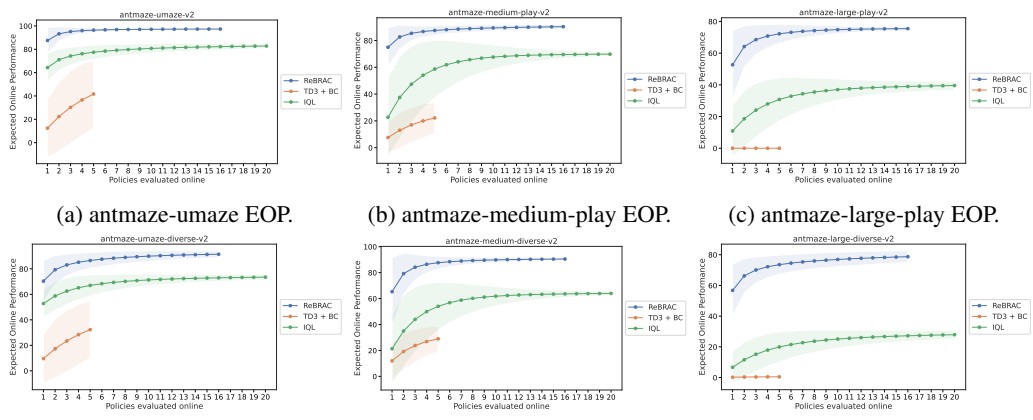

(a) antmaze-umaze EOP.    (b) antmaze-medium-play EOP.    (c) antmaze-large-play EOP.

(d) antmaze-umaze-diverse EOP.    (e) antmaze-medium-diverse EOP.    (f) antmaze-large-diverse EOP.

Figure 6: TD3+BC, IQL and ReBRAC visualised Expected Online Performance under uniform policy selection on AntMaze tasks.

Table 29: TD3+BC, IQL and ReBRAC Expected Online Performance under uniform policy selection on Pen tasks.

| Policies | human | | | cloned | | | expert | | |
|---|---|---|---|---|---|---|---|---|---|
| | TD3+BC | IQL | ReBRAC | TD3+BC | IQL | ReBRAC | TD3+BC | IQL | ReBRAC |
| 1 | 42.9 ± 32.0 | 87.1 ± 4.1 | 69.9 ± 28.2 | 33.6 ± 23.2 | 73.8 ± 5.8 | 65.8 ± 32.7 | 73.9 ± 65.2 | 130.1 ± 2.8 | 136.9 ± 25.4 |
| 2 | 60.4 ± 25.4 | 89.4 ± 2.9 | 85.8 ± 20.7 | 44.9 ± 25.4 | 76.7 ± 4.8 | 83.6 ± 21.8 | 108.9 ± 52.9 | 131.7 ± 2.1 | 149.0 ± 13.6 |
| 3 | 68.7 ± 18.9 | 90.3 ± 2.5 | 92.8 ± 14.8 | 52.6 ± 25.2 | 78.0 ± 4.9 | 90.8 ± 15.4 | 125.6 ± 38.7 | 132.4 ± 1.8 | 152.4 ± 7.0 |
| 4 | 73.0 ± 14.2 | 90.9 ± 2.2 | 96.3 ± 10.9 | 58.1 ± 24.1 | 79.0 ± 5.2 | 94.6 ± 12.2 | 134.1 ± 27.8 | 132.8 ± 1.6 | 153.5 ± 3.8 |
| 5 | 75.6 ± 11.2 | 91.4 ± 2.0 | 98.4 ± 8.3 | 62.3 ± 22.6 | 79.7 ± 5.4 | 97.0 ± 10.4 | 138.5 ± 20.0 | 133.1 ± 1.5 | 154.1 ± 2.2 |
| 6 | - | 91.7 ± 1.8 | 99.7 ± 6.5 | - | 80.3 ± 5.5 | 98.7 ± 9.3 | - | 133.4 ± 1.4 | 154.3 ± 1.5 |
| 7 | - | 92.0 ± 1.7 | 100.5 ± 5.2 | - | 80.9 ± 5.6 | 100.0 ± 8.4 | - | 133.6 ± 1.3 | 154.5 ± 1.1 |
| 8 | - | 92.2 ± 1.6 | 101.1 ± 4.4 | - | 81.4 ± 5.7 | 101.1 ± 7.6 | - | 133.7 ± 1.2 | 154.7 ± 0.9 |
| 9 | - | 92.3 ± 1.4 | 101.6 ± 3.7 | - | 81.9 ± 5.8 | 101.9 ± 7.0 | - | 133.9 ± 1.1 | 154.8 ± 0.8 |
| 10 | - | 92.5 ± 1.3 | 101.9 ± 3.3 | - | 82.3 ± 5.9 | 102.6 ± 6.4 | - | 134.0 ± 1.0 | 154.8 ± 0.7 |
| 11 | - | 92.6 ± 1.2 | 102.2 ± 2.9 | - | 82.7 ± 5.9 | 103.1 ± 5.9 | - | 134.1 ± 0.9 | 154.9 ± 0.6 |
| 12 | - | 92.7 ± 1.2 | 102.4 ± 2.6 | - | 83.0 ± 5.9 | 103.6 ± 5.4 | - | 134.1 ± 0.9 | 154.9 ± 0.6 |
| 13 | - | 92.8 ± 1.1 | 102.6 ± 2.4 | - | 83.4 ± 5.9 | 104.0 ± 5.0 | - | 134.2 ± 0.8 | 155.0 ± 0.5 |
| 14 | - | 92.9 ± 1.0 | 102.8 ± 2.2 | - | 83.7 ± 5.9 | 104.4 ± 4.6 | - | 134.2 ± 0.7 | 155.0 ± 0.5 |
| 15 | - | 92.9 ± 1.0 | 102.9 ± 2.0 | - | 84.0 ± 5.8 | 104.7 ± 4.3 | - | 134.3 ± 0.7 | 155.0 ± 0.4 |
| 16 | - | 93.0 ± 0.9 | 103.0 ± 1.8 | - | 84.3 ± 5.8 | 104.9 ± 4.0 | - | 134.3 ± 0.6 | 155.1 ± 0.4 |
| 17 | - | 93.0 ± 0.9 | 103.1 ± 1.7 | - | 84.6 ± 5.7 | 105.1 ± 3.8 | - | 134.4 ± 0.6 | 155.1 ± 0.4 |
| 18 | - | 93.1 ± 0.8 | 103.2 ± 1.6 | - | 84.8 ± 5.7 | 105.3 ± 3.5 | - | 134.4 ± 0.6 | 155.1 ± 0.4 |
| 19 | - | 93.1 ± 0.8 | 103.3 ± 1.4 | - | 85.0 ± 5.6 | 105.5 ± 3.3 | - | 134.4 ± 0.5 | 155.1 ± 0.4 |
| 20 | - | 93.2 ± 0.8 | 103.3 ± 1.3 | - | 85.3 ± 5.5 | 105.7 ± 3.1 | - | 134.5 ± 0.5 | 155.1 ± 0.4 |

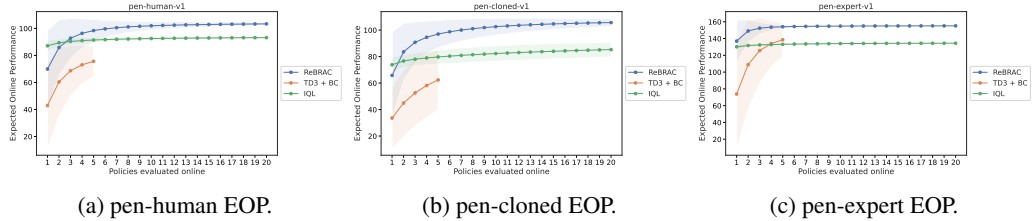

(a) pen-human EOP.      (b) pen-cloned EOP.      (c) pen-expert EOP.

Figure 7: TD3+BC, IQL and ReBRAC visualised Expected Online Performance under uniform policy selection on Pen tasks.

Table 30: TD3+BC, IQL and ReBRAC Expected Online Performance under uniform policy selection on Door tasks.

| | human | | | cloned | | | expert | | |
|---|---|---|---|---|---|---|---|---|---|
| Policies | TD3+BC | IQL | ReBRAC | TD3+BC | IQL | ReBRAC | TD3+BC | IQL | ReBRAC |
| 1 | -0.2 ± 0.1 | **4.4 ± 1.2** | -0.1 ± 0.1 | -0.1 ± 0.3 | **1.6 ± 0.8** | 0.3 ± 0.9 | 50.7 ± 46.3 | **102.1 ± 5.7** | 75.4 ± 43.0 |
| 2 | -0.1 ± 0.1 | **5.0 ± 1.0** | -0.0 ± 0.1 | 0.0 ± 0.3 | **2.0 ± 0.6** | 0.6 ± 1.2 | 75.6 ± 39.5 | **104.7 ± 2.5** | 96.1 ± 25.0 |
| 3 | -0.1 ± 0.1 | **5.4 ± 1.0** | -0.0 ± 0.0 | 0.1 ± 0.3 | **2.2 ± 0.5** | 0.8 ± 1.4 | 88.2 ± 30.4 | **105.4 ± 1.3** | 102.3 ± 13.5 |
| 4 | -0.1 ± 0.1 | **5.6 ± 0.9** | -0.0 ± 0.0 | 0.2 ± 0.3 | **2.4 ± 0.4** | 1.0 ± 1.6 | 94.9 ± 22.9 | **105.7 ± 0.8** | 104.4 ± 7.3 |
| 5 | -0.1 ± 0.1 | **5.8 ± 0.9** | -0.0 ± 0.0 | 0.2 ± 0.3 | **2.5 ± 0.4** | 1.2 ± 1.7 | 98.6 ± 17.2 | **105.8 ± 0.6** | 105.2 ± 4.1 |
| 6 | - | **5.9 ± 0.9** | 0.0 ± 0.0 | - | **2.5 ± 0.3** | 1.3 ± 1.8 | - | **105.9 ± 0.5** | 105.6 ± 2.4 |
| 7 | - | **6.0 ± 0.9** | 0.0 ± 0.0 | - | **2.6 ± 0.3** | 1.5 ± 1.8 | - | **106.0 ± 0.5** | 105.8 ± 1.5 |
| 8 | - | **6.1 ± 0.8** | 0.0 ± 0.0 | - | **2.6 ± 0.3** | 1.6 ± 1.9 | - | **106.1 ± 0.4** | 105.9 ± 1.0 |
| 9 | - | **6.2 ± 0.8** | 0.0 ± 0.0 | - | **2.6 ± 0.2** | 1.8 ± 1.9 | - | **106.1 ± 0.4** | 105.9 ± 0.7 |
| 10 | - | **6.3 ± 0.8** | 0.0 ± 0.0 | - | **2.7 ± 0.2** | 1.9 ± 1.9 | - | **106.1 ± 0.4** | 106.0 ± 0.5 |
| 11 | - | **6.4 ± 0.8** | 0.0 ± 0.0 | - | **2.7 ± 0.2** | 2.0 ± 2.0 | - | **106.2 ± 0.4** | 106.0 ± 0.4 |
| 12 | - | **6.4 ± 0.7** | 0.0 ± 0.0 | - | **2.7 ± 0.2** | 2.2 ± 2.0 | - | **106.2 ± 0.3** | 106.0 ± 0.3 |
| 13 | - | **6.5 ± 0.7** | 0.0 ± 0.0 | - | **2.7 ± 0.2** | 2.3 ± 2.0 | - | **106.2 ± 0.3** | 106.0 ± 0.2 |
| 14 | - | **6.5 ± 0.7** | 0.0 ± 0.0 | - | **2.7 ± 0.2** | 2.4 ± 2.0 | - | **106.2 ± 0.3** | 106.0 ± 0.2 |
| 15 | - | **6.6 ± 0.7** | 0.0 ± 0.0 | - | **2.7 ± 0.2** | 2.5 ± 2.0 | - | **106.3 ± 0.3** | 106.0 ± 0.2 |
| 16 | - | **6.6 ± 0.6** | 0.0 ± 0.0 | - | **2.7 ± 0.2** | 2.6 ± 1.9 | - | **106.3 ± 0.3** | 106.1 ± 0.2 |
| 17 | - | **6.6 ± 0.6** | 0.0 ± 0.0 | - | **2.8 ± 0.2** | 2.7 ± 1.9 | - | **106.3 ± 0.3** | 106.1 ± 0.2 |
| 18 | - | **6.7 ± 0.6** | 0.0 ± 0.0 | - | **2.8 ± 0.2** | 2.7 ± 1.9 | - | **106.3 ± 0.3** | 106.1 ± 0.1 |
| 19 | - | **6.7 ± 0.6** | 0.0 ± 0.0 | - | 2.8 ± 0.2 | **2.8 ± 1.9** | - | **106.3 ± 0.3** | 106.1 ± 0.1 |
| 20 | - | **6.7 ± 0.5** | 0.0 ± 0.0 | - | 2.8 ± 0.2 | **2.9 ± 1.9** | - | **106.3 ± 0.2** | 106.1 ± 0.1 |

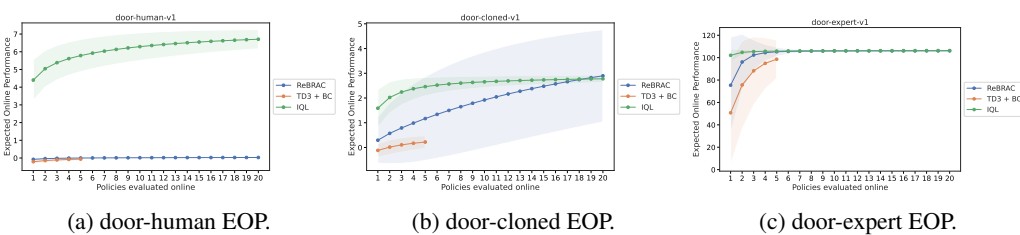

(a) door-human EOP.      (b) door-cloned EOP.      (c) door-expert EOP.

Figure 8: TD3+BC, IQL and ReBRAC visualised Expected Online Performance under uniform policy selection on Door tasks.

Table 31: TD3+BC, IQL and ReBRAC Expected Online Performance under uniform policy selection on Hammer tasks.

| | human | | | cloned | | | expert | | |
|---|---|---|---|---|---|---|---|---|---|
| Policies | TD3+BC | IQL | ReBRAC | TD3+BC | IQL | ReBRAC | TD3+BC | IQL | ReBRAC |
| 1 | $0.2 \pm 0.0$ | $1.6 \pm 0.4$ | $0.3 \pm 0.2$ | $1.1 \pm 0.8$ | $1.9 \pm 1.1$ | $4.5 \pm 5.5$ | $60.2 \pm 55.4$ | $128.7 \pm 1.1$ | $101.9 \pm 49.9$ |
| 2 | $0.3 \pm 0.0$ | $1.8 \pm 0.4$ | $0.3 \pm 0.2$ | $1.6 \pm 0.8$ | $2.5 \pm 1.1$ | $7.3 \pm 6.1$ | $90.0 \pm 48.4$ | $129.3 \pm 0.9$ | $124.3 \pm 26.8$ |
| 3 | $0.3 \pm 0.0$ | $1.9 \pm 0.4$ | $0.4 \pm 0.2$ | $1.8 \pm 0.7$ | $2.8 \pm 1.0$ | $9.2 \pm 6.2$ | $105.7 \pm 38.1$ | $129.6 \pm 0.7$ | $130.1 \pm 13.6$ |
| 4 | $0.3 \pm 0.0$ | $2.0 \pm 0.4$ | $0.4 \pm 0.2$ | $2.0 \pm 0.6$ | $3.1 \pm 0.9$ | $10.6 \pm 6.2$ | $114.3 \pm 29.3$ | $129.7 \pm 0.5$ | $131.9 \pm 7.0$ |
| 5 | $0.3 \pm 0.0$ | $2.1 \pm 0.4$ | $0.5 \pm 0.2$ | $2.1 \pm 0.6$ | $3.3 \pm 0.9$ | $11.7 \pm 6.2$ | $119.2 \pm 22.4$ | $129.8 \pm 0.4$ | $132.6 \pm 3.8$ |
| 6 | - | $2.1 \pm 0.4$ | $0.5 \pm 0.2$ | - | $3.4 \pm 0.8$ | $12.6 \pm 6.1$ | - | $129.9 \pm 0.3$ | $133.0 \pm 2.4$ |
| 7 | - | $2.2 \pm 0.4$ | $0.5 \pm 0.2$ | - | $3.5 \pm 0.7$ | $13.3 \pm 6.0$ | - | $129.9 \pm 0.2$ | $133.2 \pm 1.7$ |
| 8 | - | $2.2 \pm 0.4$ | $0.5 \pm 0.2$ | - | $3.6 \pm 0.7$ | $14.0 \pm 5.9$ | - | $129.9 \pm 0.2$ | $133.4 \pm 1.4$ |
| 9 | - | $2.3 \pm 0.3$ | $0.6 \pm 0.2$ | - | $3.7 \pm 0.6$ | $14.6 \pm 5.8$ | - | $130.0 \pm 0.1$ | $133.5 \pm 1.2$ |
| 10 | - | $2.3 \pm 0.3$ | $0.6 \pm 0.2$ | - | $3.7 \pm 0.6$ | $15.1 \pm 5.7$ | - | $130.0 \pm 0.1$ | $133.6 \pm 1.1$ |
| 11 | - | $2.3 \pm 0.3$ | $0.6 \pm 0.1$ | - | $3.8 \pm 0.5$ | $15.5 \pm 5.6$ | - | $130.0 \pm 0.1$ | $133.7 \pm 1.0$ |
| 12 | - | $2.3 \pm 0.3$ | $0.6 \pm 0.1$ | - | $3.8 \pm 0.5$ | $16.0 \pm 5.4$ | - | $130.0 \pm 0.1$ | $133.8 \pm 0.9$ |
| 13 | - | $2.4 \pm 0.3$ | $0.6 \pm 0.1$ | - | $3.9 \pm 0.5$ | $16.3 \pm 5.3$ | - | $130.0 \pm 0.1$ | $133.9 \pm 0.8$ |
| 14 | - | $2.4 \pm 0.3$ | $0.6 \pm 0.1$ | - | $3.9 \pm 0.4$ | $16.7 \pm 5.2$ | - | $130.0 \pm 0.1$ | $133.9 \pm 0.7$ |
| 15 | - | $2.4 \pm 0.3$ | $0.6 \pm 0.1$ | - | $3.9 \pm 0.4$ | $17.0 \pm 5.1$ | - | $130.0 \pm 0.1$ | $134.0 \pm 0.6$ |
| 16 | - | $2.4 \pm 0.3$ | $0.6 \pm 0.1$ | - | $4.0 \pm 0.4$ | $17.3 \pm 4.9$ | - | $130.0 \pm 0.1$ | $134.0 \pm 0.6$ |
| 17 | - | $2.4 \pm 0.2$ | $0.6 \pm 0.1$ | - | $4.0 \pm 0.4$ | $17.5 \pm 4.8$ | - | $130.0 \pm 0.0$ | $134.0 \pm 0.5$ |
| 18 | - | $2.4 \pm 0.2$ | $0.6 \pm 0.1$ | - | $4.0 \pm 0.3$ | $17.8 \pm 4.7$ | - | $130.0 \pm 0.0$ | $134.0 \pm 0.5$ |
| 19 | - | $2.5 \pm 0.2$ | $0.6 \pm 0.1$ | - | $4.0 \pm 0.3$ | $18.0 \pm 4.6$ | - | $130.0 \pm 0.0$ | $134.1 \pm 0.4$ |
| 20 | - | $2.5 \pm 0.2$ | $0.7 \pm 0.1$ | - | $4.0 \pm 0.3$ | $18.2 \pm 4.5$ | - | $130.0 \pm 0.0$ | $134.1 \pm 0.4$ |

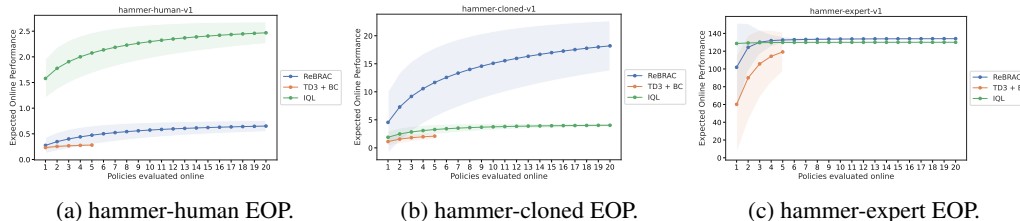

(a) hammer-human EOP.   (b) hammer-cloned EOP.   (c) hammer-expert EOP.

Figure 9: TD3+BC, IQL and ReBRAC visualised Expected Online Performance under uniform policy selection on Hammer tasks.

Table 32: TD3+BC, IQL and ReBRAC Expected Online Performance under uniform policy selection on tasks.

| | human | | | cloned | | | expert | | |
|---|---|---|---|---|---|---|---|---|---|
| Policies | TD3+BC | IQL | ReBRAC | TD3+BC | IQL | ReBRAC | TD3+BC | IQL | ReBRAC |
| 1 | $-0.2 \pm 0.1$ | $0.2 \pm 0.2$ | $-0.1 \pm 0.1$ | $-0.2 \pm 0.1$ | $-0.0 \pm 0.1$ | $0.5 \pm 0.8$ | $21.4 \pm 43.2$ | $106.0 \pm 1.4$ | $73.5 \pm 44.3$ |
| 2 | $-0.2 \pm 0.1$ | $0.2 \pm 0.2$ | $-0.1 \pm 0.1$ | $-0.2 \pm 0.1$ | $0.0 \pm 0.1$ | $0.9 \pm 0.9$ | $38.8 \pm 51.9$ | $106.8 \pm 1.0$ | $96.0 \pm 27.4$ |
| 3 | $-0.1 \pm 0.1$ | $0.3 \pm 0.2$ | $-0.0 \pm 0.0$ | $-0.2 \pm 0.1$ | $0.1 \pm 0.1$ | $1.2 \pm 0.9$ | $52.6 \pm 54.0$ | $107.2 \pm 0.8$ | $103.7 \pm 15.9$ |
| 4 | $-0.1 \pm 0.1$ | $0.3 \pm 0.2$ | $-0.0 \pm 0.0$ | $-0.1 \pm 0.1$ | $0.1 \pm 0.1$ | $1.4 \pm 0.9$ | $63.7 \pm 53.1$ | $107.4 \pm 0.7$ | $106.7 \pm 9.4$ |
| 5 | $-0.1 \pm 0.1$ | $0.4 \pm 0.2$ | $-0.0 \pm 0.0$ | $-0.1 \pm 0.0$ | $0.1 \pm 0.1$ | $1.6 \pm 0.9$ | $72.5 \pm 50.7$ | $107.5 \pm 0.6$ | $107.9 \pm 5.8$ |
| 6 | - | $0.4 \pm 0.2$ | $-0.0 \pm 0.0$ | - | $0.1 \pm 0.1$ | $1.7 \pm 0.8$ | - | $107.6 \pm 0.5$ | $108.6 \pm 3.8$ |
| 7 | - | $0.4 \pm 0.2$ | $-0.0 \pm 0.0$ | - | $0.1 \pm 0.1$ | $1.8 \pm 0.8$ | - | $107.7 \pm 0.5$ | $108.9 \pm 2.6$ |
| 8 | - | $0.4 \pm 0.2$ | $0.0 \pm 0.0$ | - | $0.1 \pm 0.1$ | $1.9 \pm 0.8$ | - | $107.7 \pm 0.5$ | $109.2 \pm 1.9$ |
| 9 | - | $0.5 \pm 0.2$ | $0.0 \pm 0.0$ | - | $0.1 \pm 0.1$ | $2.0 \pm 0.7$ | - | $107.8 \pm 0.4$ | $109.3 \pm 1.5$ |
| 10 | - | $0.5 \pm 0.2$ | $0.0 \pm 0.0$ | - | $0.1 \pm 0.1$ | $2.1 \pm 0.7$ | - | $107.8 \pm 0.4$ | $109.4 \pm 1.3$ |
| 11 | - | $0.5 \pm 0.2$ | $0.0 \pm 0.0$ | - | $0.2 \pm 0.1$ | $2.1 \pm 0.7$ | - | $107.8 \pm 0.4$ | $109.5 \pm 1.1$ |
| 12 | - | $0.5 \pm 0.2$ | $0.0 \pm 0.0$ | - | $0.2 \pm 0.1$ | $2.2 \pm 0.6$ | - | $107.9 \pm 0.4$ | $109.6 \pm 1.0$ |
| 13 | - | $0.5 \pm 0.2$ | $0.0 \pm 0.0$ | - | $0.2 \pm 0.1$ | $2.2 \pm 0.6$ | - | $107.9 \pm 0.4$ | $109.7 \pm 0.9$ |
| 14 | - | $0.5 \pm 0.2$ | $0.0 \pm 0.0$ | - | $0.2 \pm 0.1$ | $2.3 \pm 0.6$ | - | $107.9 \pm 0.4$ | $109.8 \pm 0.9$ |
| 15 | - | $0.5 \pm 0.2$ | $0.0 \pm 0.0$ | - | $0.2 \pm 0.1$ | $2.3 \pm 0.6$ | - | $107.9 \pm 0.4$ | $109.8 \pm 0.8$ |
| 16 | - | $0.5 \pm 0.2$ | $0.0 \pm 0.0$ | - | $0.2 \pm 0.1$ | $2.3 \pm 0.5$ | - | $108.0 \pm 0.4$ | $109.9 \pm 0.8$ |
| 17 | - | $0.6 \pm 0.2$ | $0.0 \pm 0.0$ | - | $0.2 \pm 0.1$ | $2.4 \pm 0.5$ | - | $108.0 \pm 0.4$ | $109.9 \pm 0.8$ |
| 18 | - | $0.6 \pm 0.2$ | $0.0 \pm 0.0$ | - | $0.2 \pm 0.1$ | $2.4 \pm 0.5$ | - | $108.0 \pm 0.3$ | $109.9 \pm 0.8$ |
| 19 | - | $0.6 \pm 0.2$ | $0.0 \pm 0.0$ | - | $0.2 \pm 0.1$ | $2.4 \pm 0.5$ | - | $108.0 \pm 0.3$ | $110.0 \pm 0.7$ |
| 20 | - | $0.6 \pm 0.1$ | $0.0 \pm 0.0$ | - | $0.2 \pm 0.1$ | $2.5 \pm 0.4$ | - | $108.0 \pm 0.3$ | $110.0 \pm 0.7$ |

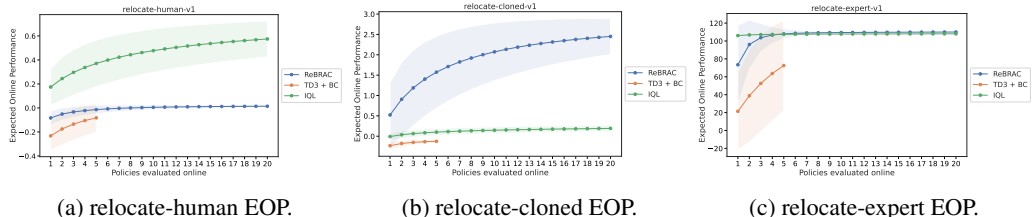

| (a) relocate-human EOP. | (b) relocate-cloned EOP. | (c) relocate-expert EOP. |

Figure 10: TD3+BC, IQL and ReBRAC visualised Expected Online Performance under uniform policy selection on Relocate tasks.

# G  D4RL tasks ablation

Table 33: ReBRAC ablations for halfcheetah tasks. We report final normalized score averaged over 4 unseen training seeds.

| Ablation | random | medium | expert | medium-expert | medium-replay | full-replay | Average |
|---|---|---|---|---|---|---|---|
| TD3+BC, paper | 11.0 ± 1.1 | 48.3 ± 0.3 | 96.7 ± 1.1 | 90.7 ± 4.3 | 44.6 ± 0.5 | - | - |
| TD3+BC, our | 2.2 ± 0.0 | 44.6 ± 0.4 | 93.8 ± 0.1 | 91.9 ± 2.3 | 40.5 ± 1.6 | 69.3 ± 0.7 | 57.0 |
| TD3+BC, tuned | 30.1 ± 1.4 (+0.7%) | 55.4 ± 0.9 (-15.2%) | 95.5 ± 0.5 (-9.6%) | 91.9 ± 2.3 (-11.4%) | 45.1 ± 1.7 (-11.0%) | 74.1 ± 2.9 (-9.5%) | 65.3 (-10.3%) |
| ReBRAC w/o LN | 32.0 ± 1.5 (+7.0%) | 64.3 ± 4.1 (-1.5%) | 61.7 ± 20.4 (-41.5%) | 86.7 ± 0.9 (-16.3%) | 52.8 ± 2.4 (+4.1%) | 82.1 ± 2.1 (+0.2%) | 63.2 (-13.1%) |
| ReBRAC w/o layer | 27.8 ± 3.4 (-7.0%) | 65.0 ± 1.6 (-0.4%) | 74.4 ± 26.7 (-29.5%) | 86.7 ± 8.8 (-16.3%) | 50.4 ± 0.7 (-0.5%) | 80.9 ± 1.1 (-1.2%) | 64.1 (-11.9%) |
| ReBRAC w/o actor penalty | 31.8 ± 4.1 (+6.3%) | 64.5 ± 0.7 (-1.2%) | 4.3 ± 4.3 (-95.9%) | 71.6 ± 12.3 (-30.9%) | 38.0 ± 27.2 (-25.0%) | 59.3 ± 41.3 (-27.5%) | 44.9 (-38.3%) |
| ReBRAC w/o critic penalty | 28.1 ± 1.6 (-6.0%) | 65.7 ± 1.4 (+0.6%) | 104.2 ± 5.9 (-1.3%) | 100.5 ± 3.1 (-3.0%) | 50.7 ± 0.1 (0.0%) | 81.7 ± 1.2 (-0.2%) | 71.8 (-1.3%) |
| ReBRAC w/o large batch | 21.0 ± 15.7 (-29.7%) | 65.8 ± 0.7 (+0.7%) | 62.6 ± 24.3 (-40.7%) | 85.2 ± 7.3 (-17.8%) | 50.7 ± 1.1 (0.0%) | 81.9 ± 1.4 (0.0%) | 61.2 (-15.9%) |
| ReBRAC | 29.9 ± 1.2 | 65.3 ± 1.1 | 105.6 ± 1.5 | 103.7 ± 3.9 | 50.7 ± 0.6 | 81.9 ± 1.4 | 72.8 |

Table 34: ReBRAC ablations for hopper tasks. We report final normalized score averaged over 4 unseen training seeds.

| Ablation | random | medium | expert | medium-expert | medium-replay | full-replay | Average |
|---|---|---|---|---|---|---|---|
| TD3+BC, paper | 8.5 ± 0.6 | 59.3 ± 4.2 | 107.8 ± 7.0 | 98.0 ± 9.4 | 60.9 ± 18.8 | - | - |
| TD3+BC, our | 10.3 ± 1.8 | 53.2 ± 2.2 | 108.7 ± 5.3 | 75.8 ± 8.9 | 64.5 ± 24.9 | 49.9 ± 9.6 | 60.4 |
| TD3+BC, tuned | 10.3 ± 1.8 (+51.5%) | 57.6 ± 6.8 (-43.2%) | 110.7 ± 2.1 (+21.1%) | 106.2 ± 2.5 (-3.4%) | 64.5 ± 24.9 (-30.8%) | 106.2 ± 2.2 (-0.6%) | 75.9 (-10.6%) |
| ReBRAC w/o LN | 12.2 ± 13.3 (+79.4%) | 1.0 ± 0.6 (-99.0%) | 111.1 ± 1.0 (+21.6%) | 112.4 ± 0.7 (+2.3%) | 57.4 ± 25.0 (-38.5%) | 107.2 ± 2.0 (+0.4%) | 66.8 (-21.3%) |
| ReBRAC w/o layer | 8.8 ± 0.6 (+29.4%) | 101.8 ± 0.2 (+0.4%) | 103.7 ± 5.1 (+13.5%) | 104.1 ± 7.7 (-5.3%) | 97.5 ± 3.5 (+4.5%) | 106.5 ± 0.3 (-0.3%) | 87.0 (+2.5%) |
| ReBRAC w/o actor penalty | 7.5 ± 4.6 (+10.3%) | 1.7 ± 2.1 (-98.3%) | 1.1 ± 0.5 (-98.8%) | 1.6 ± 1.6 (-98.5%) | 24.4 ± 8.7 (-73.8%) | 27.7 ± 23.4 (-74.1%) | 10.6 (-87.5%) |
| ReBRAC w/o critic penalty | 7.4 ± 1.1 (+8.8%) | 102.3 ± 0.5 (+0.9%) | 103.4 ± 8.6 (+13.1%) | 111.2 ± 0.7 (+1.2%) | 83.1 ± 30.9 (-10.9%) | 107.5 ± 0.1 (+0.7%) | 85.8 (+1.1%) |
| ReBRAC w/o large batch | 8.6 ± 0.5 (+26.5%) | 98.9 ± 5.2 (-2.5%) | 98.8 ± 13.4 (+8.1%) | 107.8 ± 2.9 (-1.9%) | 96.2 ± 7.6 (+3.1%) | 106.6 ± 0.2 (-0.2%) | 86.1 (+1.4%) |
| ReBRAC | 6.8 ± 3.4 | 101.4 ± 1.5 | 91.4 ± 4.7 | 109.9 ± 3.0 | 93.3 ± 7.5 | 106.8±0.6 | 84.9 |

Table 35: ReBRAC ablations for walker2d tasks. We report final normalized score averaged over 4 unseen training seeds.

| Ablation | random | medium | expert | medium-expert | medium-replay | full-replay | Average |
|---|---|---|---|---|---|---|---|
| TD3+BC, paper | 1.6 ± 1.7 | 83.7 ± 2.1 | 110.2 ± 0.3 | 110.1 ± 0.5 | 81.8 ± 5.5 | - | - |
| TD3+BC, our | 4.5 ± 2.2 | 77.1 ± 1.9 | 109.1 ± 0.5 | 108.9 ± 0.3 | 50.9 ± 13.7 | 86.7 ± 5.1 | 72.8 |
| TD3+BC, tuned | 4.5 ± 2.2 (-78.8%) | 77.1 ± 1.9 (-5.5%) | 110.1 ± 0.1 (-2.0%) | 110.2 ± 0.7 (-1.3%) | 58.8 ± 28.5 (-22.2%) | 89.4 ± 8.2 (-13.0%) | 75.0 (-10.9%) |
| ReBRAC w/o LN | 1.3 ± 1.5 (-93.9%) | 84.3 ± 2.5 (+3.3%) | 8.3 ± 3.1 (-92.6%) | 52.7 ± 53.9 (-52.8%) | 78.9 ± 8.4 (+4.4%) | 61.1 ± 46.6 (-40.6%) | 47.7 (-43.3%) |
| ReBRAC w/o layer | 11.4 ± 11.9 (-46.5%) | 86.2 ± 0.9 (+5.6%) | 112.1 ± 0.1 (-0.2%) | 111.9 ± 0.2 (+0.2%) | 83.9 ± 5.4 (+11.0%) | 101.8 ± 0.9 (-1.0%) | 84.5 (+0.4%) |
| ReBRAC w/o actor penalty | 1.1 ± 0.9 (-94.8%) | 1.7 ± 2.1 (-97.9%) | 0.9 ± 1.1 (-99.2%) | 0.8 ± 1.3 (-99.3%) | 8.9 ± 5.7 (-88.2%) | 64.2 ± 29.5 (-37.5%) | 12.9 (-84.7%) |
| ReBRAC w/o critic penalty | 19.8 ± 3.6 (-7.0%) | 81.6 ± 9.2 (0.0%) | 112.0 ± 0.1 (-0.3%) | 111.6 ± 0.3 (-0.1%) | 87.0 ± 4.5 (+15.1%) | 103.5 ± 1.5 (+0.7%) | 85.9 (+2.0%) |
| ReBRAC w/o large batch | 5.6 ± 0.2 (-73.7%) | 84.8 ± 1.0 (+3.9%) | 112.2 ± 0.2 (-0.1%) | 110.9 ± 0.2 (-0.7%) | 71.7 ± 20.2 (-5.2%) | 97.7 ± 5.8 (-5.0%) | 80.4 (-4.5%) |
| ReBRAC | 21.3 ± 0.8 | 81.6 ± 3.9 | 112.3 ± 0.0 | 111.7 ± 0.3 | 75.6±10.3 | 102.8±0.9 | 84.2 |

Table 36: ReBRAC ablations for AntMaze tasks. We report final normalized score averaged over 4 unseen training seeds.

| Ablation | umaze | umaze-diverse | medium-play | medium-diverse | large-play | large-diverse | Average |
|---|---|---|---|---|---|---|---|
| TD3+BC, paper | 78.6 | 71.4 | 10.6 | 3.0 | 0.2 | 0.0 | 27.3 |
| TD3+BC, our | 62.0 ± 2.4 | 48.0 ± 11.6 | 0.0 ± 0.0 | 0.5 ± 1 | 0.0 ± 0.0 | 0.5 ± 0.5 | 18.5 |
| TD3+BC, tuned | 62.0 ± 2.4 (-36.8%) | 48.0 ± 11.6 (-42.9%) | 39.0 ± 21.7 (-54.8%) | 18.5 ± 17.7 (-72.1%) | 0.2 ± 0.5 (-99.6%) | 0.0 ± 1.0 (-100.0%) | 27.9 (-62.9%) |
| ReBRAC w/o γ change | 0.0 ± 0.0 (-100.0%) | 90.7 ± 3.2 (+7.7%) | 1.0 ± 0.0 (-98.8%) | 0.2 ± 0.5 (-99.7%) | 19.3 ± 18.5 (-58.0%) | 15.0 ± 8.0 (-79.0%) | 21.0 (-72.1%) |
| ReBRAC w/o LN | 0.0 ± 0.0 (-100.0%) | 0.0 ± 0.0 (-100.0%) | 0.0 ± 0.0 (-100.0%) | 0.0 ± 0.0 (-100.0%) | 0.0 ± 0.0 (-100.0%) | 0.0 ± 0.0 (-100.0%) | 0.0 (-100%) |
| ReBRAC w/o layer | 31.0 ± 45.4 (-68.4%) | 61.7 ± 25.3 (-26.7%) | 0.0 ± 0.0 (-100.0%) | 16.0 ± 32.0 (-75.8%) | 0.0 ± 0.0 (-100.0%) | 0.0 ± 0.0 (-100.0%) | 18.1 (-76.0%) |
| ReBRAC w/o actor penalty | 1.0 ± 1.1 (-99.0%) | 0.0 ± 0.0 (-100.0%) | 0.0 ± 0.0 (-100.0%) | 0.0 ± 0.0 (-100.0%) | 0.0 ± 0.0 (-100.0%) | 0.0 ± 0.0 (-100.0%) | 0.1 (-99.9%) |
| ReBRAC w/o critic penalty | 98.2 ± 1.5 (0.0%) | 78.0 ± 26.3 (-7.4%) | 86.2 ± 2.6 (0.0%) | 57.5 ± 24.2 (-13.1%) | 56.7 ± 32.9 (+23.3%) | 57.0 ± 16.4 (-20.3%) | 72.2 (-4.1%) |
| ReBRAC w large batch | 60.7 ± 31.3 (-38.2%) | 68.5 ± 17.9 (-18.6%) | 43.9 ± 49.9 (-49.1%) | 34.0 ± 40.6 (-48.6%) | 39.2 ± 45.9 (-14.8%) | 0.0 ± 0.0 (-100.0%) | 41.0 (-45.6%) |
| ReBRAC | 98.2 ± 0.9 | 84.2 ± 18.5 | 86.2 ± 4.7 | 66.2 ± 16.3 | 46.0 ± 40.0 | 71.5 ± 12.3 | 75.3 |

Table 37: ReBRAC ablations for pen tasks. We report final normalized score averaged over 4 unseen training seeds.

| Ablation | human | cloned | expert | Average |
|---|---|---|---|---|
| TD3+BC, paper | 0.0 | 0.0 | 0.3 | 0.0 |
| TD3+BC, our | 65.9 ± 24.6 | 78.1 ± 5.7 | 144.9 ± 7.5 | 96.3 |
| TD3+BC, tuned | 77.6 ± 18.5 (-23.9%) | 78.1 ± 5.7 (-8.5%) | 144.9 ± 7.5 (-10.3%) | 100.2 (-12.5%) |
| ReBRAC w/o LN | 78.6 ± 14.8 (-22.9%) | 21.3 ± 11.0 (-75.1%) | 86.7 ± 59.8 (-44.6%) | 62.1 (-45.8%) |
| ReBRAC w/o layer | 89.1 ± 14.7 (-12.6%) | 106.7 ± 13.9 (+24.9%) | 147.2 ± 5.7 (-6.0%) | 114.3 (-0.3%) |
| ReBRAC w/o actor penalty | -0.5 ± 1.3 (-100.5%) | 0.6 ± 1.6 (-99.3%) | 0.0 ± 3.6 (-100.0%) | 0.0 (-100.0%) |
| ReBRAC w/o critic penalty | 99.9 ± 6.1 (-2.1%) | 75.0 ± 16.7 (-12.2%) | 154.6 ± 1.8 (-1.3%) | 109.8 (-4.2%) |
| ReBRAC w large batch | 67.2 ± 9.0 (-34.1%) | 83.2 ± 23.3 (-2.6%) | 155.0 ± 6.8 (-1.0%) | 101.8 (-11.2%) |
| ReBRAC | 102.0 ± 10.8 | 85.4 ± 24.2 | 156.6 ± 1.4 | 114.6 |

Table 38: ReBRAC ablations for door tasks. We report final normalized score averaged over 4 unseen training seeds.

| Ablation | human | cloned | expert | Average |
|---|---|---|---|---|
| TD3+BC, paper | 0.0 | 0.0 | 0.0 | 0.0 |
| TD3+BC, our | 0.0 ± 0.1 | 0.4 ± 1.0 | 102.5 ± 2.9 | 34.3 |
| TD3+BC, tuned | 0.0 ± 0.1 (-) | 0.4 ± 1.0 (+100.0%) | 105.8 ± 0.3 (+0.8%) | 35.4 (+1.1%) |
| ReBRAC w/o LN | -0.1 ± 0.0 (-) | -0.3 ± 0.0 (-250.0%) | 106.0 ± 0.8 (+1.0%) | 35.1 (+0.3%) |
| ReBRAC w/o layer | 0.0 ± 0.0 (-) | 0.1 ± 0.5 (-50.0%) | 104.4 ± 2.3 (-0.5%) | 34.8 (-0.6%) |
| ReBRAC w/o actor penalty | -0.1 ± 0.1 (-) | 0.0 ± 0.0 (-100.0%) | 0.0 ± 0.2 (-100.0%) | 0.0 (-100.0%) |
| ReBRAC w/o critic penalty | 0.0 ± 0.0 (-) | 0.1 ± 0.0 (-50.0%) | 106.1 ± 0.3 (+1.1%) | 35.4 (+1.1%) |
| ReBRAC w large batch | -0.1 ± 0.1 (-) | 0.1 ± 0.3 (-50.0%) | 106.1 ± 0.1 (+1.1%) | 35.3 (+0.9%) |
| ReBRAC | 0.0 ± 0.0 | 0.2 ± 0.3 | 104.9 ± 2.2 | 35.0 |

Table 39: ReBRAC ablations for hammer tasks. We report final normalized score averaged over 4 unseen training seeds.

| Ablation | human | cloned | expert | Average |
|---|---|---|---|---|
| TD3+BC, paper | 0.0 | 0.0 | 0.0 | 0.0 |
| TD3+BC, our | 0.3 ± 0.4 | 1.1 ± 1.1 | 127.0 ± 0.4 | 42.8 |
| TD3+BC, tuned | 0.3 ± 0.4 (+50.0%) | 1.1 ± 1.1 (-80.0%) | 127.0 ± 0.4 (-5.3%) | 42.8 (-8.1%) |
| ReBRAC w/o LN | 0.2 ± 0.0 (0.0%) | 1.0 ± 1.0 (-81.8%) | 9.9 ± 19.1 (-92.6%) | 3.6 (-92.3%) |
| ReBRAC w/o layer | 0.1 ± 0.0 (-50.0%) | 21.3 ± 19.7 (+287.3%) | 133.1 ± 0.5 (-0.8%) | 51.5 (+10.5%) |
| ReBRAC w/o actor penalty | 0.0 ± 0.0 (-100.0%) | 0.0 ± 0.1 (-100.0%) | 0.0 ± 0.1 (-100.0%) | 0.0 (-100.0%) |
| ReBRAC w/o critic penalty | 0.1 ± 0.1 (-50.0%) | 1.9 ± 0.7 (-65.5%) | 134.1 ± 0.2 (-0.1%) | 45.3 (-2.8%) |
| ReBRAC w large batch | 0.3 ± 0.8 (+50.0%) | 10.6 ± 14.0 (+92.7%) | 133.4 ± 0.5 (-0.6%) | 48.1 (+3.2%) |
| ReBRAC | 0.2 ± 0.2 | 5.5 ± 2.5 | 134.2 ± 0.4 | 46.6 |

Table 40: ReBRAC ablations for relocate tasks. We report final normalized score averaged over 4 unseen training seeds.

| Ablation | human | cloned | expert | Average |
|---|---|---|---|---|
| TD3+BC, paper | 0.0 | 0.0 | 0.0 | 0.0 |
| TD3+BC, our | 0.0 ± 0.0 | -0.1 ± 0.0 | 107.9 ± 0.6 | 35.9 |
| TD3+BC, tuned | 0.0 ± 0.0 (-) | -0.1 ± 0.0 (-105.3%) | 107.9 ± 0.6 (+1.2%) | 35.9 (-0.5%) |
| ReBRAC w/o LN | -0.2 ± 0.0 (-) | 0.0 ± 0.3 (-100.0%) | -0.1 ± 0.0 (-100.1%) | -0.1 (-100.3%) |
| ReBRAC w/o layer | 0.1 ± 0.3 (-) | 1.7 ± 2.1 (-10.5%) | 105.0 ± 3.1 (-1.5%) | 35.6 (-1.4%) |
| ReBRAC w/o actor penalty | -0.1 ± 0.0 (-) | 0.0 ± 0.0 (-100.0%) | -0.1 ± 0.1 (-100.1%) | 0.0 (-100.0%) |
| ReBRAC w/o critic penalty | 0.0 ± 0.1 (-) | 1.9 ± 1.9 (0.0%) | 109.6 ± 1.2 (+2.8%) | 37.1 (+2.8%) |
| ReBRAC w large batch | 0.0 ± 0.0 (-) | 0.1 ± 0.2 (-94.7%) | 109.6 ± 0.9 (+2.8%) | 36.5 (+1.1%) |
| ReBRAC | 0.0 ± 0.0 | 1.9 ± 2.3 | 106.6 ± 3.1 | 36.1 |

