# OpenReview forum: "Revisiting the Minimalist Approach to Offline Reinforcement Learning"
_NeurIPS.cc/2023/Conference — NeurIPS 2023 poster_

### Official Review · Reviewer_DJeX · 2023-06-25

**Soundness:** 3 good
**Presentation:** 3 good
**Contribution:** 2 fair
**Rating:** 5
**Confidence:** 4

**Summary:**

This paper proposes an offline RL algorithm based on TD3+BC and BRAC, integrating popular design elements. The proposed method achieves higher scores on D4RL and V-D4RL datasets.

**Strengths:**

The idea of exploring the popular design elements in offline RL algorithms is interesting. The authors make a great effort in conducting confirmatory experiments including various ablation studies.

**Weaknesses:**

Overall, I see the proposed method as an integrated method of existing algorithms. From my perspective, the novelty and contributions are weak. It may be more insightful if the author put more emphasis on the analyses of existing design elements rather than combining them into one new method. I do not see many insightful analyses in the current manuscript.

**Questions:**

1.	Table 6: ReBRAC w large batch. What does it mean?
2.	The performance improvements of ReBRAC seem marginal, especially in challenging tasks (door-human, door-cloned….)
3.	Table 6: it seems that the critic penalty is not crucial. Is it a universal conclusion, or just due to a lack of hyperparameter tuning?
4.	Can authors provide comparisons of computation costs?

**Limitations:**

Please see the weaknesses.

---

> ### Author Rebuttal · Authors · 2023-08-05
>
> Thank you for the review. We believe that the main concern is the limited novelty, which we address first, and then we move to more precise limitations.
>
> ----------
>
> ### On limited novelty
>
> While we agree that the paper may seem to feature constrained technical novelty, given it doesn't propose new algorithmic techniques but rather builds upon existing ones, we see merit in studies that clarify determinants of performance, be they algorithmic elements, coding strategies or hyperparameter selections. Such work can, in its own right, offer retrospective novelty.
>
> This invites us to ponder over some questions: Were we aware that this unpretentious combination of design choices could eclipse the efficiency of all other methods to this extent? Did we know prior to our study that this kind of method could perform exceptionally well in environments requiring strong stitching competencies (AntMaze, as per our understanding, has generally seen dominance from methods like IQL or CQL with ensembles (MSG))? Can we consider such seminal works as Rainbow [1] or Dreamer-v3 [2] with limited novelty, merely because they integrate known elements?
>
> We hope this dialogue serves to highlight the scope, relevance, and intrinsic merits of our study. As for the analysis of extant design elements, we have taken measures to reference a wealth of existing literature in our description, thereby providing a comprehensive overview. For instance, we pointed out how LayerNorm is instrumental in mitigating catastrophic q-value extrapolations, and that large batches can enhance and balance convergence given adequate data. Even though a more thorough analysis of the precise volume of data needed could constitute a separate study, we admit that it was beyond our current work's scope. The same goes for the increased discount factor that could use more exploration but was not the focus of our paper.
>
> We hope that our response brings more clarity and understanding to the novelty and significance of our research. We look forward to any further comments that could help us improve and refine our work.
>
> --------------------
>
> ### On Questions
>
> > Table 6: ReBRAC w large batch. What does it mean?
>
> ReBRAC with the use of large batch (the definition of the large batch can be found in the paper, essentially, the value is fixed to the 1024). We found the use of large batches to be detrimental on AntMaze datasets.
>
> > The performance improvements of ReBRAC seem marginal, especially in challenging tasks (door-human, door-cloned….)
>
> We highly value your feedback concerning the perceived marginal performance improvements of ReBRAC, especially in challenging tasks.
>
> (1) Although it may seem that the performance gains of our method over SAC-RND on Gym-MuJoCo are minimal, it is essential to note that ReBRAC achieves this level of efficiency sans the requirement for an additional network. Moreover, it surpasses other contenders by a minimum of 10%.
>
> (2) With regard to the AntMaze domain, ReBRAC outperforms the nearest competitor by a margin of 26%, which we believe demonstrates notable superiority.
>
> (3) In the Adroit domain, ReBRAC exhibits an enhanced performance level, outpacing the closest competitor by an average of 9%.
>
> (4) Furthermore, in the V-D4RL test, ReBRAC once again outperforms the nearest competitor, this time by a margin of 16%.
>
> While we acknowledge that we were not able to set a new benchmark in the most challenging tasks in the offline setting, we conducted additional experiments in the offline-to-online setting during the rebuttal phase. These tests revealed that our method exhibits superior performance in the Adroit domain and aligns with Cal-QL in the AntMaze domain. We invite you to refer to the attached .pdf for additional details.
>
> > Can authors provide comparisons of computation costs?
>
> Extensive computational costs comparisons can be found in the appendix. Thank you for noticing, we will add a reference to it in the main text.
>
> > Table 6: it seems that the critic penalty is not crucial. Is it a universal conclusion, or just due to a lack of hyperparameter tuning?
>
> We are not sure what you mean by the lack of hyperparameter tuning in this specific case, could you elaborate on that?

---

> > ### Comment · Reviewer_DJeX · 2023-08-14
> > **Response to the authors**
> >
> > Thanks for the authors' responses. Some of my concerns are addressed. However, in my view, it may be more significant to provide insightful analyses of these added components than merely to present them. I will appreciate it if the authors briefly discuss these components and provide some insights regarding why they work or why they do not work.
> >
> > Additionally, I have two questions:
> >
> > (1) It is noticed that you tune hyperparameters per dataset for each method. I’m a little concerned that the tuning effort you made for the proposed method is larger than the baseline methods. In addition, I do not think the per-dataset tuning is affordable in real-world scenarios. Hence, can the authors provide the tuning logs for baseline methods, or provide the comparing results with unifying hyperparameters?
> >
> > (2) Regarding the online finetuning experiment in the attached pdf: why adroit-human tasks are missing? And can ReBRAC outperform [1]?
> >
> > [1] Ball, Philip J., et al. "Efficient online reinforcement learning with offline data." arXiv preprint arXiv:2302.02948 (2023).

---

> > > ### Author Response · Authors · 2023-08-16
> > >
> > > Thank you for your thoughtful comments and questions.
> > >
> > > To address your first concern about the analyses of added components, we understand and appreciate the emphasis on deeper insight. The depth and nuances of each component certainly warrant dedicated attention. For example, reference [2] delves into the intricacies of large batches in offline RL, while another study [3] examines the dynamics of discount factor value adjustments. The primary goal of our paper was to integrate these insights with the minimalist offline RL framework, represented by TD3 + BC, to demonstrate its potential for state-of-the-art results across varied datasets. We believe the ablation studies, which highlight the individual contributions of each component, provide a valuable perspective on this. Finally, we acknowledge your reference to a concurrent work [1] in question (2), which, to our knowledge, embarks on analogous undertakings in a different setup.
> > >
> > > Regarding your specific questions:
> > >
> > > > (1) It is noticed that you tune hyperparameters per dataset for each method. I’m a little concerned that the tuning effort you made for the proposed method is larger than the baseline methods.
> > > >
> > >
> > > The choice to tune hyperparameters per dataset is consistent with emerging practices in offline RL, exemplified by MSG [4] (8 - 12 parameters sets) and SAC-RND (9 - 12 parameters sets). We ensured that our tuning efforts for ReBRAC were comparable to that for the baselines. Specifically, the grid dimensions for both ReBRAC and IQL are similar (16 - 20 parameters sets), with ReBRAC's performance proving superior. To ensure transparency and provide further context on the influence of hyperparameter tuning, we have shared our results here: **https://openreview.net/forum?id=vqGWslLeEw&noteId=TM1rs0kWh4**. Furthermore, optimal hyperparameters are detailed in Appendix B within the Supplementary Material.
> > >
> > > > In addition, I do not think the per-dataset tuning is affordable in real-world scenarios.
> > > >
> > >
> > > We recognize the real-world challenges of per-dataset tuning. However, it's worth noting that different domains often necessitate distinct hyperparameters. Our work emphasizes the practical utility of ReBRAC, as reflected in the Expected Online Performance (EOP) scores detailed in Table 7. The EOP metric, in particular, provides a nuanced understanding of how performance can vary based on the number of testable policies. As illustrated, ReBRAC exhibits promising results when compared to notable baselines such as IQL, especially when policy budget constraints are considered.
> > >
> > > > (2) Regarding the online finetuning experiment in the attached pdf: why adroit-human tasks are missing? And can ReBRAC outperform [1]?
> > > >
> > >
> > > The choice to use cloned datasets was primarily driven by the availability of reference scores in CORL. Given the inherent similarities between cloned datasets and their human counterparts, we expect only minor performance differences. Regarding the comparison with [1], we see potential in ReBRAC based on its performance relative to Cal-QL, which has been documented to match or surpass [1]. The absence of direct comparisons is due to the lack of numerical results from [1].
> > >
> > > We trust that our clarifications address your concerns and shed light on the novelty and significance of our work.
> > >
> > > Thank you for your time and attention.
> > >
> > > [1] Ball, Philip J., et al. "Efficient online reinforcement learning with offline data." arXiv preprint arXiv:2302.02948 (2023).
> > >
> > > [2] Nikulin, Alexander, et al. "Q-Ensemble for Offline RL: Don't Scale the Ensemble, Scale the Batch Size." arXiv preprint arXiv:2211.11092 (2022).
> > >
> > > [3] Hu, Hao, et al. "On the role of discount factor in offline reinforcement learning." International Conference on Machine Learning. PMLR, 2022.
> > >
> > > [4] Ghasemipour, Kamyar, Shixiang Shane Gu, and Ofir Nachum. "Why so pessimistic? estimating uncertainties for offline rl through ensembles, and why their independence matters." Advances in Neural Information Processing Systems 35 (2022): 18267-18281.

---

> > > > ### Comment · Reviewer_DJeX · 2023-08-18
> > > > **Follow-up Response to the authors**
> > > >
> > > > Thank you for providing clarifications in response to my previous queries. However, after careful consideration, there still remain two key concerns that warrant further discussion and clarification.
> > > >
> > > > (1) Novelty and Contribution:
> > > >
> > > > > The depth and nuances of each component certainly warrant dedicated attention. For example, reference [2] delves into the intricacies of large batches in offline RL, while another study [3] examines the dynamics of discount factor value adjustments. The primary goal of our paper was to integrate these insights with the minimalist offline RL framework, represented by TD3 + BC, to demonstrate its potential for state-of-the-art results across varied datasets.
> > > >
> > > > I appreciate the authors' responses in highlighting the significance of each individual component in the proposed approach. References [2] and [3] certainly delve into the intricate aspects of large batches in offline RL and the dynamics of discount factor value adjustments, respectively. The authors' objective to integrate these insights within the minimalist offline RL framework (TD3 + BC) to demonstrate its potential across diverse datasets is noted.
> > > >
> > > > **Nevertheless, I am still concerned about the overall contribution of the paper.** The combined approach appears to compile **existing validated components** without introducing novel insights. For instance, if the benefit of large batches for offline RL algorithms is already established from [2], this paper seems to lack new perspectives beyond that.
> > > >
> > > > (2) Comparative Experiment Settings:
> > > >
> > > > >   We ensured that our tuning efforts for ReBRAC were comparable to that for the baselines. Specifically, the grid dimensions for both ReBRAC and IQL are similar (16 - 20 parameters sets), with ReBRAC's performance proving superior. To ensure transparency and provide further context on the influence of hyperparameter tuning, we have shared our results here: https://openreview.net/forum?id=vqGWslLeEw&noteId=TM1rs0kWh4. Furthermore, optimal hyperparameters are detailed in Appendix B within the Supplementary Material.
> > > >
> > > > I acknowledge the authors' response regarding the tuning efforts for ReBRAC and the baseline methods.  **However, my query was not directly addressed.** Specifically, I had requested "tuning logs" for both baseline methods and ReBRAC under varying hyperparameter sets. This refers to the results achieved by these methods across different parameter configurations as tuned by the authors. Alternatively, I had sought a comparison of results achieved by these methods under a unified set of hyperparameters.
> > > >
> > > > Should providing these tuning logs or unifying results pose any challenges or limitations, I kindly request that you elaborate on these constraints in order to foster a thorough understanding.

---

> > > > > ### Author Response · Authors · 2023-08-18
> > > > >
> > > > > We thank you for considering our previous response.
> > > > >
> > > > > > References [2] and [3] certainly delve into the intricate aspects of large batches in offline RL and the dynamics of discount factor value adjustments, respectively. The authors' objective to integrate these insights within the minimalist offline RL framework (TD3 + BC) to demonstrate its potential across diverse datasets is noted.
> > > > > >
> > > > > >
> > > > > > **Nevertheless, I am still concerned about the overall contribution of the paper.**
> > > > > >
> > > > >
> > > > > When comparing our work to [2] and [3], it is natural to question whether our contributions indeed bring novel insights. Specifically, with respect to the discount factor, **we uncover a previously unexplored avenue—increasing the discount factor default value can lead to enhanced performance** in certain scenarios.
> > > > >
> > > > > Regarding the utilization of large batches, we acknowledge that the referenced work sought to apply this technique primarily for reducing the number of ensemble networks, thereby accelerating runtime improvements. However, our approach differs in that **we leverage large batches to significantly enhance algorithm performance on a specific domain and no to reduce the ensemble size**.
> > > > >
> > > > > Moreover, our research shows that state-of-the-art performance can be achieved using the combination of the design choices introduced in other works and how strongly they impact.
> > > > >
> > > > > > Specifically, I had requested "tuning logs" for both baseline methods and ReBRAC under varying hyperparameter sets.
> > > > > >
> > > > >
> > > > > Addressing your specific request for "tuning logs" under varying hyperparameter sets, we regret that we are unable to provide these logs in an anonymized and meaningful format during the current stage. However, our paper meticulously reports the Expected Online Performance (EOP) metric [1]. This metric is derived from evaluations of policies across different hyperparameter sets, capturing the anticipated performance based on the number of evaluated policies. Each evaluated policy corresponds to a unique set of hyperparameters that we systematically search through. Thus, EOP provides simple yet powerful comparison of methods performances with the same amount of hyperparameters tested. Detailed per-dataset EOP scores are available in Appendix E of our paper.
> > > > >
> > > > > > Alternatively, I had sought a comparison of results achieved by these methods under a unified set of hyperparameters.
> > > > > >
> > > > >
> > > > > We agree, that such comparison would be valuable extension. However, it a common practice in offline RL that each algorithm propose new parameters which are tuned keeping other untouched. So, it is not possible to introduce the unified set of hyperparameters due to the fact that different hyperparameters must be tuned for different offline RL algorithms.  For example, for ReBRAC we tune the coefficient of BC component in actor and critic losses but there are no such parameter in IQL at all and we tune pairs of $\beta$ and $\tau$ prameters which are introduced in it and tuned by IQL authors. Full sets of hyperparameters tuned are available in Appendix A.
> > > > >
> > > > > We hope that our answer resolves all of the concerns you have.
> > > > >
> > > > > [1] Kurenkov, Vladislav, and Sergey Kolesnikov. "Showing your offline reinforcement learning work: Online evaluation budget matters." International Conference on Machine Learning. PMLR, 2022.
> > > > >
> > > > > [2] Nikulin, Alexander, et al. "Q-Ensemble for Offline RL: Don't Scale the Ensemble, Scale the Batch Size." arXiv preprint arXiv:2211.11092 (2022).
> > > > >
> > > > > [3] Hu, Hao, et al. "On the role of discount factor in offline reinforcement learning." International Conference on Machine Learning. PMLR, 2022.

---

> > > > > > ### Comment · Reviewer_DJeX · 2023-08-21
> > > > > > **Final Response to the authors**
> > > > > >
> > > > > > Thanks for the authors' follow-up responses. Although the major concerns (i.e., overall contribution and per-dataset tuning)  from my previous comment remain, I am inclined to align my score with other reviewers.

---

### Official Review · Reviewer_8F2Q · 2023-07-02

**Soundness:** 3 good
**Presentation:** 3 good
**Contribution:** 3 good
**Rating:** 6
**Confidence:** 2

**Summary:**

This paper revisited recent methods in the offline RL area and discussed how different design choice impact offline RL methods' performance. In particular, the authors focus on four hyper-parameter choices (i) number of network layers, (ii) using LayerNorm, (iii) batch size and (iv) discounting factor $\gamma$, as well as one algorithmic choice (v) using actor/critic regularization, and conducted comprehensive experiments to show the importance of different factors. Their method, ReBRAC, achieved strong empirical performance on 51 datasets.

**Strengths:**

1. The paper presents comprehensive experiments.

2. The proposed method demonstrates strong empirical performance in the conducted experiments.

3. The paper is clearly written.

4. This work shows great engineering values and has the potential to offer valuable guidance in design choices for the offline RL community.

**Weaknesses:**

1. A combination of existing techniques/tricks. In particular, most of the design choices (i) number of network layers, (ii) using LayerNorm, (iii) batch size and (iv) tuning $\gamma$ are hyper-parameters or standard deep NN techniques.

2. Table 6: ReBRAC w/o LN has a 38.0 average score over all domains, which is noticeably lower than the authors' reproduction of TD3+BC (52.2), leading to my speculation that adding LN alone could achieve great empirical performance. It would be nice if TD3+BC w/ LN alone could be shown. (For reference, Rainbow [1] shows the performance of all individual choices in their Figure 1 and none of them were close to Rainbow.)

3. Table 6: Critic penalty only gives marginal improvement. As the batch size, using LN, number of layers, and tuning $\gamma$ are mostly hyper-parameters rather than RL algorithm designs, marginal benefits from the critic penalty further limit its algorithmic contribution.

4. According to table 1, SAC-RND seems to have all five components. It would be nice if the authors could elaborate on the difference between SAC-RND and ReBRAC, for example, in section 2 or 3.


Overall I find this paper has limited novelty, especially considering some components mentioned are merely hyper-parameters. However, I find it might be a significant contribution to achieve superior performance on 51 datasets, providing potential implementation guidance to the offline RL community. I therefore voted for a boarderline acceptance with a low confidence score.


[1] Rainbow: Combining Improvements in Deep Reinforcement Learning. https://arxiv.org/pdf/1710.02298.pdf

**Questions:**

I have no further question as it was clearly written.

**Limitations:**

See above.

---

> ### Author Rebuttal · Authors · 2023-08-05
>
> First of all, we would like to thank you for your time reviewing our paper and valuable comments. We believe that the main concern is the limited novelty, which we address first, and then we move to more precise limitations.
>
> ------------
>
> ### On limited novelty
> We appreciate the reviewer's comments and accept that while our research represents a judicious amalgamation of existing techniques and hyperparameters, the novelty lies in the distinctive combination and application.
>
> Indeed, insights into performance enhancement – whether through algorithmic or code tweaks or hyperparameter tuning – carry intrinsic merit. Our work served to highlight certain empirical results that may otherwise have remained undisclosed. For instance, the fact that our chosen blend of design choices significantly outperforms other methods is noteworthy. Moreover, it was hitherto unknown that this category of methods performs exceptionally well in challenging environments, such as AntMaze. Previously, the prevalent belief favoured the efficacy of methods like IQL or CQL with ensembles (MSG).
>
> Hence, if we scrutinize papers like Rainbow [1] or Dreamer-v3 [2], which are clever syntheses of known elements, should we consider their innovation limited? We hope that this perspective will elucidate our paper's aim and its inherent merits.
>
> We have also augmented our submission with offline-to-online experiments (refer to the .pdf file). These reveal ReBRAC to be a potent baseline in this context, further expanding our contribution's dimensions. Therefore despite its perceived limited novelty, we propose that our paper provides essential guidance for offline RL implementation, thereby contributing significantly to this field of research.
>
> ---------------
>
> ### On other outlined weaknesses
>
> > Table 6: ReBRAC w/o LN has a 38.0 average score over all domains, which is noticeably lower than the authors' reproduction of TD3+BC (52.2), leading to my speculation that adding LN alone could achieve great empirical performance. It would be nice if TD3+BC w/ LN alone could be shown. (For reference, Rainbow [1] shows the performance of all individual choices in their Figure 1 and none of them were close to Rainbow.)
>
> This is a great observation and suggestion. To address it, we include Rainbow-like experiments, i.e., adding each individual choice separately to the TD3+BC (see the attached .pdf file). Notably, LN and deeper networks improve performance of the TD3+BC but still lack far behind the ReBRAC.
>
>
> > According to table 1, SAC-RND seems to have all five components. It would be nice if the authors could elaborate on the difference between SAC-RND and ReBRAC, for example, in section 2 or 3.
>
> Sure, we will add elaboration on the difference in the final version. The main difference is that SAC-RND uses penalization based on the RND bonus in both actor and critic, which requires pre-training a predictor network.
>
> --------------
>
> ### References
>
> [1] Hessel, Matteo, et al. "Rainbow: Combining improvements in deep reinforcement learning." Proceedings of the AAAI conference on artificial intelligence. Vol. 32. No. 1. 2018.
>
> [2] Hafner, D., Pasukonis, J., Ba, J., & Lillicrap, T. (2023). Mastering diverse domains through world models. arXiv preprint arXiv:2301.04104.

---

> > ### Comment · Reviewer_8F2Q · 2023-08-15
> > **Thank you for the response**
> >
> > I would like to thank the authors for the responses and additional experiments.
> >
> > > we propose that our paper provides essential guidance for offline RL implementation, thereby contributing significantly to this field of research.
> >
> > The implementation guidance to the offline RL community was the reason for my original positive vote.
> >
> > > we include Rainbow-like experiments
> >
> > Thanks for the experiments, my concern regarding this point has been addressed.
> >
> > I'll raise my score to 6.

---

### Official Review · Reviewer_HdeU · 2023-07-03

**Soundness:** 4 excellent
**Presentation:** 4 excellent
**Contribution:** 3 good
**Rating:** 7
**Confidence:** 5

**Summary:**

This paper revisits several minor design choices in recent offline RL literature and equips a standard method TD3+BC (or more generally BRAC) with these designs to attain a strong baseline for offline RL with state-of-the-art performance on both D4RL and V-D4RL benchmarks. These critical designs include deeper networks, LayerNorm, larger batches, decoupled actor-critic penalty, and adjusted discount factor. Extensive benchmarking experiments and ablation studies are conducted across a range of domains.

**Strengths:**

1. Careful look into detailed design choices from a large amount of literature, which naturally motivates the proposed solution
2. Extensive experiments across D4RL’s Gym-MuJoCo, AntMaze, and Adroit tasks, and even vision-based V-D4RL tasks
3. Great efforts for a fair comparison with baseline methods, including hyperparameter tuning for baselines and measuring Expected Online Performance
4. Well written

Overall, I really appreciate the contribution to the offline RL community by revisiting a strong yet overlooked baseline method, BRAC.

**Weaknesses:**

1. A probably incorrect assertion to a technical detail of BRAC (see Question 1 below)
2. Limited technical novelty (but I think it doesn't matters for this paper focusing on a retrospective analysis)

**Questions:**

1. If I understand correctly, BRAC has adopted actor penalization and critic penalization at the same time. Indeed, BRAC-v in the original paper adds a penalty to both actor and critic learning objectives (Eq. 6 and 7 in the BRAC paper). Thus, some assertions in this paper (Lines 65 and 111) are incorrect and should be revised. Nevertheless, the idea of decoupling actor and critic penalty has not been explored to my knowledge.
2. Following the above question, an ablation study comparing decoupled actor-critic penalties and coupled ones (using only one hyperparameter) is missing in Table 6. Can authors present these results?
3. What do you mean by "Since our approach principally builds upon TD3+BC, the differences in their performances should be considered the most important ones." in line 124?
4. In line 185, the authors claim, "ReBRAC outperforms TD3+BC not because of different implementations or actor regularization parameter choice". However, in my opinion, the differences between  ReBRAC and TD3+BC are indeed implementation choices rather than algorithmic innovations. I understand that the efforts of building ReBRAC upon TD3+BC are non-trivial, but the authors should consider revising this sentence to make it much clearer.
5. Implementation details for vision-based V-D4RL tasks, including CNN architecture to encode visual observations, are missing in both the main paper and the appendix. Although I have found the implementations in supplementary source code, I recommend clarifying these details in a future revision of the paper.
6. TD3+BC uses state normalization on Gym-MuJoCo tasks. Did the authors include this implementation detail in their experiments for both TD3+BC and ReBRAC? Since in the TD3+BC paper, state normalization does not impact significantly, in my opinion, it is okay not to include it, but I recommend the authors clarify this in the paper.

**Limitations:**

This work has discussed its limitations and future work in the conclusion. There does not seem to be any negative social impact of this paper that should be discussed.

---

> ### Author Rebuttal · Authors · 2023-08-05
>
> Thank your for the review and identified weaknesses, we address them and your questions as follows:
>
> > If I understand correctly, BRAC has adopted actor penalization and critic penalization at the same time. Indeed, BRAC-v in the original paper adds a penalty to both actor and critic learning objectives (Eq. 6 and 7 in the BRAC paper). Thus, some assertions in this paper (Lines 65 and 111) are incorrect and should be revised. Nevertheless, the idea of decoupling actor and critic penalty has not been explored to my knowledge.
>
> You're correct in your assessment, we will revise our statements in the final version of the paper.
>
> > Following the above question, an ablation study comparing decoupled actor-critic penalties and coupled ones (using only one hyperparameter) is missing in Table 6. Can authors present these results?
>
> Yes, please, see the attached .pdf file in the official comment. Notably, performance with coupled penalties drops by a small margin when compared to decoupled ones. We will make sure to include this information in the appendix, as this significantly reduces the hyperparameter space search and should improve the expected online performance on smaller budgets when compared to other methods.
>
> > What do you mean by "Since our approach principally builds upon TD3+BC, the differences in their performances should be considered the most important ones." in line 124?
>
> This was meant to highlight that ReBRAC improvements do not simply come from the actor-regularization hyperparameter search for TD3+BC. Hopefully, this clarifies what we meant, we will make sure to update the corresponding text.
>
> Moreover, we also included results for experiments where we add one design choice at a time to the TD3+BC to further demonstrate the preference of the found combination of choices.
>
> > Implementation details for vision-based V-D4RL tasks, including CNN architecture to encode visual observations, are missing in both the main paper and the appendix. Although I have found the implementations in supplementary source code, I recommend clarifying these details in a future revision of the paper.
>
> Sure, we will provide a clear reference to the original architecture appearence and will elaborate on it in the appendix.
>
> > TD3+BC uses state normalization on Gym-MuJoCo tasks. Did the authors include this implementation detail in their experiments for both TD3+BC and ReBRAC? Since in the TD3+BC paper, state normalization does not impact significantly, in my opinion, it is okay not to include it, but I recommend the authors clarify this in the paper.
>
> As you rightfully noted, state normalization does not impact performance much (this is what we also observed in the preliminary experiments), therefore we decided not to use it -- keeping in mind that this may complicate further application in the offline-to-online setting which we did not cover in the original submission. To further illustrate the use, we conducted a set of experiments in the offline-to-online setting and found ReBRAC to be a strong competitior, especially on the Adroit domain (for results, again, see the attached .pdf).

---

> > ### Comment · Reviewer_HdeU · 2023-08-11
> >
> > I thank the authors for their responses. Most of my issues are solved.
> >
> > I will maintain my score since I did not have a major concern with the paper. I really appreciate the effort made for the paper and wish the authors good luck with the submission.

---

### Official Review · Reviewer_7C4n · 2023-07-07

**Soundness:** 4 excellent
**Presentation:** 4 excellent
**Contribution:** 3 good
**Rating:** 7
**Confidence:** 4

**Summary:**

In this work, a number of recent advancements are added to the minimalist TD3+BC baseline, and the authors found that the resulting algorithm leads to a new SOTA performance on D4RL benchmark with raw state and visual input. Extensive empirical results and ablations are provided and the authors show a dedication to fairness and reliability of their comparisons.

**Strengths:**

**originality**
- the paper studies existing methods, and the novelty of the work is main on its empirical findings and ablations. However, due to the extensive experiments and attention to fairness and details, these results can be considered novel findings.

**quality**
- quality is great, paper is well structured, important things are highlighted and performance changes are quantified.
- technical details are given, things like computational speed discussed, code provided, overall good.
- authors even tuned and tested on different seeds.

**clarity**
- paper is very clear and easy to read.

**significance**
- Finding a new sota baseline for offline RL setting, with a full set of ablations understanding the impact of each of its components can be a good and significant contribution.
- Although no new algorithm is introduced, these results can be very helpful to researcher who want to really understand what is helping the performance (whether it's an algorithmic component or just a code hack or hyperpararmeter choice).

**Weaknesses:**

**originality**
- the paper focuses on an empirical study of existing methods, which will reduce the novelty a bit.

Other than this, I think within its own scope, the paper is very good. No major concern.

**Questions:**

- The authors mentioned that to be fair, the competing methods have been extensively tuned, I wonder how much performance gain you are able to achieved for these competing methods compared to their reported results in their original papers?
- when you tune for d4rl, do you fine-tune a different set of hyperparamter for each task? Is this done for all algorithms in the comparisons?

**Limitations:**

Authors discussed the limitations of the work.

---

> ### Author Rebuttal · Authors · 2023-08-05
>
> Thank you for the review. Regarding your questions:
>
> > when you tune for d4rl, do you fine-tune a different set of hyperparamter for each task? Is this done for all algorithms in the comparisons?
>
> Yes, we tune hyperparameters per-dataset for each method.
>
> > The authors mentioned that to be fair, the competing methods have been extensively tuned, I wonder how much performance gain you are able to achieved for these competing methods compared to their reported results in their original papers?
>
> Here are the scores over four training seeds. The "paper" label stands for hyperparameters taken from the original papers and the "tuned" label stands for the best hyperparameters we found. As you can see below, the gains from per-dataset hyperparameter tuning are significant. This is often done for new methods and missed for the baselines considered. We believe this is expected, as different datasets probably benefit from varied levels of penalization.
>
> | **Dataset**               | **IQL, paper** | **IQL, tuned** | **TD3 + BC, paper** | **TD3 + BC, tuned** |
> |---------------------------|----------------|----------------|---------------------|---------------------|
> | halfcheetah-random        | 9.4            | 19.6           | 2.2                 | 30.1                |
> | halfcheetah-medium        | 48.3           | 50.2           | 44.6                | 55.4                |
> | halfcheetah-expert        | 96.4           | 96.6           | 93.8                | 95.5                |
> | halfcheetah-medium-expert | 94.7           | 94.7           | 91.9                | 91.9                |
> | halfcheetah-medium-replay | 44.4           | 45.0           | 40.5                | 45.1                |
> | halfcheetah-full-replay   | 74.9           | 75.6           | 69.3                | 74.1                |
> | hopper-random             | 7.5            | 11.8           | 10.3                | 10.3                |
> | hopper-medium             | 67.5           | 67.5           | 53.2                | 57.6                |
> | hopper-expert             | 100.0          | 112.7          | 108.7               | 110.7               |
> | hopper-medium-expert      | 80.7           | 112.4          | 75.8                | 106.2               |
> | hopper-medium-replay      | 97.4           | 97.6           | 64.5                | 64.5                |
> | hopper-full-replay        | 104.4          | 108.4          | 49.9                | 106.2               |
> | walker2d-random           | 4.0            | 12.0           | 4.5                 | 4.5                 |
> | walker2d-medium           | 80.9           | 82.5           | 77.1                | 77.1                |
> | walker2d-expert           | 112.8          | 113.8          | 109.1               | 110.1               |
> | walker2d-medium-expert    | 111.7          | 112.4          | 108.9               | 110.2               |
> | walker2d-medium-replay    | 82.1           | 83.0           | 50.9                | 58.8                |
> | walker2d-full-replay      | 97.7           | 98.2           | 86.7                | 89.4                |
> | **Gym-MuJoCo avg**        | 73.0           | 77.4 __(+6%)__           | 63.4                | 72.0 __(+13%)__               |
> | antmaze-umaze             | 76.0           | 84.2           | 62.0                | 62.0                |
> | antmaze-umaze-diverse     | 59.5           | 75.0           | 48.0                | 48.0                |
> | antmaze-medium-play       | 69.7           | 70.2           | 0.0                 | 39.0                |
> | antmaze-medium-diverse    | 63.0           | 64.7           | 0.5                 | 18.5                |
> | antmaze-large-play        | 41.5           | 41.5           | 0.0                 | 0.2                 |
> | antmaze-large-diverse     | 22.5           | 29.5           | 0.5                 | 0.5                 |
> | **Antmaze avg**           | 55.3           | 60.8 __(+10%)__           | 18.5                | 28.0 __(+80%)__                |
> | pen-human                 | 87.1           | 93.6           | 65.9                | 77.6                || pen-cloned                | 73.2           | 89.4           | 78.1                | 78.1                |
> | pen-expert                | 130.9          | 134.6          | 144.9               | 144.9               |
> | door-human                | 3.5            | 7.0            | 0.0                 | 0.0                 |
> | door-cloned               | 1.0            | 2.8            | 0.4                 | 0.4                 |
> | door-expert               | 106.0          | 106.5          | 102.5               | 105.8               |
> | hammer-human              | 1.5            | 2.5            | 0.3                 | 0.3                 |
> | hammer-cloned             | 1.4            | 4.1            | 1.1                 | 1.1                 |
> | hammer-expert             | 127.7          | 130.0          | 127.0               | 127.0               |
> | relocate-human            | 0.0            | 0.6            | 0.0                 | 0.0                 |
> | relocate-cloned           | 0.0            | 0.2            | -0.1                | -0.1                |
> | relocate-expert           | 106.0          | 108.2          | 107.9               | 107.9               |
> | **Adroit avg**            | 53.1           | 56.6 __(+6%)__           | 52.3                | 53.5 __(+2%)__                |
> ----------

---

> > ### Comment · Reviewer_7C4n · 2023-08-13
> > **Thank you for the rebuttal**
> >
> > Thank you to the authors, paper looks solid, I'm increasing my score to 7.

---

### Author Rebuttal · Authors · 2023-08-05

We would like to thank the reviewers for their work, hopefully, we provided a sufficient level of response to all, if not, we're open to continue our discussion. Here, we include additional results requested by the reviewers explicitly or implicitly so in order to provide comprehensive empirical results for our answers:
- Adding design elements separately to the TD3+BC
- Ablating decoupled penalization
- Offline-to-online experiments

---

### Decision · Program_Chairs · 2023-09-21

**Decision:**

Accept (poster)

**Comment:**

The paper explores simple modifications to improve offline RL algorithms. The paper is notable for its extensive and well-designed experiments provide a number of insights and advancements.